# BENCHMARKING LLMS' SWARM INTELLIGENCE

## ABSTRACT

Large Language Models (LLMs) show reasoning potential, but their capacity for emergent coordination in Multi-Agent Systems (MAS) under strict swarm-like constraints (e.g., limited local perception and communication) remains unexplored. Existing benchmarks often overlook the challenges of decentralized coordination with incomplete spatio-temporal information. We introduce **SwarmBench**, a benchmark to systematically evaluate the swarm intelligence of LLMs as decentralized agents. SwarmBench features five MAS coordination tasks (Pursuit, Synchronization, Foraging, Flocking, Transport) in a 2D grid where agents rely on local sensory input ($k \times k$ view) and local communication. We propose metrics for coordination effectiveness and analyze emergent group dynamics. Zero-shot evaluations of leading LLMs (e.g., `deepseek-v3`, `o4-mini`) reveal task-dependent performance variations. While showing rudimentary coordination, current LLMs struggle with long-range planning and adaptive strategy formation under decentralized uncertainty. Assessing LLMs under such constraints is crucial for their application in future decentralized systems. We release SwarmBench as an open, extensible toolkit with environments, prompts, evaluation scripts, and comprehensive datasets. It aims to foster research into LLM-based MAS coordination under severe informational decentralization.

## 1 INTRODUCTION

The language capabilities of LLMs (Zhao et al., 2023) have spurred their use as autonomous agents for perception, tool use, and collaboration (Xi et al., 2025; Gao et al., 2024). Research now investigates their collaborative potential in tasks requiring spatial reasoning, connecting to broader studies of collective intelligence (Woolley et al., 2010). However, current evaluations often focus on individual skills or multi-agent scenarios with ample communication, global visibility, or predefined structures (Zhu et al., 2025; Sun et al., 2025; Chen et al., 2023). These settings sidestep the fundamental challenge of coordination under the severe decentralized constraints found in natural swarms.

Inspired by swarm research, a key unexplored question is whether LLM-driven agents can coordinate effectively under the strict decentralization, limited local perception, and minimal local communication of natural swarms. This principle defines Swarm Intelligence, which studies how complex group behaviors arise from simple, local interactions (Bonabeau et al., 1999). While natural systems (e.g., army ants forming structures (Reid et al., 2015; Lutz et al., 2021), locusts marching (Buhl et al., 2006; Sayin et al., 2025), microswimmers forming vortices (Wang et al., 2023)) and seminal simulations (e.g., Reynolds' flocking model (Reynolds, 1987)) demonstrate emergent global patterns from local rules, it is unknown if sophisticated LLMs can achieve comparable collective action under such decentralized constraints. This paradigm is applied in Swarm Robotics for tasks like shape formation with simple robots (Rubenstein et al., 2014; Zhu et al., 2024).

Existing benchmarks often bypass these classical swarm intelligence constraints by focusing on individual skills (Wang et al., 2024a; Paglieri et al., 2024; Ruoss et al., 2025), richer communication (Zhu et al., 2025;

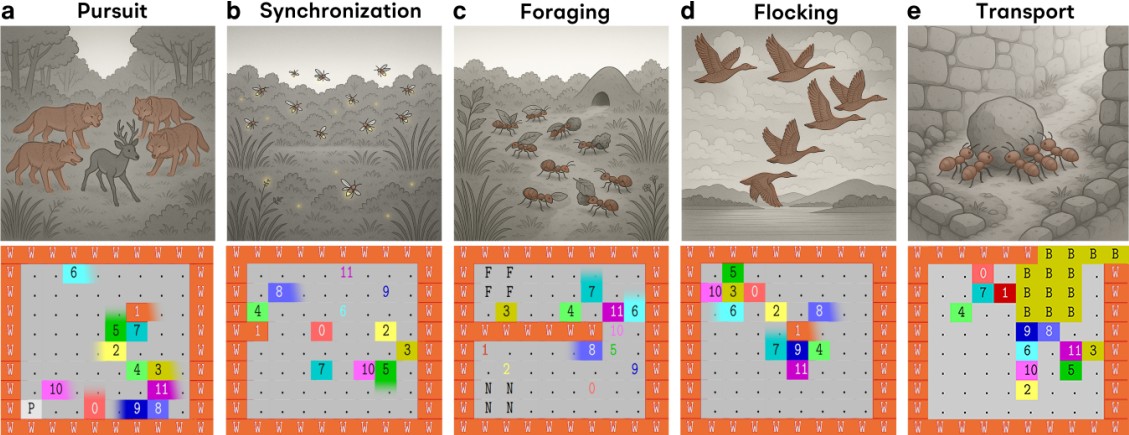

Figure 1: **Swarm Intelligence: Natural Inspiration and SwarmBench Tasks. Top row:** Examples of collective behavior in nature driven by local interactions: **a.** Cooperative wolf pursuit, **b.** firefly synchronization, **c.** ant foraging (Reid et al., 2015; Lutz et al., 2021), **d.** bird flocking (Reynolds, 1987), and **e.** cooperative ant transport. **Bottom row:** Corresponding abstract tasks simulated in SwarmBench's 2D grid environment, depicting agents (represented by colored squares) facing analogous coordination challenges involving the prey (P), food (F), nests (N), and obstacles (B), constrained by walls (W). Agents rely solely on *local* perception and communication, providing a testbed for emergent decentralized coordination. Detailed SwarmBench environment definition and examples can be found in Appendix A and Appendix D respectively. Replay videos can be found in Supplementary Materials (see Supplementary Videos)

Sun et al., 2025; Agashe et al., 2023), or imposed structures (Guo et al., 2024a; Dong et al., 2024), rather than the emergent decentralized coordination under informational limitations that SwarmBench targets. Consequently, whether LLM collectives can exhibit complex swarm phenomena (e.g., the role of noise or diversity (Guo et al., 2024a; Raoufi et al., 2023; Yates et al., 2009) under such conditions) is a critical open question, highlighting a gap SwarmBench aims to address.

We introduce SwarmBench to evaluate the emergent coordination (i.e., group order from local rules without central control) of LLM agents in a decentralized swarm. Inspired by ARC-AGI (Chollet et al., 2025) and SnakeBench (Kamradt, 2025), SwarmBench presents five fundamental coordination challenges (e.g., Pursuit, Synchronization, Foraging, Flocking, and Transport) within a flexible 2D grid world. Agents operate with restricted local perception and minimal local communication, forcing them to develop implicit coordination strategies from local cues. We propose metrics to quantify task success, efficiency, and emergent collective behavior, including behavioral diversity.

The SwarmBench framework (Fig. 2) was used to conduct extensive zero-shot evaluations of several prominent LLMs. Our contributions are:

- **SwarmBench**: A novel benchmark grounded in swarm intelligence constraints, designed to assess emergent decentralized coordination in LLM swarms under strict perception and communication constraints.

- A systematic **evaluation** of contemporary LLMs on SwarmBench, characterizing their current abilities and limitations in canonical swarm scenarios.

- An **analysis** of emergent group dynamics, connecting LLM swarm behavior (e.g., behavioral variability, failure modes) to established collective intelligence concepts.

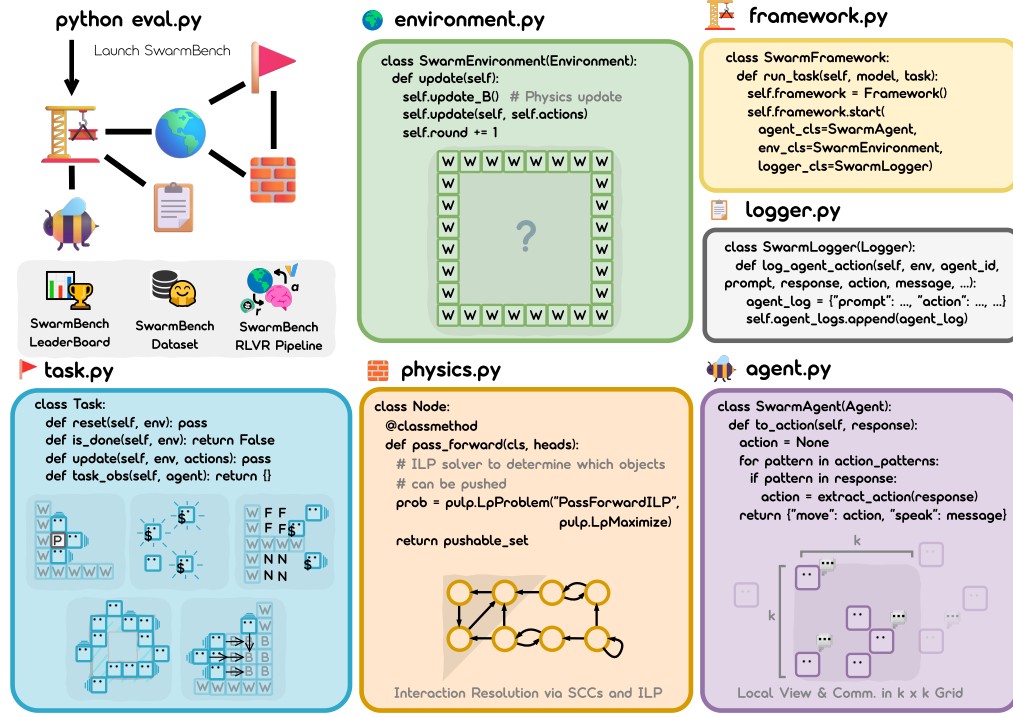

Figure 2: **Conceptual Architecture of SwarmBench.** The diagram shows SwarmBench's modular design. It orchestrates task, environment, physics, LLM-agents, and logger to benchmark LLM swarm intelligence, generate agent-environment interaction datasets, and serve as a swarm intelligence RLVR (Reinforcement Learning with Verifiable Rewards) environment. Our codes can be found in Supplementary Materials (see Supplementary Codes 1)

- An **open-source toolkit** including the physical system (Appendix B), environments, prompts, evaluation scripts, and generated datasets (Appendix K), to facilitate reproducible research into LLM-based swarm intelligence.

Our findings indicate that while LLMs show basic coordination potential, they struggle significantly with long-range planning and robust spatial reasoning under severe decentralization. SwarmBench provides a dedicated platform to measure progress and guide future research towards developing LLMs capable of more robust collective intelligence in decentralized settings operating under local information constraints. Understanding such capabilities is vital, given the growing focus on collective behavior in artificial and human systems (Cheong & Jones, 2021; Bak-Coleman et al., 2021).

## 2 RELATED WORK

**Swarm Intelligence and Self-Organization**    Swarm intelligence studies emergent group behaviors (e.g., ant bridges (Reid et al., 2015; Lutz et al., 2021), locust marches (Buhl et al., 2006), bird flocks (Reynolds, 1987), microswimmer vortices (Wang et al., 2023)) from local interactions of simpler individuals, inspiring swarm robotics (e.g., Kilobots (Rubenstein et al., 2014; Zhu et al., 2024), ARGoS (Pinciroli et al., 2018),

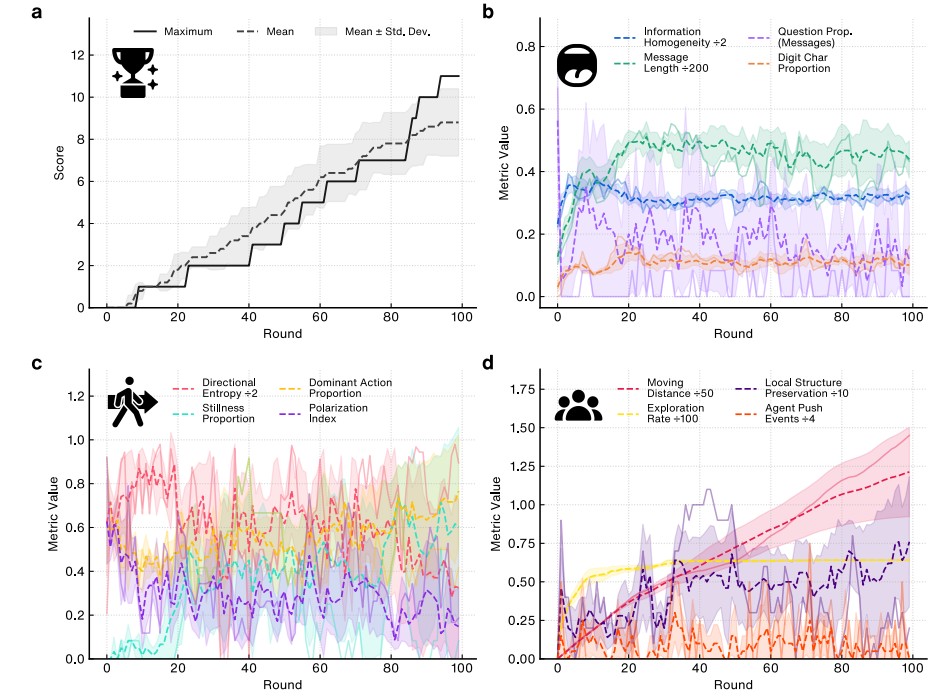

Figure 3: **Metrics for `gemini-2.0-flash` on the Pursuit task.** This figure illustrates score progression and various group dynamic metrics. Detailed definitions for all metrics are provided in Appendix F. Comprehensive visualizations for all evaluated models and tasks can be found in Fig. S.36–S.99, Appendix M. The shaded area represents the standard deviation of five simulation runs on this task.

Kilombo (Jansson et al., 2015), self-organizing neural systems (Zhu et al., 2024)). SwarmBench applies these core constraints (local sensing, minimal communication) to LLM agents while also enabling the study of factors like diversity/noise (Yates et al., 2009; Raoufi et al., 2023), making its focus on behavioral emergence distinct from data processing approaches like Swarm Learning (Warnat-Herresthal et al., 2021).

**LLM-Driven Multi-Agent Systems**    LLMs as agent decision-makers (Xi et al., 2025; Wang et al., 2024b) are growing, applied to MAS in diverse contexts from software development (Qian et al., 2023; Hong et al., 2023) and scientific discovery (Gottweis et al., 2025; Boiko et al., 2023) to social simulation (Park et al., 2023; Gao et al., 2024; AL et al., 2024; Yang et al., 2024) and code generation (Ishibashi & Nishimura, 2024). While promising for cooperation and Theory of Mind (Li et al., 2023; Woolley et al., 2010), with imposed structures aiding efficiency (Guo et al., 2024a), many studies use richer communication or pre-defined roles (Li et al., 2025), unlike SwarmBench's focus on decentralized emergence from local constraints with potential noise/limited propagation (Sharma et al., 2023).

**Benchmarking LLM Coordination and Spatial Reasoning**    Existing MAS benchmarks for LLMs (e.g., using cooperative games (Agashe et al., 2023; Sun et al., 2025; Wu et al., 2024), competitive games (Kamradt, 2025), or complex task simulations (Zhu et al., 2025; Dong et al., 2024; Park et al., 2023)) often differ from SwarmBench by not strictly imposing classical swarm intelligence constraints, thus not directly testing emergent coordination from severe decentralization. Foundational reasoning benchmarks (Wang et al., 2024a;

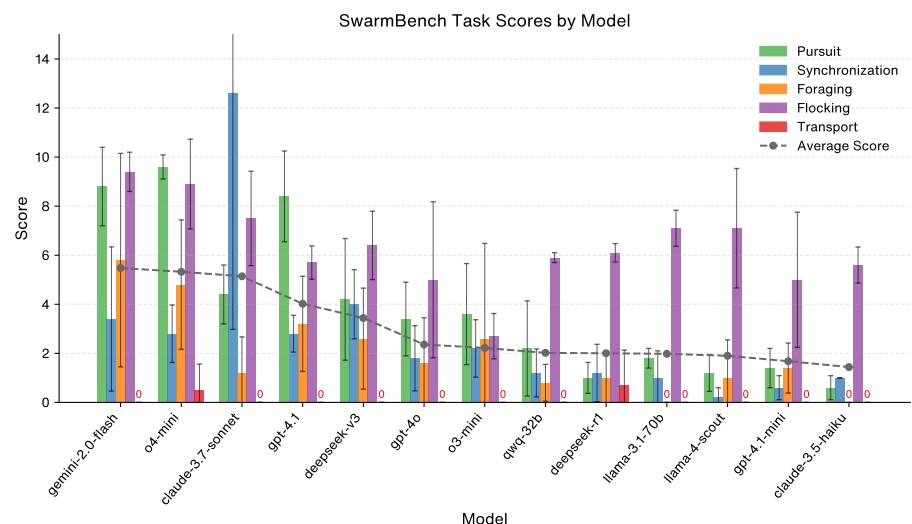

Figure 4: **Overview of LLM Performance on SwarmBench Tasks.** Average scores by LLMs across five core tasks. Bars: mean score over 5 runs. The difficulty of these five tasks varies. It is worth noting that even for the seemingly simple Transport task (i.e., Moving a large, irregularly shaped obstacle blocking the map exit by following appropriate steps, see Fig. S.12), only `o4-mini` and `deepseek-r1` were able to achieve a non-zero average score. Performance varies significantly by model and challenge. Details in Table S.1, Appendix E.

Paglieri et al., 2024; Ruoss et al., 2025) also diverge, and LLMs reportedly struggle with some multi-agent patterns like flocking (Li et al., 2024). SwarmBench concentrates on emergent decentralized coordination, adopting classical swarm intelligence constraints and analyzing collective dynamics.

**LLM-Driven Coordination in Embodied Simulations** While other embodied LLM research (Kannan et al., 2024; Yu et al., 2023; Guo et al., 2024b; Zhang et al., 2024c; Mandi et al., 2024; Chen et al., 2024; Zhang et al., 2024a;b; Garg et al., 2024; Liu et al., 2024) tackles complex, application-specific scenarios, often with richer sensory inputs or communication, SwarmBench provides a complementary evaluation. It focuses on fundamental swarm intelligence constraints: emergent coordination from decentralized LLMs under severe local constraints in a simplified, extensible 2D grid world, aligning with explorations of direct LLM integration in individual swarm robots (Strobel et al., 2024).

## 3 SWARMBENCH

To evaluate LLM capacity for emergent decentralized coordination under swarm intelligence constraints, we introduce SwarmBench (Fig. 2). This benchmark offers multi-agent tasks in a configurable 2D grid-world, coupled with standardized evaluation protocols. SwarmBench emphasizes scenarios where agents possess only limited local perception and communication, necessitating emergent collective strategies from decentralized interactions. Further details are in Appendix A.

**Environments** SwarmBench employs a 2D grid world, underpinned by a customizable physics engine (detailed in Appendix B) that simulates multi-object dynamics and interactions. It features five core multi-

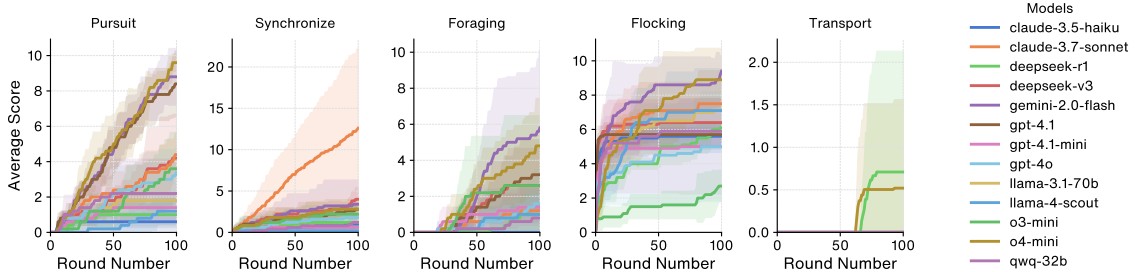

Figure 5: **LLM Score Progression on SwarmBench Tasks Over Time.** Average task score accumulation over 100 rounds. Lines: mean score trajectory; shaded areas: std. dev. Illustrates performance dynamics.

agent tasks: Pursuit, Synchronization, Foraging, Flocking, and Transport. These tasks (Fig. 1, Appendix A, D, and Supplementary Videos) probe different facets of emergent swarm behavior under local constraints. The extensible framework supports procedural generation of instances.

**Observations, Actions, and Communication**  Agents operate with restricted local perception (a $k \times k$ view of their immediate surroundings, their own status, and any messages received in the previous round). Based on this, they output a primary action (e.g., movement, task-specific interaction) and, optionally, a short, anonymous message for local broadcast. This forces reliance on local cues and implicit coordination (details in Appendix A, prompt in Appendix C).

**Evaluation Setting and Models**  We use a zero-shot protocol: each agent is an independent LLM instance, with memory managed via prompt. SwarmBench is model-agnostic. Our main experiments (Section 4) use several contemporary LLMs with fixed sampling parameters (`temperature=1.0, top_p=1.0`) to ensure comparability, though we find that performance on some dynamic tasks can benefit from higher diversity (see Appendix L.7 for a detailed sensitivity analysis). Further methodology details are in Appendix A.

**Evaluation Metrics**  To quantify emergent collective behaviors, we compute metrics from agent positions, messages, and actions, capturing behavioral/language diversity and movement coordination. These metrics (Appendix F, Fig. 3) facilitate analysis of emergent strategies and performance.

## 4 RESULTS

To contextualize the performance of LLM-driven swarms, we first compared them against a range of rule-based and heuristic agents. While specialized heuristics could match LLM performance on simpler, decomposable tasks like `Pursuit`, LLMs demonstrated superior generalist capabilities, outperforming all baselines on tasks requiring more flexible adaptation such as `Foraging` and `Synchronization` (see Appendix L.1 for full details). This highlights the potential of LLMs to handle diverse coordination challenges without task-specific engineering.

Following this baseline comparison, we evaluated thirteen LLMs on SwarmBench's five tasks in a zero-shot protocol, where agents operated with a $5 \times 5$ local view and local communication (see Fig. 4). All simulation data are logged for release (Appendix K). The performance, averaged over five runs, reveals significant variation across both models and tasks, highlighting the inherent difficulty of achieving decentralized coordination under strict local constraints.

In addition to these core evaluations, we conducted further analyses to probe the limits of our framework and the LLM swarms. We confirmed the framework's scalability for larger-scale research (Appendix L.2), tested the swarms' robustness against communication noise and delays (Appendix L.4), and compared their performance to a centralized variant with a global view (Appendix L.3). These supplemental results underscore the nuanced challenges of decentralized control, revealing, for instance, that global information offers little advantage in tasks requiring complex spatial micromanagement like Pursuit.

### 4.1 TASK PERFORMANCE COMPARISON

As shown in Figs. 4 and 5, performance varied significantly across both LLMs and tasks (details in Appendix E, Table S.1). No single LLM consistently excelled, indicating that the challenges of decentralized coordination are highly task-specific. For instance, Flocking was the highest-scoring task overall, while Synchronization showed the most divergence. Models like gemini-2.0-flash and o4-mini thrived in spatial tasks (Pursuit, Foraging), whereas claude-3.7-sonnet was a top performer in Synchronization. This demonstrates that successful coordination hinges on the emergence of task-appropriate strategies from purely local information.

### 4.2 ANALYSIS OF EMERGENT GROUP DYNAMICS AND COMMUNICATION CORRELATES

Our analysis revealed that physical group dynamics (e.g., behavioral variability, movement efficiency) were strong indicators of task success (metrics in Appendix F, visualizations in Appendix G). In contrast, the semantic content of communication (e.g., coordinate sharing, message homogeneity) showed much weaker correlations. This suggests that under SwarmBench's constraints, effective coordination emerges implicitly from agents observing each other and the environment, rather than from the explicit content of their broadcasts. Indeed, further analysis shows that while agents often converge towards a simplified communication protocol, this convergence is not always beneficial; in fact, for complex tasks, it can negatively correlate with success, suggesting that maintaining communicative diversity is crucial (Appendix L.6). We further investigate the direct influence of these information sources on agent decision-making in Section 4.4.

### 4.3 ANALYSIS OF FAILURE MODES

Common failure modes included repetitive suboptimal behaviors, insufficient coordinated action, and inefficient exploration (Appendices G, E). Figure 6 illustrates several key patterns.

For instance, in "Movement Bias" (Fig. 6a), agents like those from claude-3.5-haiku exhibit a strong directional preference, causing them to neglect global objectives. Another common issue is the "Information Silo" (Fig. 6b), where a subgroup clusters locally but fails to coordinate with the broader swarm. "Traffic Jams" (Fig. 6c) also frequently occurred, with agent over-aggregation obstructing movement and access to resources. Finally, "Memory of a goldfish" (Fig. 6d) reflects poor long-term spatial recall, likely due to a constrained memory buffer (last 5 frames), forcing agents to re-explore forgotten locations.

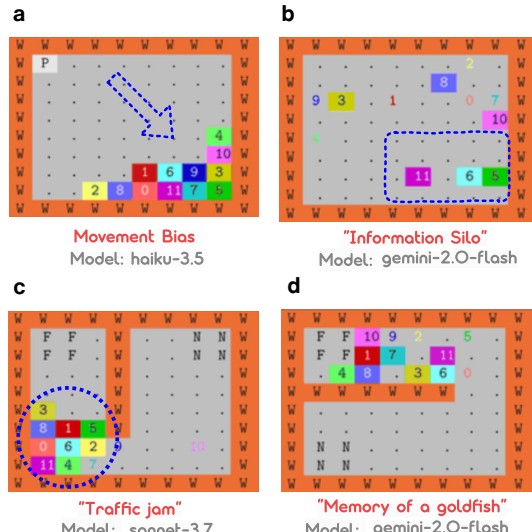

Figure 6: Illustrative LLM agent failure modes in SwarmBench.

These failure modes highlight challenges in robust planning under the uncertainty of local views, often preventing useful communication from translating into effective action. Moreover, these qualitative patterns can be quantitatively linked to established theories in collective behavior (see Appendix L.5).

## 4.4 ACTION ATTRIBUTION ANALYSIS

To quantify the influence of different information sources, we trained a Random Forest classifier (Breiman, 2001) to predict agent actions based on embeddings of their local observation and received messages. We then used permutation importance to assess the relative influence of each feature type (Fig. 7).

Across several tasks, messages received from other agents showed higher permutation importance than visual observations in predicting an agent's next action (details in Appendix J). This indicates that communicated information strongly influences immediate, local decisions.

This finding presents a compelling contrast with our earlier results (Section 4.2): while communication has a potent *tactical* impact on individual actions, its *strategic* value for emergent group performance appears limited in our zero-shot setting. This highlights a critical disconnect between local influence and global effectiveness under SwarmBench's decentralized constraints, a point we elaborate on in Section 5.

## 4.5 IMPACT OF AGENT DENSITY AND PERCEPTION RANGE

We investigated the impact of agent density ($N$) and local perception range ($k \times k$ view) on LLM swarm coordination. Methodologies and full results are in Appendix I.

These parameters influenced task performance in a task-dependent manner. Increasing perception from $k = 3$ to $k = 5$ generally improved outcomes for tasks like `Pursuit`, `Synchronization`, `Foraging`, and `Flocking`. However, further expansion to $k = 7$ had diminishing returns and sometimes decreased performance (e.g., in `Transport` vs. $k = 5$). This suggests a trade-off, as a broader view might increase the LLM's reasoning complexity. A larger input could obscure critical local cues with less relevant information, diluting the agent's focus on features needed for tightly coupled maneuvers, as required in the `Transport` task. This

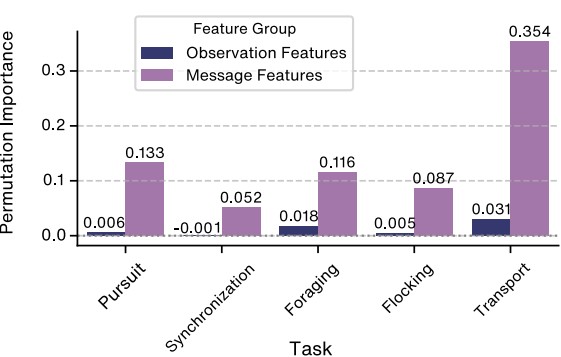

Figure 7: **Aggregated feature importance by task.** Permutation importance for observation and message (from other agents) features. Higher values indicate greater importance in predicting agent actions.

non-monotonic performance improvement is further discussed in Section 5.

The effect of agent number ($N$) also varied. The `Transport` task benefited from a larger group ($N = 16$ outperformed $N = 8$), as it relies on cumulative force. Conversely, `Foraging` performance degraded with more agents, likely due to congestion. `Pursuit` showed peak performance at an intermediate size ($N = 12$), while the `Synchronization` task shows the opposite pattern. These diverse scaling behaviors highlight the challenge of maintaining coordination as group size changes and underscore the need for adaptive strategies in LLM swarms. Figure 8 summarizes these trends.

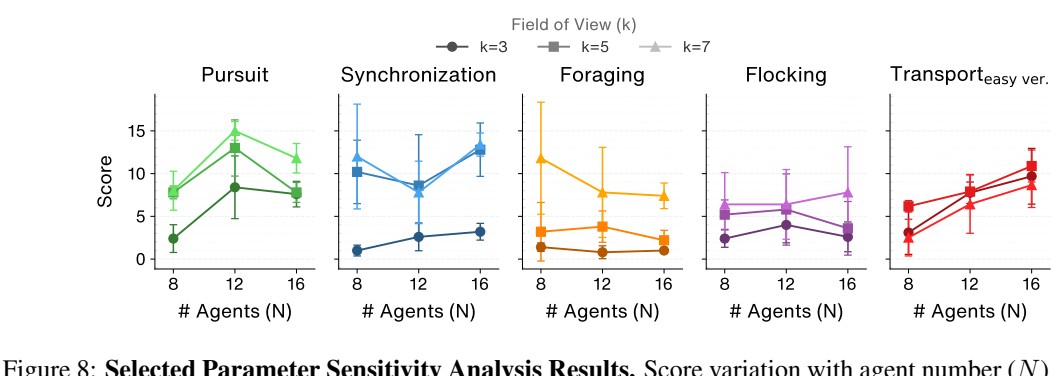

Figure 8: **Selected Parameter Sensitivity Analysis Results.** Score variation with agent number ($N$) and field of view ($k \times k$) for key tasks. This figure is a condensed representation of findings detailed in Appendix I. Transport task use the case where the obstacle is a solid square (rather than the more difficult irregular obstacle).

## 5 DISCUSSION

SwarmBench evaluations show LLM swarm success under decentralization hinges on emergent physical coordination. Physical dynamics metrics (Appendices F, G) strongly correlated with task outcomes, while explicit message content correlated weakly. LLMs thus coordinate implicitly by observing their environment and peers, resembling natural swarms (Bonabeau et al., 1999).

Action attribution analysis reveals a disconnect: messages influence local actions (Sec. 4.4) but fail to improve global success, as their content correlates weakly with outcomes (Sec. 4.2), indicating a lack of strategic depth. Coordination is thus determined by emergent physical patterns. Furthermore, sensitivity to agent density ($N$) and perception range ($k$) (Appendix I) underscores that robust swarm intelligence requires adaptive strategies.

While an abstracted 2D grid, SwarmBench's value lies in unifying LLM agency with *local* perception, communication, and emergent collective abilities. It is thus a valuable tool to explore phenomena from widespread agent deployment and help navigate the implications of future decentralized AI.

## 6 CONCLUSION

We introduced SwarmBench to assess emergent decentralized coordination in LLMs under swarm intelligence constraints. Our evaluations show that while current LLMs exhibit nascent coordination, they struggle with robust collective behavior under strict local information limits, highlighting a critical research gap in multi-agent AI.

SwarmBench provides a platform for developing LLMs with adaptive collective behavior, vital for decentralized systems. Its abstracted 2D grid (Appendix B, I), aligned with benchmarks (Chollet et al., 2025; Kamradt, 2025; Ruoss et al., 2025), is a deliberate design choice making it a foundational tool for dissecting emergent coordination in LLM agents.

Future work includes agent adaptation via RLVR pipelines (DeepSeek-AI, 2025; Xin et al., 2024; Jin et al., 2025), extension to 3D environments, and investigating communication and prompting (Appendix C). The benchmark also enables exploring biologically-inspired questions: can LLM swarms exhibit locust consensus (Sayin et al., 2025), form multi-scale structures (Khona et al., 2025), or achieve spontaneous modularity? Answering these questions will advance artificial collective intelligence.

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

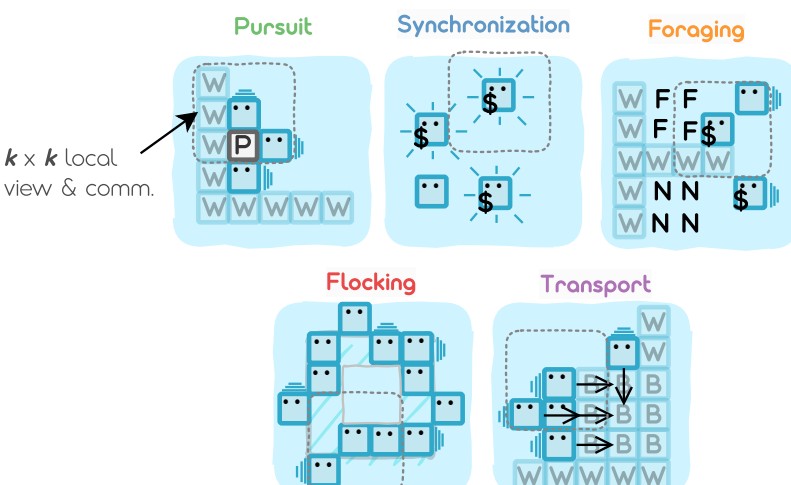

Figure S.1: **SwarmBench tasks.** The diagram shows SwarmBench's task design, including Pursuit, Synchronization, Foraging, Flocking, and Transport. Each agent is limited to observing a $k \times k$ local view.

# A    SWARMBENCH SYSTEM AND PROTOCOL DETAILS

This appendix provides a detailed description of the SwarmBench environment, agent capabilities, evaluation protocol, and task-specific scoring mechanisms used in our experiments, complementing Section 3 of the main text.

## A.1    ENVIRONMENT DETAILS

SwarmBench utilizes a simulation environment based on a 2D grid world where multiple agents ($N$ agents), controlled by LLMs, operate and interact. The adoption of a 2D grid world, while an abstraction, is a deliberate design choice aligned with foundational AI benchmarking practices (e.g., ARC-AGI tests (Chollet et al., 2025), SnakeBench (Kamradt, 2025), and LMAct (Ruoss et al., 2025)). This facilitates a focused investigation of core coordination dynamics while maintaining tractable complexity for initial explorations. This environment itself is designed as a customizable and scalable physical system, where mechanical interactions such as forces and multi-object dynamics (further detailed in Appendix B) are explicitly modeled.

The simulation proceeds in discrete time steps (rounds, $t = 1, \ldots, T$). In each round, all agents perceive their local environment (including messages from the previous round) simultaneously and decide upon their next action and potential message based on the state at the beginning of the round. Environment updates, including agent movement and object interactions, occur only after all agents have committed to their actions for that round. Interactions between agents and objects, particularly pushing and collision resolution, are governed by this discrete physics simulation that handles complex multi-object dynamics, ensuring that the mechanical properties of the system are consistently applied.

The benchmark includes several core multi-agent coordination tasks designed to probe different facets of emergent swarm behavior. These tasks are visualized in Fig. 1 in the main text, with a consolidated overview provided in Fig. S.1 (this appendix), and are further detailed with examples in Appendix D and Supplementary Videos:

- **Pursuit:** Agents (e.g., '0'–'9') must collaboratively track and corner a faster-moving prey ('P'). Tests coordination for containment, potentially aided by communication.

- **Synchronization:** Agents aim to synchronize an internal binary state ('Number' vs. '$Number') across the swarm and collectively alternate this state via a SWITCH action. Assesses consensus formation leveraging local cues and communication.

- **Foraging:** Agents navigate an environment with walls ('W') to find a food source ('F'), transport it to a nest ('N'), changing appearance ('Number' to '$Number') when carrying. Evaluates exploration, pathfinding, and potential communication-driven task allocation.

- **Flocking:** Agents must move as a cohesive group, maintaining alignment and separation while potentially navigating towards a target or avoiding obstacles. Tests emergent formation control and coordinated movement.

- **Transport:** Multiple agents must cooperate to push a large object ('B') towards a designated goal area. Tests coordinated force application and navigation around obstacles.

The environment framework supports additional tasks and is extensible. Interactions follow simplified physics rules detailed in Appendix B. Environment instances, including initial agent positions, object placements, and potentially other environmental features, are procedurally generated based on a random seed. To ensure robust evaluation, prevent overfitting to specific scenarios, and guarantee fair comparison across different models or trials, evaluation runs for different models utilize the same predefined set of seeds. This practice ensures that all models are benchmarked under identical initial conditions and environmental layouts for each corresponding seed, providing a fair and consistent basis for performance comparison. Furthermore, using a diverse set of seeds ensures the benchmark itself is robust, testing models across varied conditions to provide a more reliable assessment of their general coordination abilities rather than performance on a single, potentially idiosyncratic, scenario.

## A.2 AGENT PERCEPTION, ACTION, AND COMMUNICATION DETAILS

Consistent with the goal of studying emergent behavior from local information, agents operate with significantly restricted perception. The primary input is an egocentric $k \times k$ grid view (e.g., $5 \times 5$ in our main experiments) centered on the agent at position $\mathbf{x}_{i,t} \in \mathbb{R}^2$. This view displays local entities using symbols: the agent itself ('Y'), other agents (by ID, e.g., '1'/'$1'), walls ('W'), obstacles ('B'), empty space ('.'), off-map markers ('*'), and task-specific objects ('P', 'N', 'F'). The view includes global coordinate labels.

The full observation provided to the LLM includes:

- The local $k \times k$ grid view.

- The agent's global coordinates $\mathbf{x}_{i,t}$.

- Task-specific status (e.g., carrying_food).

- Messages received from other agents in the previous round ($t - 1$). Messages are received only from agents within the sender's local perception range at time $t - 1$.

- The task description and current progress indicators (e.g., score).

- A limited history of the agent's own recent observations and actions (e.g., last memory=5 rounds).

The detailed structure and content of the prompt given to the LLM are provided in Appendix C.

Based on this observation, the agent's LLM must decide on two outputs for round $t$:

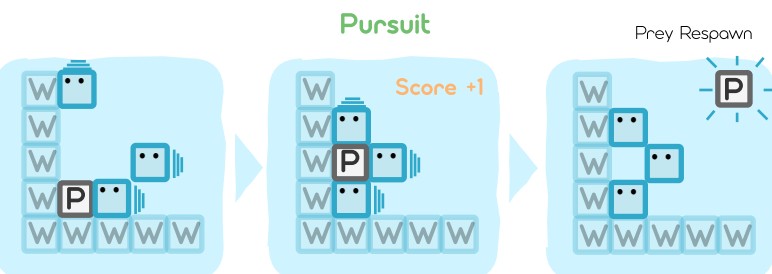

Figure S.2: **Pursuit tasks illustration.** This figure shows agents surrounding a prey (`'P'`). Upon successful capture (middle panel), the prey respawns (right panel).

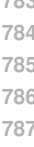
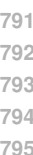

1. A primary action $A_{i,t}$ chosen from a set $\mathcal{A}$ typically including basic movements (UP, DOWN, LEFT, RIGHT, STAY). Movement actions correspond to an agent attempting to apply a directed force (default $F = 2$). Agents and objects possess inherent weight (referred to as mass in the simulation, default agent $m = 1$ calculated from a $1 \times 1$ size). Movement or pushing only occurs if the net applied force overcomes the resistance (mass) of the target object(s), considering potentially complex chain reactions resolved by the physics engine (see Appendix B). Task-specific actions (e.g., SWITCH, PICKUP, DROP) are also included.

2. A message $M_{i,t}$ (a string, potentially empty) intended for local broadcast via the MSG action.

The message $M_{i,t}$ (if non-empty) is broadcast locally and anonymously to agents within the sender's local perception range, becoming part of their observation in the next round ($t + 1$). Messages are subject to a character limit (e.g., 120 characters). This setup compels reliance on interpreting local visual cues and utilizing the constrained communication channel for effective coordination.

## A.3 EVALUATION PROTOCOL DETAILS

We define a standardized protocol focusing on zero-shot LLM evaluation. Each agent $i$ is controlled by an independent LLM instance. In round $t$, the agent receives its full observation (including received messages from $t - 1$), formulates a prompt containing this information (see Appendix C), and queries the LLM. Persistence is managed via the prompt's explicit inclusion of observation history and received messages.

The LLM response is parsed to extract the intended primary action $A_{i,t} \in \mathcal{A}$ and the message content $M_{i,t}$. An episode ends upon task success criteria being met or reaching a maximum round limit (max_round).

Our experiments (Section 4) utilize several contemporary closed-source and open-source LLMs, evaluated without task-specific fine-tuning to assess their inherent zero-shot coordination potential derived from pre-training.

## A.4 TASK-SPECIFIC SCORING MECHANISMS

This subsection details the specific scoring rules for each of the five core tasks in SwarmBench, which are broadly depicted in Fig. S.1. To further clarify the scoring process for several of these tasks, Figures S.2, S.3, S.4, S.5, and S.6 provide specific illustrations for the Pursuit, Synchronization, Foraging, Flocking, and Transport tasks, respectively. These rules are implemented within the simulation environment to quantify agent performance based on their success in achieving the defined task objectives. The scores reported in Section 4 are derived from these mechanisms.

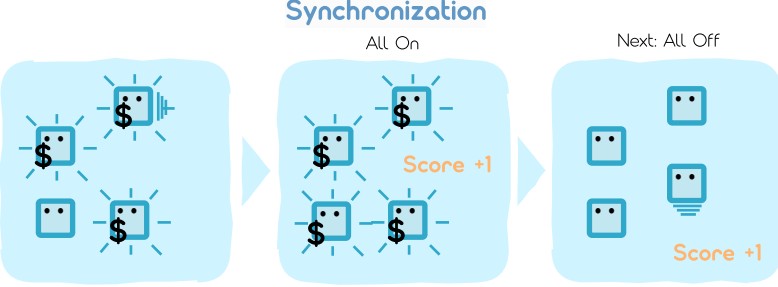

Figure S.3: **Synchronization tasks illustration.** Agents toggle an internal state (indicated by '$' symbol or its absence). Scoring occurs when all agents are in the same state (e.g., all '$', middle panel) and this state alternates from the previously scored unanimous state.

### A.4.1 PURSUIT SCORING

The `Pursuit` task, depicted in Fig. S.2, involves agents cooperatively cornering a faster-moving prey ('P'). The illustration clearly shows the stages: agents maneuvering to surround the prey (left), the prey being successfully cornered leading to a score increment (center), and the prey subsequently respawning at a new location (right), allowing the task to continue.

- **Scoring Event (Prey Caught):** The prey is considered caught if all four of its adjacent cells (up, down, left, right) are occupied by other agents or walls ('W'). This condition is visually represented in the middle panel of Fig. S.2.

- **Score Awarded:** When the prey is caught, the task score is incremented by 1.

$$\text{score} \leftarrow \text{score} + 1 \tag{1}$$

- **Prey Respawn:** After being caught, the prey is removed and respawned at a new, empty location on the map. This location $(x_s, y_s)$ is selected from several random candidate empty cells by choosing the one that minimizes a *threat heuristic*, $H$. For any cell $(x, y)$, this heuristic is calculated based on an $8 \times 8$ subview $V_{8 \times 8}(x, y)$ centered around it:

$$H(x,y) = N_A(V_{8 \times 8}(x,y)) + w_W \cdot N_W(V_{8 \times 8}(x,y)) \tag{2}$$

where $N_A(V)$ is the number of agents within subview $V$, $N_W(V)$ is the number of wall cells within $V$, and $w_W$ is a weight for walls (set to $0.9$ in our implementation). The prey respawns at the candidate location with the minimum $H$ value. This process is illustrated in the right panel of Fig. S.2.

- **Prey Movement:** If not caught, the prey attempts to move two steps in each round. It considers all valid two-step sequences. A sequence is valid if both intermediate and final cells are empty. From these valid sequences, it selects the one whose destination cell $(x_d, y_d)$ minimizes the threat heuristic $H(x_d, y_d)$ as defined in Eq. 2 (calculated for the $8 \times 8$ subview around the destination $(x_d, y_d)$). The prey aims to move to safer locations by avoiding high densities of agents and walls.

- **Total Score:** The cumulative number of times the prey has been successfully caught.

The scoring directly rewards the primary objective: successfully surrounding and immobilizing the prey, with repeated opportunities as the prey respawns.

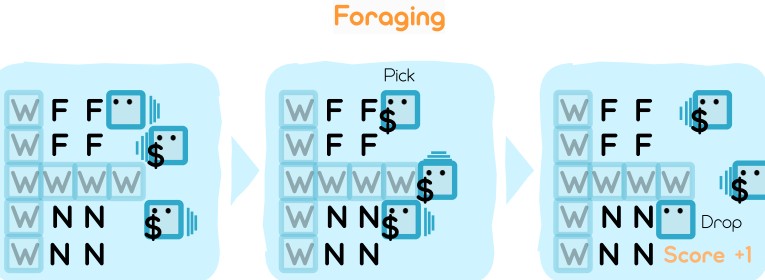

Figure S.4: **Foraging tasks illustration.** Agents pick up food ('F', indicated by '$' on the agent in the middle panel) and transport it to a nest ('N'). A score is awarded upon dropping food at the nest (right panel).

### A.4.2 SYNCHRONIZATION SCORING

In the `Synchronization` task, illustrated in Fig. S.3, agents must synchronize an internal binary state (e.g., light on/off, represented by agent symbols 'A'/'a' or, as in the figure, by the presence or absence of a '$' sign on the agent and status like '$Number') and collectively alternate this synchronized state. The figure demonstrates agents transitioning to a unanimous "All On" state (middle panel), followed by a subsequent target of "All Off" to score again. Agents can use a SWITCH action to toggle their own state.

- **Agent States:** Each agent $i$ has an internal boolean state, tracked by `agent_state[i]`, representing whether its light is on or off.
- **Scoring Condition:** A point is scored if the following two conditions are met:
  1. **Unanimity:** All agents currently have their lights in the same state (i.e., all lights are on, or all lights are off). This collective state is referred to as `state` in the implementation.
  2. **Alternation:** This newly achieved unanimous `state` (e.g., all on) is different from the unanimous state (`self.prev_state`) for which a point was last awarded (e.g., if the last score was for all off). This ensures the group must alternate between collective states to continue scoring.
- **Score Awarded:** If both unanimity and alternation conditions are satisfied, the task score is incremented by 1. The `self.prev_state` variable is then updated to record the current unanimous `state` for future alternation checks.

$$\text{score} \leftarrow \text{score} + 1 \tag{3}$$

- **Total Score:** The cumulative number of successful, alternating synchronizations.

This scoring mechanism incentivizes not just achieving a common state, but also the ability to collectively switch to the opposite common state, testing robust group consensus and coordination over time.

### A.4.3 FORAGING SCORING

The `Foraging` task, shown in Fig. S.4, requires agents to navigate an environment containing walls ('W'), pick up food ('F') from a source, and deliver it to a nest ('N'). The illustration highlights agents changing appearance (e.g., acquiring a '$' symbol, as in the "Pick" stage, middle panel) when carrying food and scoring upon successful delivery ("Drop" stage, right panel) to the nest. Agents carrying food are visually distinct (e.g., agent symbol '$Number') from those not carrying food (e.g., agent symbol 'Number').

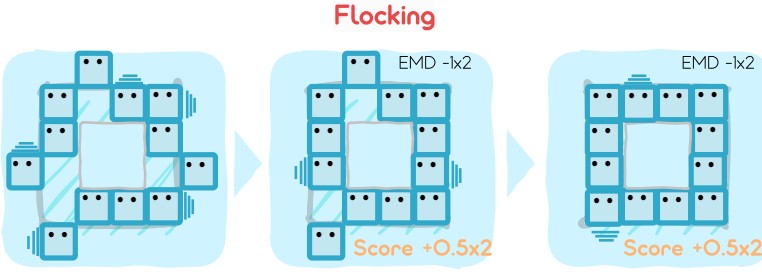

Figure S.5: **Flocking tasks illustration.** This figure depicts the Flocking task where agents aim to arrange themselves into a predefined target shape, here a hollow square. The panels show the progression from an initial agent configuration towards achieving the target.

- **Picking Up Food:** If an agent is adjacent to the food source (`'F'`) and not currently carrying food, its internal state, tracked by `food_state[name]`, is updated to indicate it is now carrying food. Its visual representation also changes accordingly, as shown in the transition from the left to the middle panel of Fig. S.4.

- **Dropping Food (Scoring Event):** If an agent is adjacent to the nest (`'N'`) and its `food_state[name]` indicates it is currently carrying food, then:

  - The task score is incremented by 1 (see the right panel of Fig. S.4).

$$\text{score} \leftarrow \text{score} + 1 \qquad (4)$$

  - The agent's `food_state[name]` is updated to indicate it is no longer carrying food (it drops the food), and its visual representation reverts.

- **Total Score:** The cumulative number of food items successfully delivered to the nest by all agents.

The score directly reflects the collective efficiency in the foraging cycle: finding food, transporting it, and returning it to the nest.

### A.4.4 FLOCKING SCORING

In the `Flocking` task (visualized in the Flocking panel of Fig. S.1), agents aim to arrange themselves to match a predefined target shape (e.g., a hollow square made of agents). Performance evaluation in this task leverages a metric inspired by the Earth Mover's Distance (EMD) (i.e., Wasserstein metric).

- **Core Metric (Translation-Invariant Assignment Cost):** The Earth Mover's Distance (EMD) generally measures the minimum work required to transform one probability distribution into another. In this task, we compute a specific variant to assess the dissimilarity between the current spatial configuration of agents and the target shape. The computation proceeds as follows:

  1. **Coordinate Extraction:** The coordinates of agents (`src`) and the coordinates defining the target shape (`tgt`) are extracted.

  2. **Candidate Translations:** A set of candidate global translation vectors $(\Delta x, \Delta y)$ is generated. These candidates are derived from all pairwise coordinate differences between individual agents in `src` and points in `tgt`.

  3. **Optimal Assignment under Translation:** For each candidate global translation $(\Delta x, \Delta y)$:

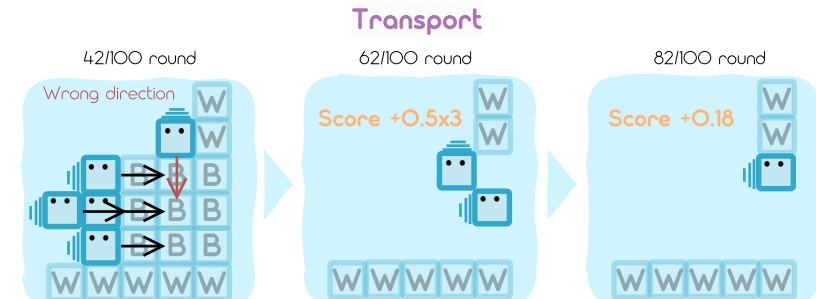

Figure S.6: **Transport tasks illustration.** This figure shows agents collaborating to push a block (`'B'`) and then escaping. Scores are awarded based on the speed of escape, as exemplified by the progression from 42/100 to 82/100 rounds.

  - A cost matrix is constructed. Each entry $C_{ij}$ in this matrix represents the Manhattan distance required to match agent $i \in$ src to target point $j \in$ tgt, assuming the entire src configuration is first translated by $(-\Delta x, -\Delta y)$ (or equivalently, tgt is translated by $(\Delta x, \Delta y)$). The cost is $\frac{1}{2}\left|(x_{\text{src}_i} - x_{\text{tgt}_j}) - \Delta x\right| + \frac{1}{2}\left|(y_{\text{src}_i} - y_{\text{tgt}_j}) - \Delta y\right|$.
  - An optimal assignment algorithm (such as the Hungarian method or similar techniques for solving the assignment problem) is then applied to this cost matrix. This finds a pairing between agents and target points that minimizes the total sum of Manhattan distances for the current global translation.

4. **Minimum Cost Selection:** The lowest sum of assignment costs found across all candidate global translations is selected as the final dissimilarity score. This score, referred to as cur_dis in the implementation, effectively represents the minimum "work" (i.e., sum of Manhattan distances) to make the agent formation match the target shape, after finding the optimal global alignment (translation).

• **Initial Distance:** Upon task reset, an initial dissimilarity score (init_dis) is calculated using the same method, based on the random initial placement of agents. This serves as a baseline.

• **Scoring Update:** At each simulation step, the current dissimilarity score (cur_dis) is recalculated. The task score reflects the cumulative reduction in this dissimilarity from the initial state, ensuring the score is non-decreasing. The overall task score is then updated to be the maximum progress achieved so far:

$$\text{score} \leftarrow \max(\text{score}, \text{init\_dis} - \text{cur\_dis}) \tag{5}$$

• **Task Completion:** The task is considered successfully completed if the cur_dis reaches 0, indicating that the agents have perfectly matched the target shape under some translation.

This scoring mechanism incentivizes agents to collectively maneuver towards and achieve the target configuration. By finding the optimal translational alignment before computing the assignment cost, the metric robustly measures shape conformance irrespective of the absolute global position of the formation.

### A.4.5   TRANSPORT SCORING

The Transport task requires agents to collaboratively push a large obstacle (`'B'`, with $m = 5$) out of an exit in the surrounding walls and then escape themselves. Fig S.6 depicts various stages of this task,

highlighting how agents must first coordinate to move the heavy block (e.g., note the arrows indicating intended push direction) and subsequently earn points by escaping the map, with scores reflecting the remaining time.

- **Scoring Event:** An agent $i$ with coordinates $(y_i, x_i)$ successfully escapes if its position is outside the defined map boundaries. Let $H$ and $W$ be the map's height and width. The escape condition is met if:

$$(y_i < 0) \lor (y_i \geq H) \lor (x_i < 0) \lor (x_i \geq W). \tag{6}$$

- **Score Awarded:** For each agent that escapes, a score is awarded. This score is proportional to the remaining time in the simulation:

$$\texttt{score} \leftarrow \texttt{score} + \frac{(\texttt{max\_round} - \texttt{current\_round})}{\texttt{max\_round}} \tag{7}$$

  This encourages agents to complete the task (pushing the obstacle and escaping) as quickly as possible.

- **Total Score:** The cumulative sum of scores from all escaped agents.

- **Task Completion:** The task is considered done when all agents have escaped the map.

The primary challenge involves the coordinated push of the heavy obstacle, as individual agents cannot move it. The scoring incentivizes both successful obstacle removal and efficient individual escape.

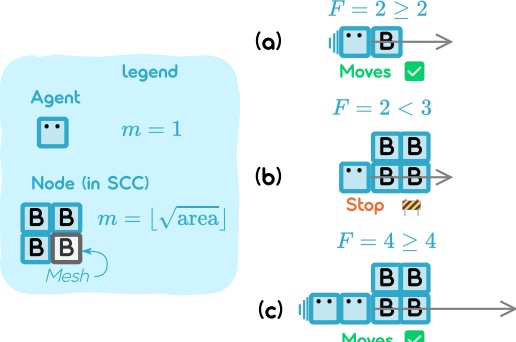

Figure S.7: **Illustration of the physics engine's movement resolution.** The legend defines: an `Agent` ($m = 1$); a `Mesh` ('B' block, area $= 1$, $m = 1$); and a `Node` (in `SCC`) as a rigid aggregate of `Meshes` whose intrinsic mass is $m = \lfloor \sqrt{\text{area}} \rfloor$ (e.g., the $2 \times 2$ `Node` in the legend has area 4, $m = 2$). Movement requires the net applied force $F$ to be $\geq$ the total mass $M_{\text{system}}$ of the rigidly connected components attempting to move together. **a.** External force $F = 2$ acts on one `Agent` and one `Mesh` B. These move as a unit. $M_{\text{system}} = m_{\text{Agent}} + m_{\text{Mesh}} = 1 + 1 = 2$. Since $F = 2 \geq M_{\text{system}} = 2$, the system moves. **b.** An `Agent` (propulsive force $F = 2$) pushes a $2 \times 2$ block of `Meshes`. The block acts as a rigid object with area 4, thus $m_{\text{block}} = \lfloor \sqrt{4} \rfloor = 2$. The system attempting to move includes the `Agent` itself. $M_{\text{system}} = m_{\text{Agent}} + m_{\text{block}} = 1 + 2 = 3$. Since $F = 2 < M_{\text{system}} = 3$, the system stops. **c.** External force $F = 4$ acts on two `Agents` and the $2 \times 2$ block ($m_{\text{block}} = 2$). The entire group acts as the movable system. $M_{\text{system}} = 2 \times m_{\text{Agent}} + m_{\text{block}} = (1 + 1) + 2 = 4$. Since $F = 4 \geq M_{\text{system}} = 4$, the system moves, demonstrating cooperative transport.

## B  PHYSICS SIMULATION DETAILS

The SwarmBench simulation employs a discrete physics engine to govern interactions between agents and objects within the 2D grid world. This engine resolves resolve complex multi-object pushing scenarios, ensuring that collective actions, e.g., in the `Transport` task, are subject to consistent and non-trivial physical laws, and can be effectively and conveniently extended to simulate more complex custom tasks beyond the current five core scenarios.

### B.1  CORE PHYSICAL ENTITIES AND PROPERTIES

Two primary constructs define physical entities:

- **Mesh**: Represents a discrete physical object on the grid. Each `Mesh` has:
  - `pos`: Its global top-left coordinate $(i, j)$.
  - `shape`: A 2D array defining its footprint, which represents the occupied grid cells. This system supports both regular and irregular objects. For an irregular object, the `shape` array would detail its specific cell occupancy, often defined within its overall bounding box (e.g., a $1 \times 1$ square for an agent, a $1 \times 4$ rectangle for a large obstacle, or a custom pattern of occupied cells for an L-shaped object within its rectangular bounding box). The area, used for mass calculation, is the count of these explicitly occupied cells (non-empty cells) within the `shape` array.
  - `static`: A boolean indicating if the object is immovable (e.g., walls 'W').

- mass ($m$): Resistance to motion, calculated as $m = \lfloor\sqrt{\text{area}}\rfloor$, where area is the number of non-empty cells in its shape. For a standard $1 \times 1$ agent or Mesh (like 'B'), area is 1, thus mass $m = 1$, as shown in Fig. S.7.

- **Node**: A computational representation used during physics resolution. A Node can represent a single Mesh or, crucially, an aggregate of Meshes that form a Strongly Connected Component (SCC) in the interaction graph (see below). Each Node aggregates:

  - Total effective mass ($m_v$) and static status of the components it represents.
  - Net force ($F_{\text{net},v}$) acting on it. This force can originate from agents external to this Node (see Fig. S.7**a**, **c**), or be the propulsive force generated by agents within this Node that are attempting to move it (see Fig. S.7**b**).

Agents are a specific type of Mesh with mass $m = 1$. When an agent performs a movement action (e.g., UP, RIGHT), it attempts to apply a directed propulsive force, typically of magnitude $F_p = 2$, to an adjacent entity or into empty space. If this agent is the primary source of motive force for a rigidly connected system (an SCC) of which it is a part, its propulsive force $F_p$ acts as the $F_{\text{net},v}$ for that SCC, and the SCC's total effective mass $m_v$ includes the agent's own mass.

## B.2 INTERACTION RESOLUTION VIA SCCs AND ILP

The simulation resolves all potential movements and pushes within a single time step through a multi-stage process:

1. **Contact Graph Construction**: The engine identifies all Mesh objects that are adjacent and could potentially exert force on one another based on intended agent actions or ongoing pushes. This forms a directed graph where an edge $v \to u$ indicates that Mesh $v$ could potentially push Mesh $u$.

2. **Strongly Connected Component (SCC) Reduction**: Tarjan's algorithm (Lengauer & Tarjan, 1979) is applied to the contact graph to identify all SCCs. An SCC represents a group of Meshes that are mutually pushing each other or form a rigid cluster that must move as one unit (or not at all). Each SCC is collapsed into a single aggregate Node. Meshes not part of any cycle become individual Nodes. This process transforms the potentially cyclic contact graph into a Directed Acyclic Graph (DAG) of Nodes, representing the pathways of force transmission. The properties of an aggregate Node (like its total effective mass $m_v$ for movement checks and the net force $F_{\text{net},v}$) are determined by its constituent Meshes and any internal propulsive forces.

3. **Integer Linear Program (ILP) Formulation and Solution**: The core of the physics resolution is an ILP problem formulated and solved using the PuLP (Mitchell et al., 2011) library.

   - **Variables**: Binary variables $x_v$ indicate if Node $v$ moves; continuous variables represent net forces on nodes and forces transmitted between connected nodes in the DAG.
   - **Objective**: Maximize $\sum x_v$ — i.e., maximize the number of (aggregate) Nodes that are successfully moved.
   - **Key Constraints**:
     - *Movement Condition*: A Node $v$ can only move ($x_v = 1$) if the total net force $F_{\text{net},v}$ acting on it in the direction of potential movement is greater than or equal to its total effective mass $m_v$ (i.e., $F_{\text{net},v} \geq m_v$). Here, $m_v$ is the sum of the intrinsic masses of all components forming the rigid Node $v$, where the intrinsic mass of any sub-aggregate is $m = \lfloor\sqrt{\text{area}}\rfloor$ and individual components (Agent or Mesh) have $m = 1$. If $F_{\text{net},v}$ comes from an agent within the Node, that agent's mass is included in $m_v$. This condition is illustrated in Fig. S.7.

- *Force Transmission*: Force is transmitted along the DAG. A `Node` $v$ can only exert force on its children in the DAG if it itself moves ($x_v = 1$) and has sufficient "leftover" force ($F_{\text{net},v} - m_v$).
- *Static Objects*: `Nodes` marked as `static` (e.g., containing walls) are constrained such that $x_v = 0$.
- *Grid Boundaries*: Movement is implicitly constrained by grid boundaries and collisions with other static objects, handled by the graph construction.

The ILP solver finds the optimal set of `Nodes` that can move simultaneously while satisfying all physical constraints.

4. **Position Update**: The global positions of the `Meshes` belonging to the `Nodes` determined to be movable by the ILP solution are updated on the simulation grid.

This physics model, particularly the SCC reduction and ILP-based resolution, allows SwarmBench to simulate complex, emergent physical interactions that require genuine coordination, such as multiple agents cooperatively pushing a heavy object that no single agent could move alone.

## C   PROMPT DESIGN

The following tcolorbox shows the exact structure and content of the prompt string generated by the `gen_prompt` function and provided to each LLM agent in SwarmBench at each decision step. Placeholders within curly braces (e.g., {name}, {task_desc}, {view_str}) are dynamically filled with actual simulation data during runtime.

---

**SwarmBench Agent Prompt Template**

```
"""You are Agent {name}, operating in a multi-agent environment.  Your goal
is to complete the task through exploration and collaboration.

Task description:
{task_desc}

Round:  {round_num}

Your recent {self.memory}-step vision (not the entire map):
{view_str}

Your current observation:
{level_obs_str}

Message you received:
{messages_str}

Your action history:
{history_str}

Symbol legend:
- Number:  An agent whose id is this number (do not mistake column no.  and
line no.  as agent id).
- Y: Yourself.  Others see you as your id instead of "Y".
- W: Wall.
- B: Pushable obstacle (requires at least 5 agents pushing in the same
direction).
- .:  Empty space (you can move to this area).
- *:  Area outside the map.
And other symbols given in task description (if any).

Available actions:
1.  UP: Move up
2.  DOWN: Move down
3.  LEFT: Move left
4.  RIGHT: Move right
5.  STAY: Stay in place
6.  MSG: Send a message
And other actions given in task description (if any).

Physics rules:
1.  Your own weight is 1, and you can exert a force of up to 2.
2.  An object (including yourself) can only be pushed if the total force in
one direction is greater than or equal to its weight.
```

---

```
3.  Static objects like W (walls) cannot be pushed; only B can be pushed.
4.  Force can be transmitted, but only between directly adjacent objects.
That means, if an agent is applying force in a direction, you can push that
agent from behind to help.
5.  Only pushing is allowed - there is no pulling or lateral dragging.  In
other words, to push an object to the right, you must be on its left side
and take the RIGHT action to apply force.

Message rules:
1.  A message is a string including things you want to tell other agents.
2.  Your message can be received by all agents within your view, and you
can receive messages from all agents within your view.
3.  Messages are broadcast-based.  The source of a message is anonymous.
4.  Write only what's necessary in your message.  Avoid any ambiguity in
your message.
5.  Messages is capped to no more than 120 characters, exceeding part will
be replaced by "...".

Other rules:
1.  Coordinates are represented as (i, j), where i is the row index and j
is the column index.  Your 5x5 vision uses global coordinates, so please
use global coordinates.
2.  The direction of increasing i is downward, and increasing j is to the
right.
3.  Objects that are completely outside the map (marked with "*") will be
removed.

Please think carefully and choose your next action.  You will need to
collaborate with other agents to successfully complete the task.

Your response should include:
1.  Analysis of the current situation
2.  Your decision and reasoning
3.  The message to be left (if any)

End your response clearly with your chosen action:  "ACTION: [YOUR_ACTION]"
and/or "MSG: [Your message (no line breaks).]"
"""
```

**Example SwarmBench Prompt (Agent_5, Round 62, Transport Task)**

```
You are Agent Agent_5, operating in a multi-agent environment.  Your goal
is to complete the task through exploration and collaboration.

Task description:
The boundary of the map is surrounded by walls (denoted as W), with a gap
leading to the outside of the map (denoted as '*').  The gap is blocked by
an obstacle (denoted as B).
The goal is to first locate the obstacle (B), then have five robots
simultaneously push it through the exit, and finally escape to the outside
of the map (denoted as '*').
```

```
Round:  62

Your recent 5-step vision (not the entire map):
Current Step:
     1   2   3   4   5
 3   1   B   B   B   B
 4   9   8   0   11  10
 5   .   .   Y   .   7
 6   .   .   .   .   .
 7   .   .   .   .   .

1 Steps Before:
     1   2   3   4   5
 3   1   B   B   B   B
 4   .   9   8   0   11
 5   .   .   Y   .   7
 6   .   .   .   .   .
 7   .   .   .   .   .

2 Steps Before:
     1   2   3   4   5
 3   1   B   B   B   B
 4   .   9   8   0   11
 5   .   .   Y   .   7
 6   .   .   .   .   .
 7   .   .   .   .   .

3 Steps Before:
     1   2   3   4   5
 3   1   B   B   B   B
 4   .   9   8   0   11
 5   .   .   Y   .   7
 6   .   .   .   .   .
 7   .   .   .   .   .

4 Steps Before:
     1   2   3   4   5
 3   1   B   B   B   B
 4   .   9   8   0   11
 5   .   .   Y   .   7
 6   .   .   .   .   .
 7   .   .   .   .   .

Your current observation:

Message you received:
Message:  "At (5,3), moving UP to (4,3) for LEFT push on B at (3,3).  Ready
for 5-force."
Message:  "Moving to (4,5) to join LEFT push on B at (3,5).  Ready for
5-force push."
Message:  "Moving RIGHT to (4,3) then (4,4) for LEFT push on B. Ready to
```

```
  join force."
  Message: "At (4,5) pushing LEFT on B at (3,5).  5-force confirmed.
  Executing now."

  Your action history:
  Round 58:  Action:  UP, Message:  "Moving to (4,3) then (3,4) for LEFT push
  on B. Need 4 more at (3,4)/(4,4)."
  Round 59:  Action:  UP, Message:  "Moving to (4,3) then (3,4) for LEFT push
  on B. Need 4 more at (3,4)/(4,4)."
  Round 60:  Action:  UP, Message:  ""Moving to (3,3) for LEFT push on B.
  Need 1 more at (3,3)/(4,3) to complete 5-force.""
  Round 61:  Action:  UP, Message:  ""Moving to (4,3) for LEFT push on B at
  (3,3).  Ready to join 5-force push.""
  Round 62:  Action:  UP, Message:  "At (5,3), moving UP to (4,3) for LEFT
  push on B at (3,3).  Ready for 5-force."

  Symbol legend:
 - Number:  An agent whose id is this number (do not mistake column no.  and
  line no.  as agent id).
 - Y: Yourself.  Others see you as your id instead of "Y".
 - W: Wall.
 - B: Pushable obstacle (requires at least 5 agents pushing in the same
  direction).
 - .:  Empty space (you can move to this area).
 - *:  Area outside the map.
  And other symbols given in task description (if any).

  Available actions:
  1.  UP: Move up
  2.  DOWN: Move down
  3.  LEFT: Move left
  4.  RIGHT: Move right
  5.  STAY: Stay in place
  6.  MSG: Send a message
  And other actions given in task description (if any).

  Physics rules:
  1.  Your own weight is 1, and you can exert a force of up to 2.
  2.  An object (including yourself) can only be pushed if the total force in
  one direction is greater than or equal to its weight.
  3.  Static objects like W (walls) cannot be pushed; only B can be pushed.
  4.  Force can be transmitted, but only between directly adjacent objects.
  That means, if an agent is applying force in a direction, you can push that
  agent from behind to help.
  5.  Only pushing is allowed – there is no pulling or lateral dragging.  In
  other words, to push an object to the right, you must be on its left side
  and take the RIGHT action to apply force.

  Message rules:
  1.  A message is a string including things you want to tell other agents.
  2.  Your message can be received by all agents within your view, and you
```

```
can receive messages from all agents within your view.
3.  Messages are broadcast-based.  The source of a message is anonymous.
4.  Write only what's necessary in your message.  Avoid any ambiguity in
your message.
5.  Messages is capped to no more than 120 characters, exceeding part will
be replaced by "...".

Other rules:
1.  Coordinates are represented as (i, j), where i is the row index and j
is the column index.  Your 5x5 vision uses global coordinates, so please
use global coordinates.
2.  The direction of increasing i is downward, and increasing j is to the
right.
3.  Objects that are completely outside the map (marked with "*") will be
removed.

Please think carefully and choose your next action.  You will need to
collaborate with other agents to successfully complete the task.

Your response should include:
1.  Analysis of the current situation
2.  Your decision and reasoning
3.  The message to be left (if any)

End your response clearly with your chosen action:  "ACTION: [YOUR_ACTION]"
and/or "MSG: [Your message (no line breaks).]"
```

**Example Agent Response (JSON)**

```
{
    "response": "### Analysis of the Current Situation:\n1. **Obstacle (B)
    Location**: The obstacle B is located at positions (3,2), (3,3), (3,4),
    and (3,5). The exit is likely at one of these positions, but the exact
    exit (*) is not visible in the current vision. However, based on the
    messages, the agents are focusing on pushing B at (3,3) and (3,5) LEFT
    (towards column 1).\n2. **Agent Positions**: \n   - You (Y) are at
    (5,3). \n   - Other agents are at (4,3), (4,5), and possibly others not
    visible in the current vision. \n   - Messages indicate that agents
    are ready for a 5-force push on B at (3,3) and (3,5).\n3. **Messages**:
    \n   - Agents are coordinating to push B LEFT at (3,3) and (3,5). \n
    - A 5-force push is confirmed for B at (3,5), and agents are moving
    into position for B at (3,3).\n4. **Your Role**: You are at (5,3) and
    can move UP to (4,3) to join the push on B at (3,3). Alternatively, you
    could move to (4,5) to assist with the push on B at (3,5), but the
    messages suggest the push at (3,5) is already confirmed with enough
    agents.\n\n### Decision and Reasoning:\n- **Primary Goal**: Assist in
    the 5-force push on B at (3,3). The messages indicate that agents are
    ready at (4,3) and (4,4) for this push. You are at (5,3) and can move
    UP to (4,3) to join the push.\n- **Action**: Move UP to (4,3) to align
    yourself for the LEFT push on B at (3,3). This will contribute to the
    5-force needed to move the obstacle.\n- **Message**: Confirm your
    readiness to join the push at (3,3) to ensure coordination.\n\n###
    Message:\n\"At (5,3), moving UP to (4,3) for LEFT push on B at (3,3).
    Ready for 5-force.\"\n\n### Action:\nACTION: UP  \nMSG: \"At (5,3),
    moving UP to (4,3) for LEFT push on B at (3,3). Ready for 5-force.\"",
    "action": "UP",
    "message": "\"At (5,3), moving UP to (4,3) for LEFT push on B at (3,3).
    Ready for 5-force.\""
}
```

# D EXAMPLES

This appendix provides visual examples of the simulation environment for each of the five core SwarmBench tasks. Each figure shows a snapshot from a simulation run, illustrating agent positions, environment features, and the overall task objective.

## D.1 PURSUIT

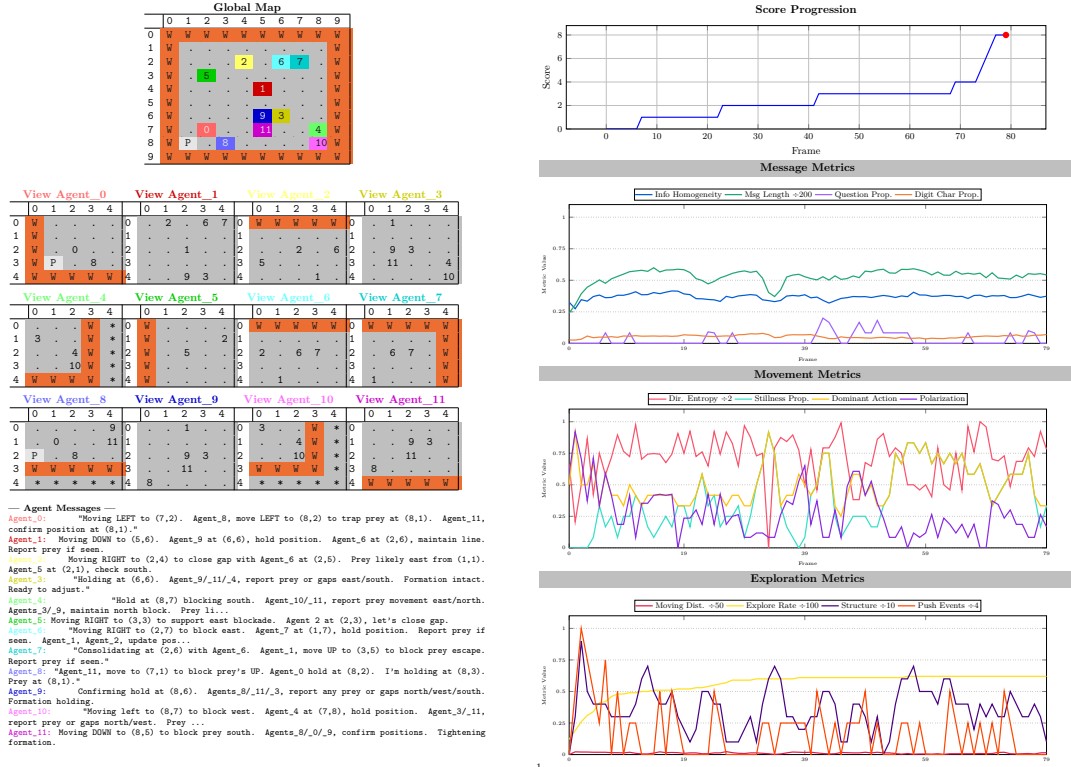

Figure S.8: Example visualization for the Pursuit task. Agents (0-11) attempt to surround the prey (P). Replay videos can be found in Supplementary Materials (see Supplementary Video 1)

## D.2 SYNCHRONIZATION

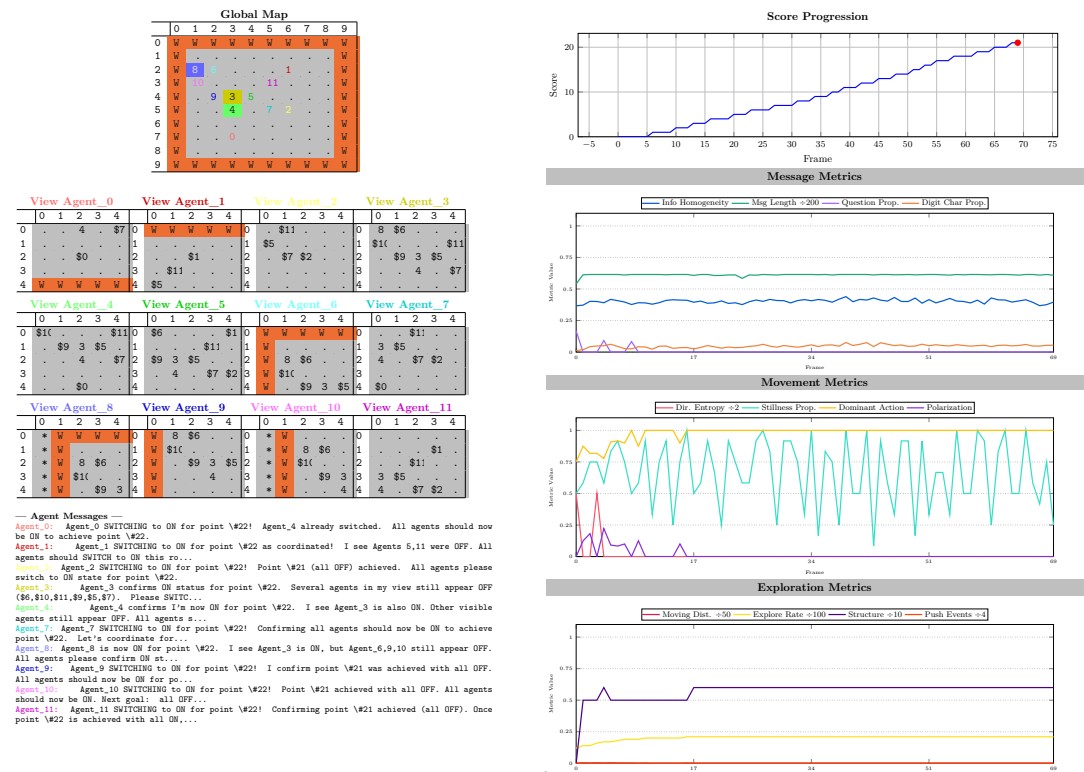

Figure S.9: Example visualization for the Synchronization task. Agents (Number/$Number) aim to reach a consensus state. Replay videos can be found in Supplementary Materials (see Supplementary Video 2)

### D.3 FORAGING

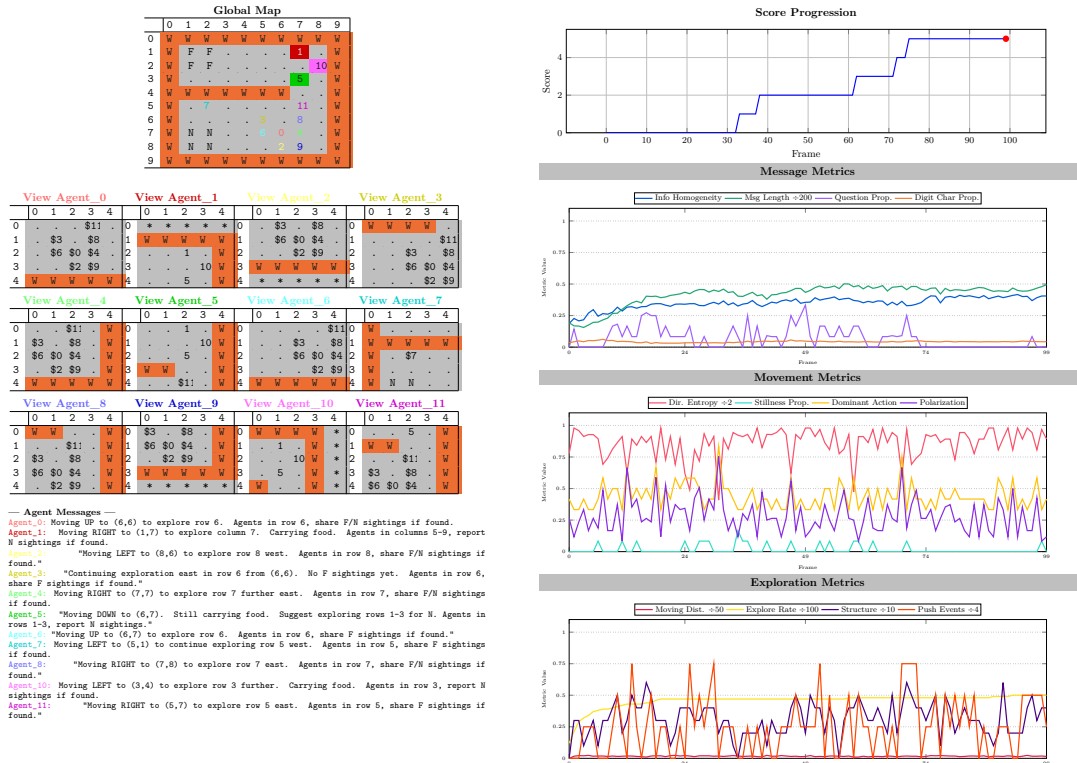

Figure S.10: Example visualization for the `Foraging` task. Agents (`Number`/`$Number`) collect food (`F`) and return it to the nest (`N`). Replay videos can be found in Supplementary Materials (see Supplementary Video 3)

### D.4 FLOCKING

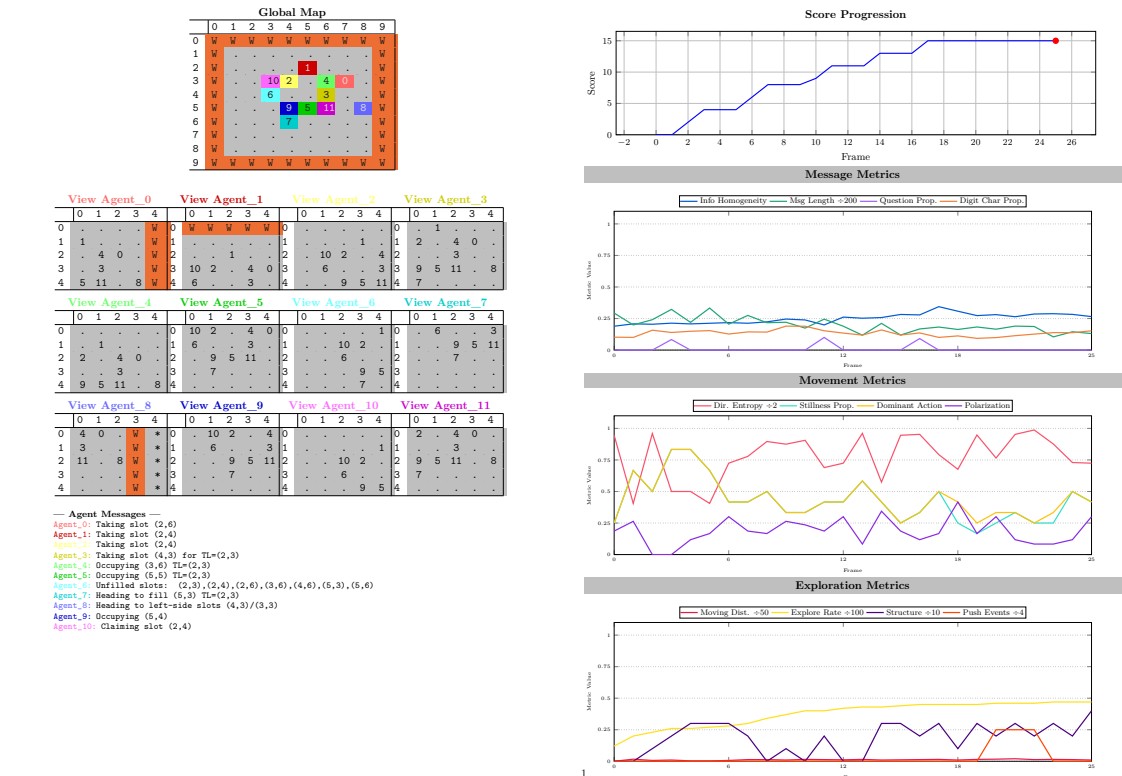

Figure S.11: Example visualization for the `Flocking` task. Agents (`0-11`) attempt to move cohesively. Replay videos can be found in Supplementary Materials (see Supplementary Video 4)

## D.5 TRANSPORT

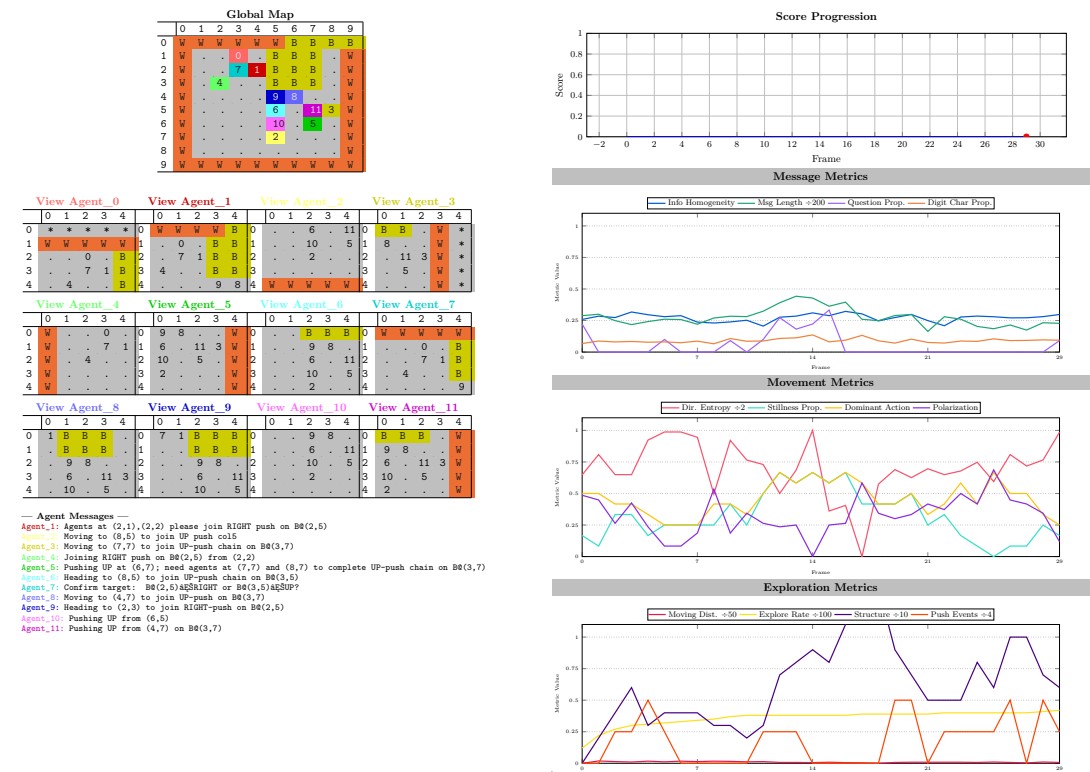

Figure S.12: Example visualization for the Transport task. Agents (0-11) coordinate to push a large obstacle (B). Replay videos can be found in Supplementary Materials (see Supplementary Video 5)

# E  DETAILED TASK PERFORMANCE DATA

Table S.1 provides the detailed numerical results corresponding to the performance overview presented in Fig. 4 in the main text (Section 4.1). It shows the mean scores and standard deviations for each evaluated LLM across the five SwarmBench tasks, averaged over 5 simulation runs. Models are ordered by their total score (sum across the five tasks) in descending order.

Table S.1: **Detailed average scores with standard deviations for various LLMs across five SwarmBench tasks.** Tasks: `Pursuit`, `Synchronization`, `Foraging`, `Flocking`, `Transport`. Scores averaged over 5 simulations. Models ordered by Total Score. This data is visualized in Fig. 4.

| Model | Pursuit | Synchroni-zation | Foraging | Flocking | Transport | Total Score |
|---|---|---|---|---|---|---|
| gemini-2.0-flash | $8.80 \pm 1.60$ | $3.40 \pm 2.94$ | $5.80 \pm 4.35$ | $9.40 \pm 0.80$ | $0.00 \pm 0.00$ | 27.40 |
| o4-mini | $9.60 \pm 0.49$ | $2.80 \pm 1.17$ | $4.80 \pm 2.64$ | $8.90 \pm 1.83$ | $0.52 \pm 1.04$ | 26.62 |
| claude-3.7-sonnet | $4.40 \pm 1.20$ | $12.60 \pm 9.62$ | $1.20 \pm 1.47$ | $7.50 \pm 1.93$ | $0.00 \pm 0.00$ | 25.70 |
| gpt-4.1 | $8.40 \pm 1.85$ | $2.80 \pm 0.75$ | $3.20 \pm 1.94$ | $5.70 \pm 0.68$ | $0.00 \pm 0.00$ | 20.10 |
| deepseek-v3 | $4.20 \pm 2.48$ | $4.00 \pm 1.41$ | $2.60 \pm 2.06$ | $6.40 \pm 1.40$ | $0.00 \pm 0.00$ | 17.20 |
| gpt-4o | $3.40 \pm 1.50$ | $1.80 \pm 1.33$ | $1.60 \pm 1.85$ | $5.00 \pm 3.18$ | $0.00 \pm 0.00$ | 11.80 |
| o3-mini | $3.60 \pm 2.06$ | $2.20 \pm 1.17$ | $2.60 \pm 3.88$ | $2.70 \pm 0.93$ | $0.00 \pm 0.00$ | 11.10 |
| qwq-32b | $2.20 \pm 1.94$ | $1.20 \pm 0.98$ | $0.80 \pm 0.75$ | $5.90 \pm 0.20$ | $0.00 \pm 0.00$ | 10.10 |
| deepseek-r1 | $1.00 \pm 0.63$ | $1.20 \pm 1.17$ | $1.00 \pm 1.10$ | $6.10 \pm 0.38$ | $0.71 \pm 1.42$ | 10.01 |
| llama-3.1-70b | $1.80 \pm 0.40$ | $1.00 \pm 1.10$ | $0.00 \pm 0.00$ | $7.10 \pm 0.74$ | $0.00 \pm 0.00$ | 9.90 |
| llama-4-scout | $1.20 \pm 0.75$ | $0.20 \pm 0.40$ | $1.00 \pm 1.55$ | $7.10 \pm 2.44$ | $0.00 \pm 0.00$ | 9.50 |
| gpt-4.1-mini | $1.40 \pm 0.80$ | $0.60 \pm 0.49$ | $1.40 \pm 1.02$ | $5.00 \pm 2.76$ | $0.00 \pm 0.00$ | 8.40 |
| claude-3.5-haiku | $0.60 \pm 0.49$ | $1.00 \pm 0.00$ | $0.00 \pm 0.00$ | $5.60 \pm 0.74$ | $0.00 \pm 0.00$ | 7.20 |

## F  DETAILED GROUP DYNAMICS METRICS

To quantitatively analyze emergent collective behaviors, we compute metrics based on agent positions $\mathbf{x}_{i,t}$, their primary actions $A_{i,t}$, and messages $M_{i,t}$. These metrics are calculated per round and then typically averaged over the duration of a simulation run for correlation with the final score. The specific variable names used in our analysis scripts correspond to these conceptual definitions.

**Communication-based Metrics**

- **Proportion of Question Sentences:** The average per-round proportion of non-empty messages that contain a question mark ('?').
- **Proportion of Digit Characters:** The average per-round proportion of all characters in non-empty messages that are digits. This may indicate sharing of numerical data like coordinates.
- **Mean Message Length:** The average per-round mean character length of non-empty messages.
- **Standard Deviation of Message Length:** The average per-round standard deviation of character lengths of non-empty messages.
- **Information Homogeneity:** The average per-round pairwise cosine similarity of embeddings of unique non-empty messages. Embeddings are generated using a Sentence-BERT model (e.g., `all-mpnet-base-v2`). This measures semantic coherence.

**Action-based Metrics**  Let $\mathcal{A}_{\text{move}} = \{\text{UP}, \text{DOWN}, \text{LEFT}, \text{RIGHT}\}$ be movement actions and $\mathcal{A}_{\text{coord}} = \mathcal{A}_{\text{move}} \cup \{\text{STAY}\}$ be coordination-relevant actions.

- **Directional Entropy:** The average per-round Shannon entropy of actions in $\mathcal{A}_{\text{move}}$ taken by agents. Measures the unpredictability or variability of movement directions.

$$H_t(\mathcal{A}_{\text{move}}) = - \sum_{a \in \mathcal{A}_{\text{move}}} p_t(a) \log_2 p_t(a) \tag{8}$$

  where $p_t(a)$ is the proportion of agents performing action $a$ in round $t$ from the set $\mathcal{A}_{\text{move}}$.
- **Stillness Proportion:** The average per-round proportion of agents executing the STAY action.
- **Dominant Action Proportion:** The average per-round proportion of agents performing the single most frequent action within the set $\mathcal{A}_{\text{coord}}$. A high value indicates strong action consensus.
- **Polarization Index:** The average per-round magnitude of the mean movement vector. Action vectors $\mathbf{v}(a)$ are assigned (e.g., $\mathbf{v}(\text{UP}) = (0, -1)$, $\mathbf{v}(\text{STAY}) = (0, 0)$).

$$P_t = \left\| \frac{1}{N_t^{\text{coord}}} \sum_{i \text{ s.t. } A_{i,t} \in \mathcal{A}_{\text{coord}}} \mathbf{v}(A_{i,t}) \right\|_2 \tag{9}$$

  where $N_t^{\text{coord}}$ is the number of agents performing an action from $\mathcal{A}_{\text{coord}}$ in round $t$. Indicates overall movement alignment.

**Position and Interaction-based Metrics**

- **Average Moving Distance:** The average per-round cumulative Manhattan distance moved by each agent from its previous position.
- **Exploration Rate:** The average per-round number of unique grid cells occupied by any agent up to that round.

- **Local Structure Preservation Count:** The average per-round count of pairs of agents that were adjacent (Manhattan distance 1) in round $t-1$ and remain adjacent in round $t$.

- **Agent Push Events:** The average per-round count of events where agent A, intending to move into agent B's adjacent cell, successfully does so, and agent B is displaced in the same direction as A's intended movement. This indicates a successful cooperative push.

## G  TASK-SPECIFIC EMERGENT DYNAMICS ANALYSIS VISUALIZATIONS

This appendix provides detailed visualizations supporting the analysis of emergent group dynamics and their correlation with task performance, as summarized in Section 4.2. For each of the five core SwarmBench tasks, we present a series of plots to illustrate these relationships. The twelve dynamic features analyzed are defined in Appendix F.

For each task, we show:

1. A heatmap of the Pearson correlation coefficients between all pairs of the twelve dynamic features and the final task score (e.g., Fig. S.13). This provides an overview of inter-feature relationships and feature-score correlations.

2. A bar plot showing the Pearson correlation coefficient of each dynamic feature specifically with the final task score. Asterisks (*, **, ***) indicate statistical significance ($p < 0.05, p < 0.01, p < 0.001$ respectively) (e.g., Fig. S.14). This highlights the individual predictive power of each feature.

3. A scatter plot illustrating the relationship between the dynamic feature with the highest absolute Pearson correlation with the score and the final task score, including a linear regression trend line (e.g., Fig. S.15). This visualizes the strength and direction of the strongest individual relationship.

4. A swarm plot of the feature importance (using SHAP (Lundberg & Lee, 2017)) when predicting the final task score using all dynamic features (e.g., Fig. S.16). This indicates the relative contribution of each feature in a multivariate context.

These visualizations offer a task-specific deep dive into how different emergent behaviors and communication patterns relate to performance, providing the detailed evidence for the trends discussed in the main text.

## G.1 PURSUIT TASK DYNAMICS

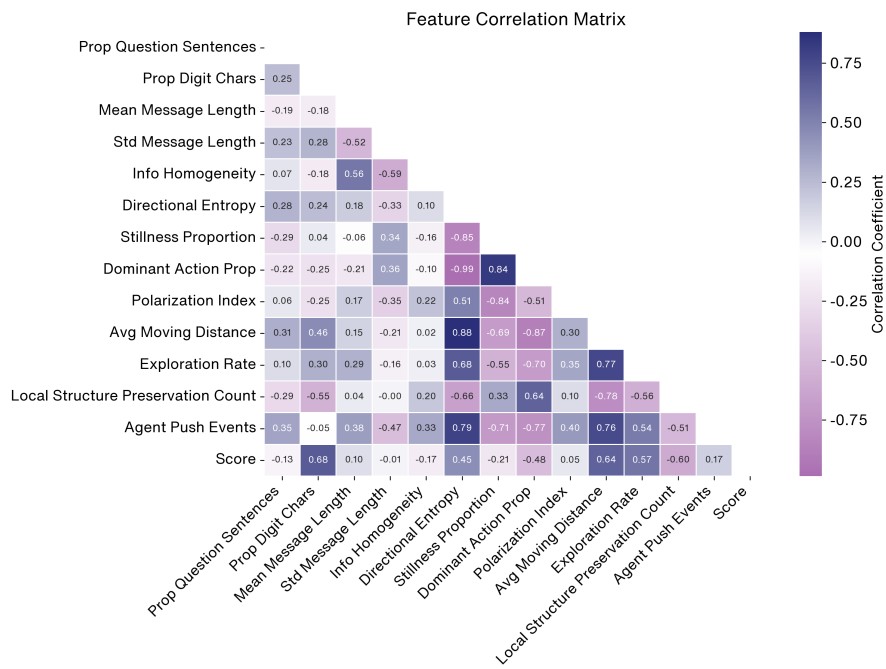

Figure S.13: Feature correlation matrix for the Pursuit task. This heatmap shows Pearson correlation coefficients between all pairs of dynamic features and the task score.

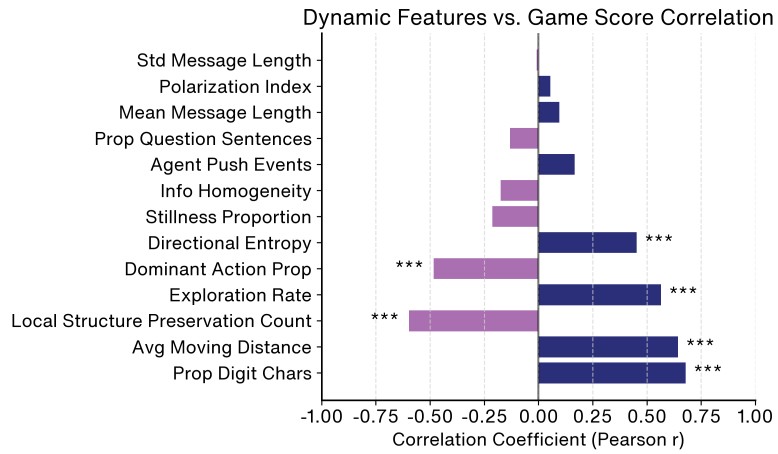

Figure S.14: Correlation of dynamic features with score for the Pursuit task. Bars represent Pearson's $r$; * indicates $p < 0.05$,** indicates $p < 0.01$,*** indicates $p < 0.001$.

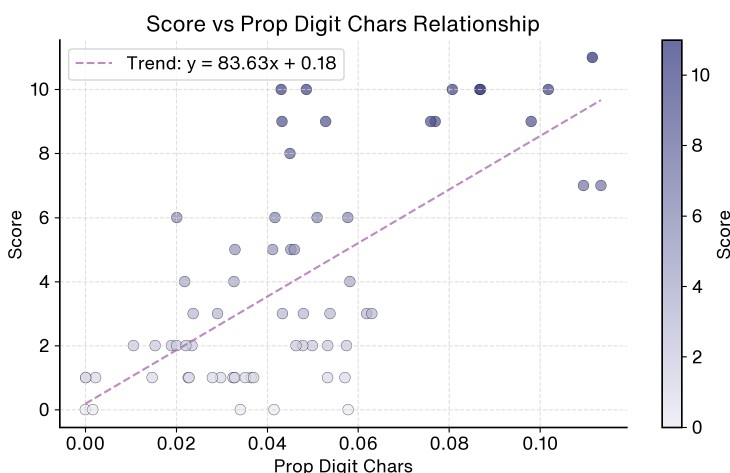

Figure S.15: Relationship between the top predictive dynamic feature (Proportion of Digit Characters in Message) and score for the Pursuit task, with linear regression trend.

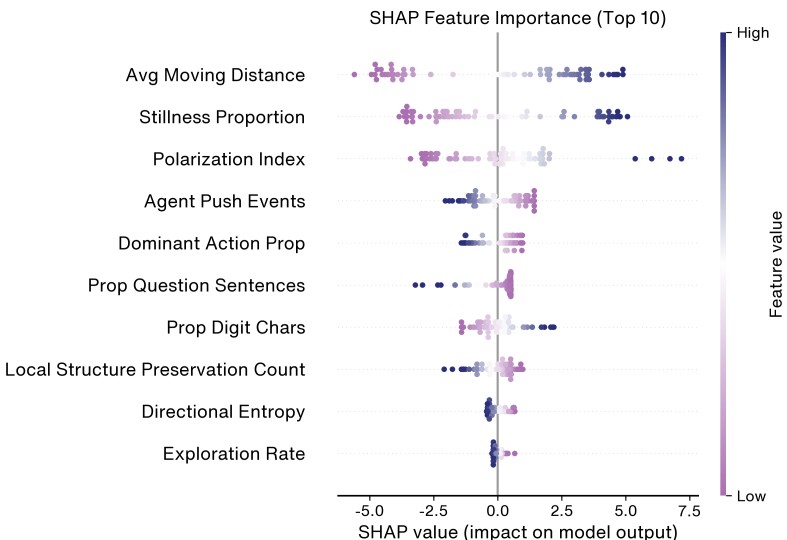

Figure S.16: Feature importance from the linear regression model for the Pursuit task.

## G.2 SYNCHRONIZATION TASK DYNAMICS

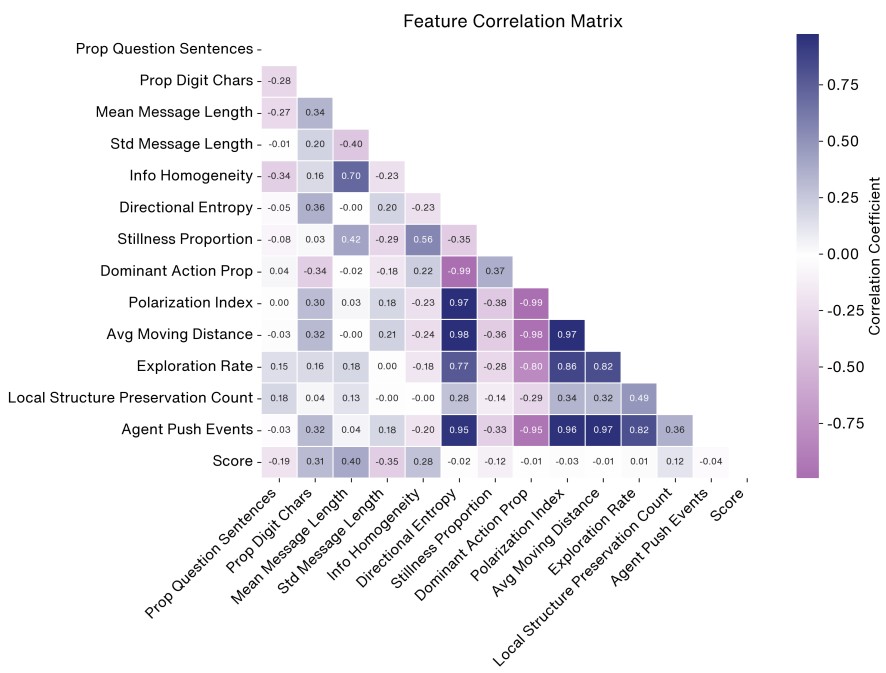

Figure S.17: Feature correlation matrix for the Synchronization task.

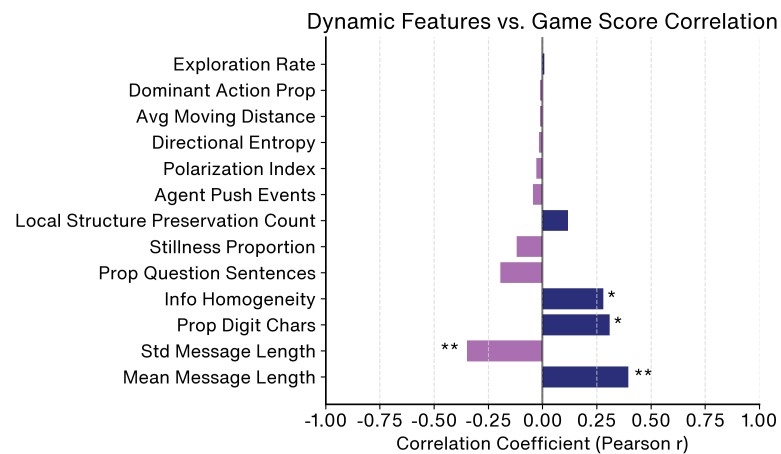

Figure S.18: Correlation of dynamic features with score for the Synchronization task. Bars represent Pearson's $r$; * indicates $p < 0.05$,** indicates $p < 0.01$,*** indicates $p < 0.001$.

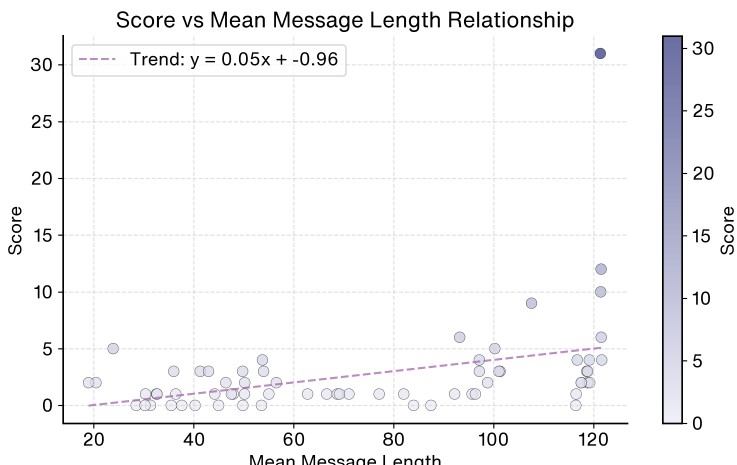

Figure S.19: Relationship between the top predictive dynamic feature (Mean Message Length) and score for the Synchronization task, with linear regression trend.

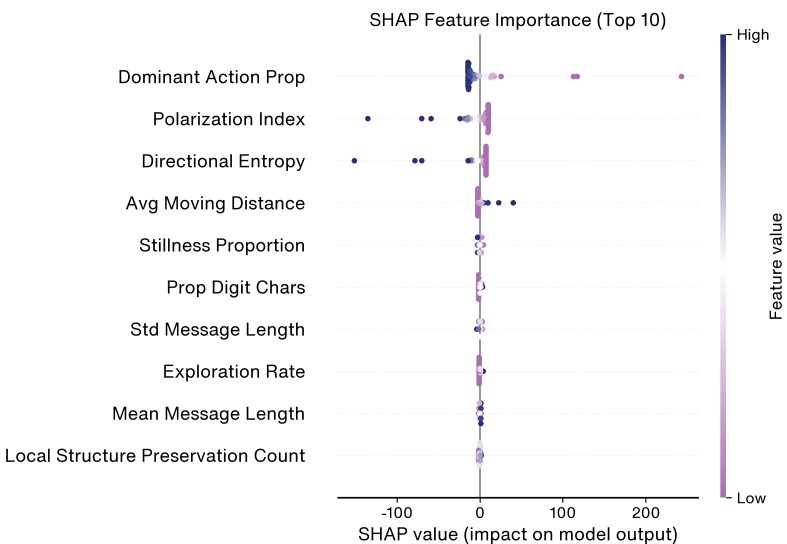

Figure S.20: Feature importance from the linear regression model for the Synchronization task.

### G.3 FORAGING TASK DYNAMICS

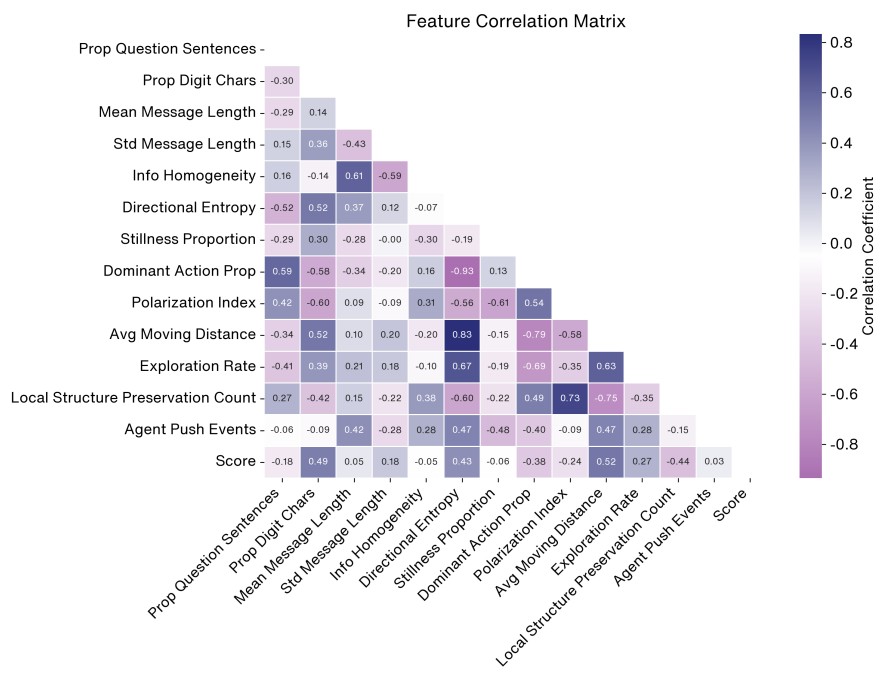

Figure S.21: Feature correlation matrix for the Foraging task.

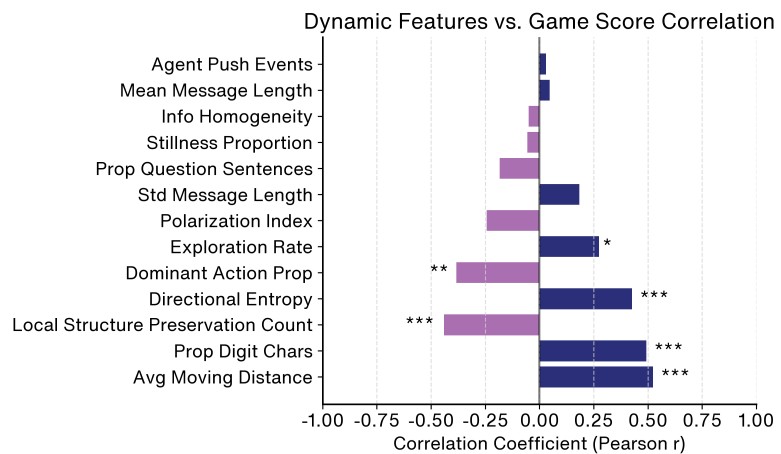

Figure S.22: Correlation of dynamic features with score for the Foraging task. Bars represent Pearson's $r$; * indicates $p < 0.05$,** indicates $p < 0.01$,*** indicates $p < 0.001$.

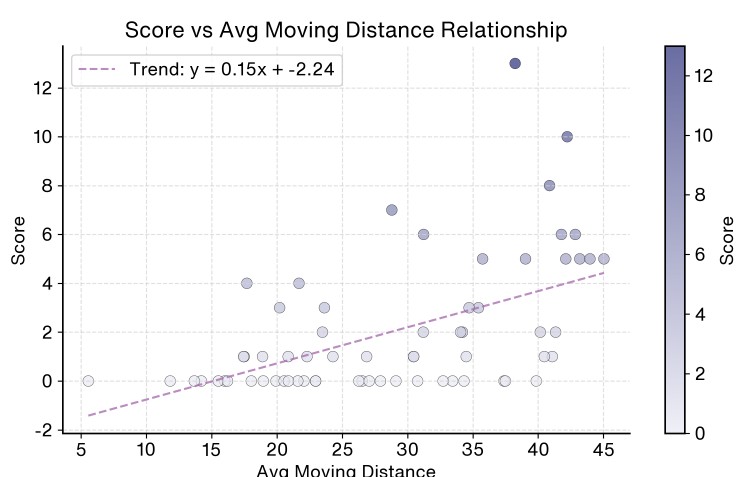

Figure S.23: Relationship between the top predictive dynamic feature (Avg. Moving Distance) and score for the Foraging task, with linear regression trend.

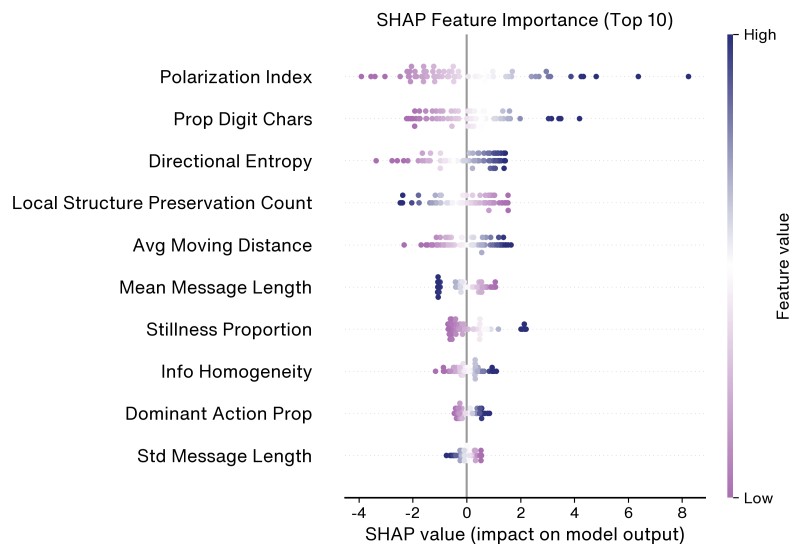

Figure S.24: Feature importance from the linear regression model for the Foraging task.

## G.4 FLOCKING TASK DYNAMICS

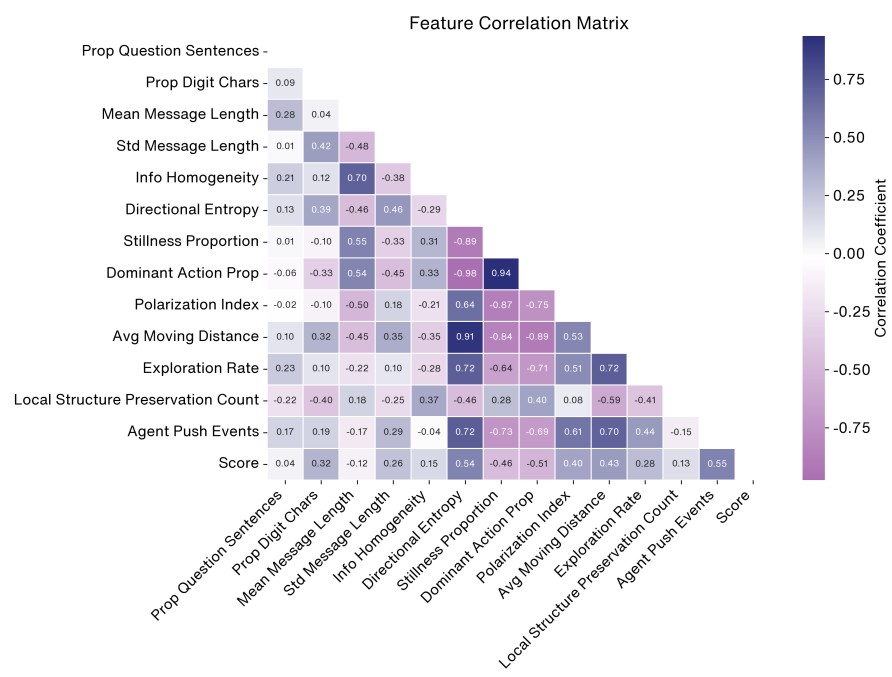

Figure S.25: Feature correlation matrix for the Flocking task.

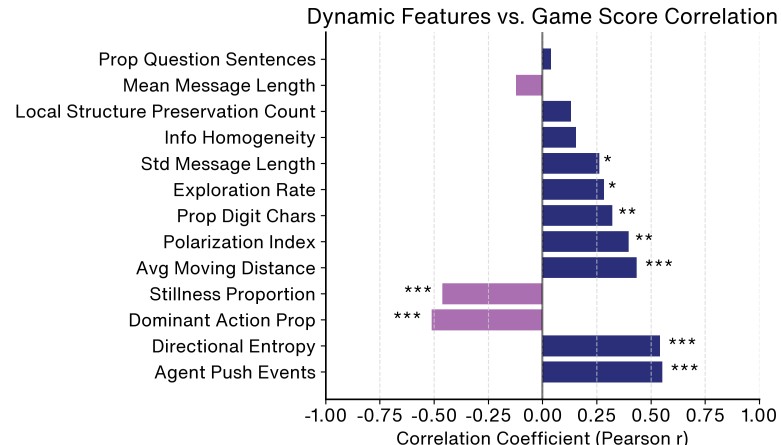

Figure S.26: Correlation of dynamic features with score for the Flocking task. Bars represent Pearson's $r$; * indicates $p < 0.05$,** indicates $p < 0.01$,*** indicates $p < 0.001$.

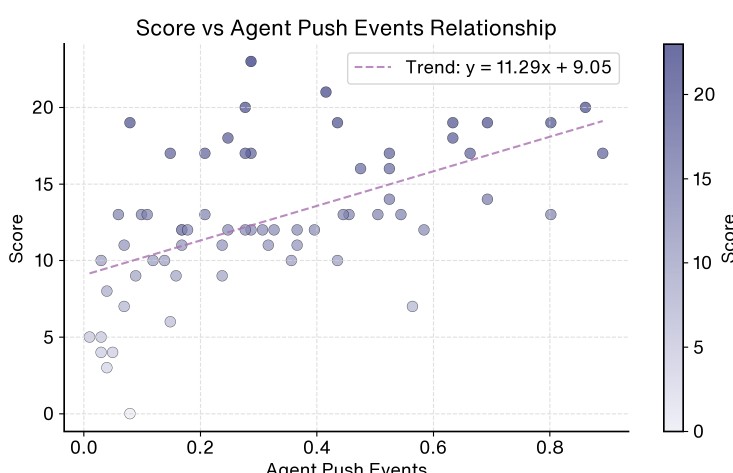

Figure S.27: Relationship between the top predictive dynamic feature (Avg. Agent Push Events) and score for the Flocking task, with linear regression trend.

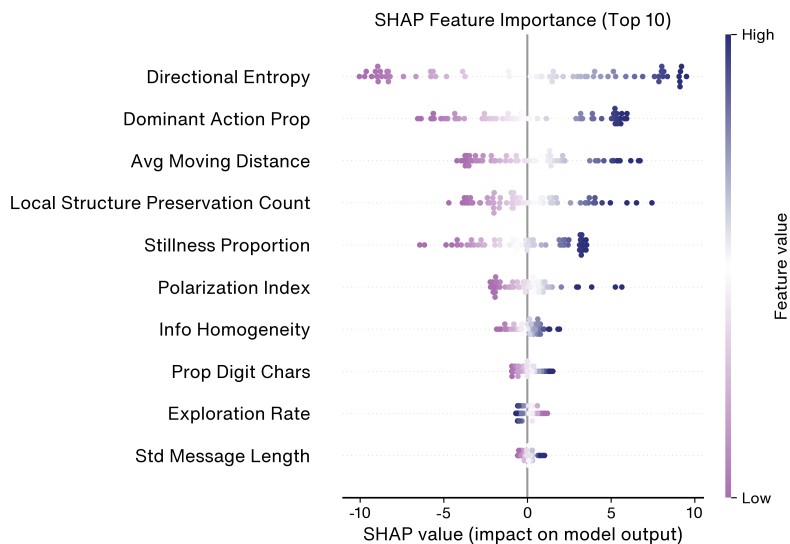

Figure S.28: Feature importance from the linear regression model for the Flocking task.

### G.5 Transport Task Dynamics

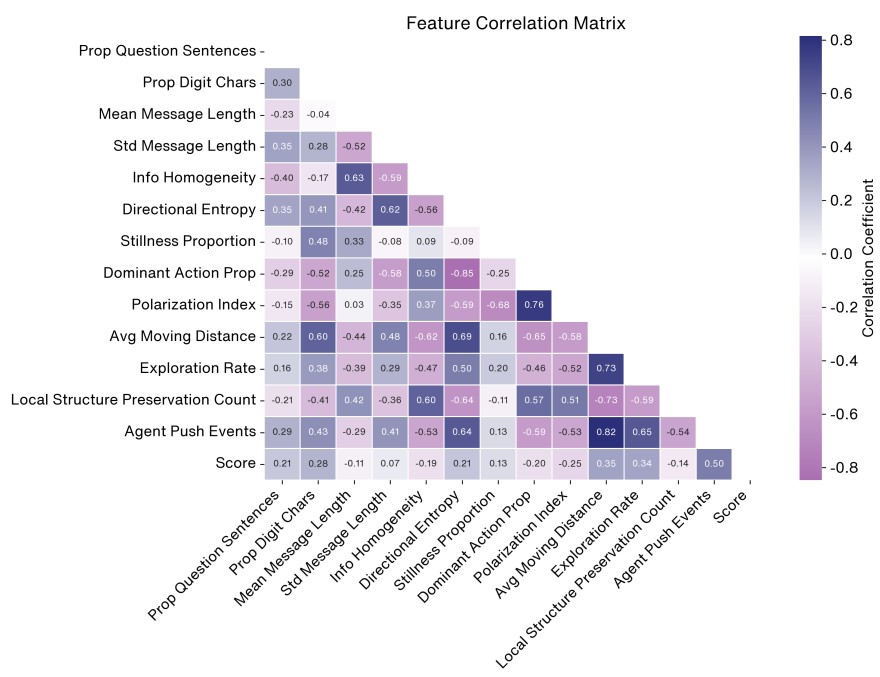

Figure S.29: Feature correlation matrix for the Transport task.

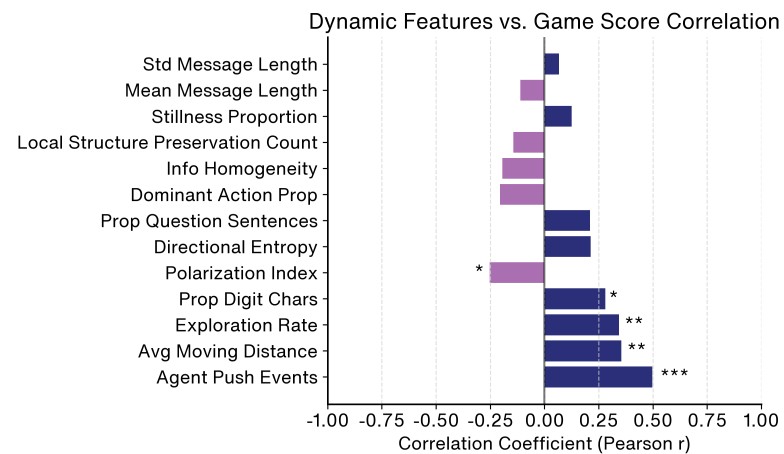

Figure S.30: Correlation of dynamic features with score for the Transport task. Bars represent Pearson's $r$; * indicates $p < 0.05$,** indicates $p < 0.01$,*** indicates $p < 0.001$.

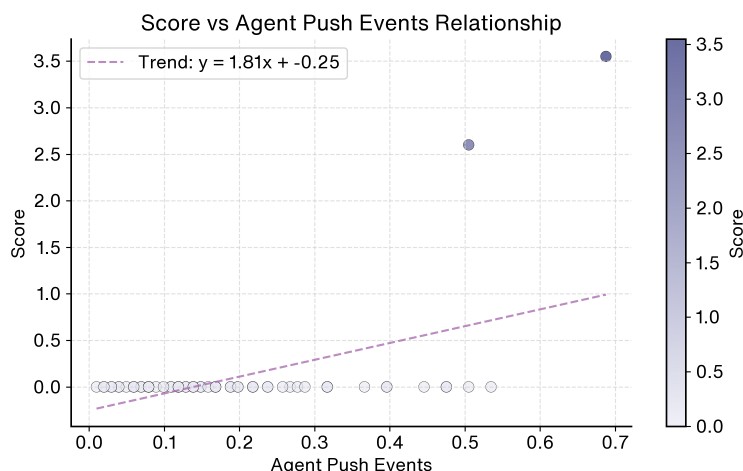

Figure S.31: Relationship between the top predictive dynamic feature (Avg. Agent Push Events) and score for the Transport task, with linear regression trend.

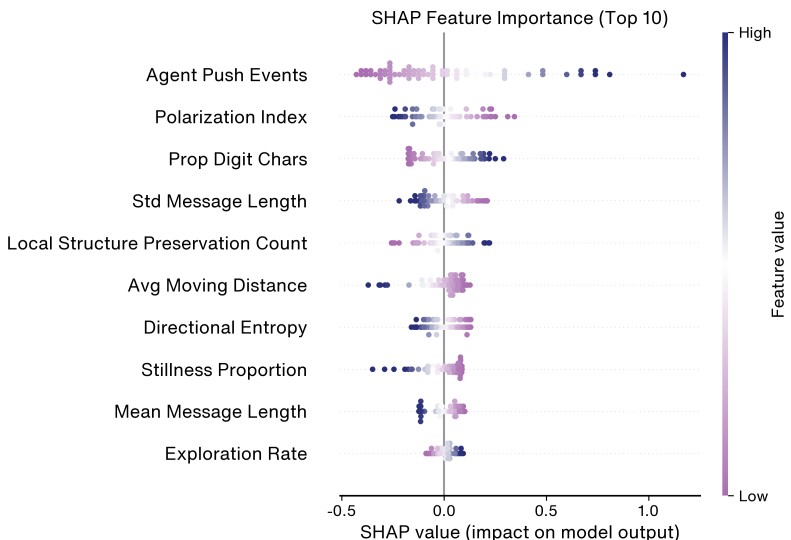

Figure S.32: Feature importance from the linear regression model for the Transport task.

## H KEYWORD ANALYSIS

We performed keyword extraction on sampled message data to understand terminology used by different models across tasks. Messages were preprocessed (lowercasing, punctuation removal, English stopword removal using NLTK (Bird et al., 2009)) and frequent terms identified for each model-task combination. This analysis, visualized in Fig. S.33, confirms agents' messages contained task-relevant vocabulary. The figure reveals keyword usage variations between LLM models performing the same task, suggesting model-specific communication styles. While relevant keywords indicate task understanding in communication, their frequency does not translate to coordination effectiveness, which appeared more linked to emergent physical dynamics and semantic consistency than keyword usage (Section 4.2).

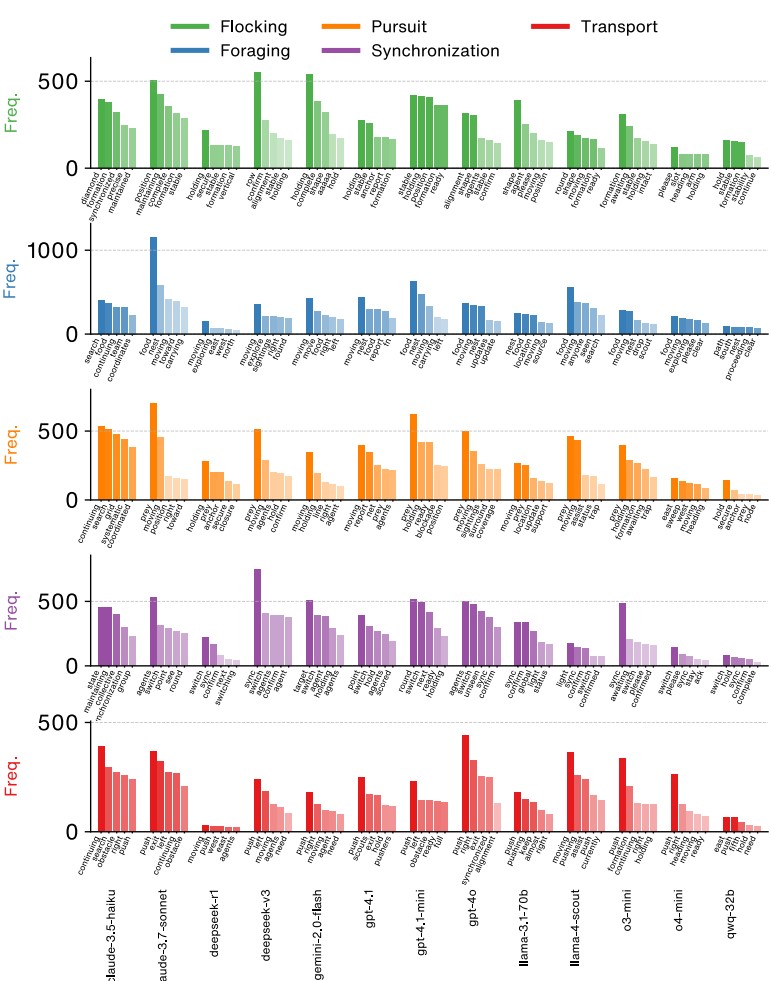

Figure S.33: **Keyword Frequency Analysis from Agent Messages.** Frequency of the top keywords extracted from agent messages, grouped by LLM model and task. Message data was preprocessed before frequency counting. The visualization highlights task-specific terminology (e.g., 'push' in Transport, 'food' in Foraging) and reveals variations in keyword usage across different models for the same task.

# I PARAMETER SENSITIVITY ANALYSIS

This appendix provides a more detailed textual elaboration on how agent performance responds to changes in local perception range ($k$, the size of the square view) and group size ($N$, the number of agents). A summary of these findings and their implications, along with a visual representation of the key trends, is presented in Section 4.5 of the main text, specifically in Figure 8. Here, we expand on the specific observations and the nuances of the analysis that underpin those summarized conclusions.

Our systematic investigation involved varying $N \in \{8, 12, 16\}$ and $k \in \{3, 5, 7\}$ for key tasks, using the `gemini-2.0-flash` model. Performance was measured by task-specific scores averaged over multiple simulation runs, the results of which are visually summarized in Figure 8 in the main text.

The data, as shown in Figure 8, reveals several important trends. Expanding the field of view from $k = 3$ to $k = 5$ consistently improved outcomes across diverse tasks like `Pursuit`, `Synchronization`, `Foraging`, and `Flocking`. This suggests that a minimal level of environmental awareness is crucial for agents to effectively coordinate, likely enabling better anticipation and response to neighbors' actions. However, as also indicated by Figure 8 and discussed in the main text, further increasing the view to $k = 7$ yielded only marginal gains and was sometimes less effective than $k = 5$, particularly in the `Transport` task which demands precise collective alignment. This plateau, and in some cases like the `Transport` task a performance dip with $k = 7$ compared to $k = 5$, implies a potential trade-off. While more information can be beneficial, an overly broad view might lead to information overload, making it harder for the LLM agents to discern critical local cues from a larger, potentially noisier, perceptual field. This could dilute focus on immediately relevant neighbors or environmental features crucial for tightly coupled maneuvers, such as the precise alignment needed in `Transport`. The increased cognitive load of processing a larger input space without a corresponding improvement in strategic depth might thus be counterproductive in certain scenarios. The effectiveness of $k = 5$ in our main experiments (Section 4) likely reflects a more optimal balance between sufficient environmental awareness and manageable perceptual complexity for the current LLM architectures in these zero-shot settings.

The influence of group size ($N$) presented a more complex picture, strongly modulated by task demands, as also detailed visually in Figure 8. Predictably, performance in `Transport` improved with more agents ($N = 16$ vs $N = 8$), as the task fundamentally relies on accumulating sufficient physical force. Conversely, `Foraging` performance deteriorated as $N$ increased, suggesting that larger groups introduced detrimental effects like congestion or interference near critical locations (nest 'N', food 'F'), outweighing any potential benefits. Intriguingly, `Pursuit` exhibited peak performance at an intermediate size ($N = 12$ compared to $N = 8$ and $N = 16$), hinting that while more agents can help initially encircle a target, too many may hinder coordinated containment through increased complexity and potential self-obstruction. `Flocking` remained relatively robust to changes in $N$ within the tested range.

These detailed textual elaborations are intended to complement the summarized findings and the visual data presented in Section 4.5 (specifically Figure 8). The varied scaling behaviors highlight a core challenge for LLM-based swarms: managing the increased interaction density and potential for conflicting local decisions in larger groups without centralized control. The sensitivity to both $k$ and $N$ underscores that robust swarm intelligence requires strategies adaptable to varying information availability and group dynamics, motivating evaluation across diverse parametric settings as discussed in Section 5.

# J ACTION ATTRIBUTION

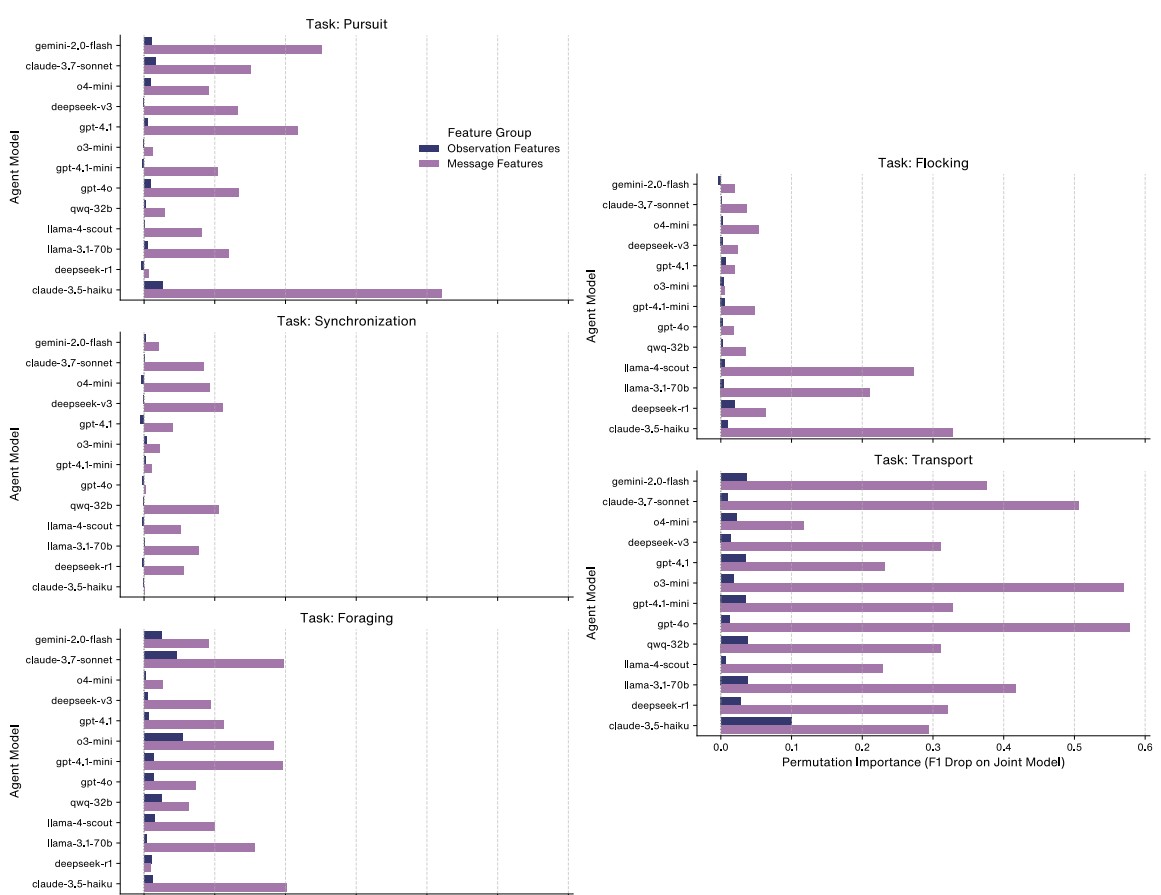

Figure S.34: Permutation importance (F1 Drop) of Observation Features and Message Features for predicting agent actions, broken down by task and individual LLM agent model. The subplots detail these importance scores across the five SwarmBench tasks for each evaluated model.

## K  THE SWARMBENCH DATASET

To support reproducibility and further research, we will release a comprehensive dataset encompassing all experimental runs detailed in this paper. The dataset is structured into experimental batches, with each batch containing a collection of JSON files that log the simulation parameters and detailed execution traces.

The primary components for each experimental batch are:

- **`meta_log.json`**: This central JSON file serves as an index for all individual simulation runs within a batch. It is a dictionary where each key is a unique run identifier (`run_id`). The corresponding value for each `run_id` is an object detailing the high-level configuration of that specific run, including parameters such as the Large Language Model employed (`model`), the number of participating agents (`num_agents`), and the maximum configured simulation rounds (`max_round`).

- **`agent_log_<run_id>.json`** files: For every run identified in `meta_log.json`, a corresponding agent log file is generated. This file stores a JSON array, with each element representing a detailed record for a single agent at a specific simulation round. These records capture the agent's local perception (`view`), the full `prompt` provided to its controlling LLM, the raw `response` from the LLM, and the subsequently parsed `action` (e.g., movement, task-specific command) and any `message` the agent chose to broadcast.

- **`game_log_<run_id>.json`** files: Complementing the agent logs, a game log file is also generated for each run. This file contains a JSON array, where each element chronicles the global state of the simulation environment at each round. This includes the complete 2D environment `grid`, the current `score` for the task, an array detailing the `id`, and global `x`, `y` coordinates for all `agents`, and a list of all `messages` that were broadcast by agents in the immediately preceding round and are thus available for perception in the current round.

This structured data will allow for in-depth analysis of agent behavior, communication patterns, and emergent group dynamics.

## L  ADDITIONAL EXPERIMENTS AND ANALYSES

### L.1  COMPARISON WITH RULE-BASED AND HEURISTIC BASELINES

To compare with rule-based MAS, We implemented and tested 15 rule-based agents for all five tasks, with strategies varying in complexity from simple reactive rules to communication-based heuristics (e.g., using BFS, potential fields, or role assignment; see Supplemetary Code `naive_strategies.py` for details), and each experiment was repeated 20 times for reliability. (see Table S.2)

The data shows a clear difference. For tasks requiring flexible adaptation, such as `Foraging` and `Synchronization`, zero-shot LLMs consistently outperformed all rule-based approaches. For problems that can be decomposed into simpler rules, like `Pursuit`, a specialized heuristic can match LLM performance. This demonstrates that the LLM's key advantage lies in its generality. A single general-purpose model achieves good performance across diverse problems, while the 15 specialized heuristics could not achieve similar breadth.

Table S.2: Performance of Rule-Based Baselines

| Method | Score | Std Dev |
|---|---|---|
| Flocking-1 | 2.85 | ±0.45 |
| Flocking-2 | 4.00 | ±0.00 |
| Flocking-3 | 4.68 | ±0.23 |
| Foraging-1 | 0.42 | ±0.60 |
| Foraging-2 | 0.00 | ±0.00 |
| Foraging-3 | 0.95 | ±0.97 |
| Pursuit-1 | 3.05 | ±4.19 |
| Pursuit-2 | 9.15 | ±6.12 |
| Pursuit-3 | 8.37 | ±5.13 |
| Sync.-1 | 0.00 | ±0.00 |
| Sync.-2 | 0.00 | ±0.00 |
| Sync.-3 | 0.35 | ±0.48 |
| Transport-1 | 0.00 | ±0.00 |
| Transport-2 | 0.00 | ±0.00 |
| Transport-3 | 0.00 | ±0.00 |

Table S.3: Performance of Human Baselines

| Method | Score | Std Dev |
|---|---|---|
| Flocking | 11.7 | ± 2.33 |
| Foraging | 20.40 | ± 2.82 |
| Sync. | 41.95 | ± 2.20 |
| Transport | 4.61 | ± 0.98 |

We also conducted experiments of 4 additional human player baselines. These baselines are intended to demonstrate scores that can be achieved when correct strategies are used. See Table S.3

In these human player baselines, `Foraging` and `Transport` use human player scores (i.e. replacing LLM with human), since they are too complex for any simple strategies. `Synchronization` uses an intuitive strategy, where agents first gather at a position so they can see each other, and then negotiate a common state (whether to turn on/off the light in the following round) basing on a voting mechanism that ensures synchronization. Flocking uses a simple strategy where agents choose their target location and try to move towards it, which requires no communication because they dynamically adjust among currently unoccupied locations.

## L.2 SCALABILITY ANALYSIS

We also investigate the scalability of SwarmBench with more agents. See Table S.4 below.

Table S.4: Scalability results on larger environments and more agents. The view size is set to $v = 5$. Each was run 3 times on `gemini-flash-2.0`.

| Task | Grid Size | $N$ | Avg Score | Std Dev |
|------|-----------|-----|-----------|---------|
| Flocking | 15×15 | 20 | 53.33 | ±6.60 |
| | | 30 | 23.33 | ±8.38 |
| | 20×20 | 20 | 62.67 | ±17.21 |
| | | 30 | 37.33 | ±7.59 |
| Foraging | 15×15 | 20 | 0.00 | ±0.00 |
| | | 30 | 0.00 | ±0.00 |
| | 20×20 | 20 | 0.00 | ±0.00 |
| | | 30 | 0.00 | ±0.00 |
| Pursuit | 15×15 | 20 | 1.00 | ±0.00 |
| | | 30 | 3.00 | ±0.82 |
| | 20×20 | 20 | 0.00 | ±0.00 |
| | | 30 | 0.67 | ±0.47 |
| Synchronization | 15×15 | 20 | 0.33 | ±0.47 |
| | | 30 | 0.33 | ±0.47 |
| | 20×20 | 20 | 0.67 | ±0.47 |
| | | 30 | 0.67 | ±0.47 |
| Transport | 15×15 | 20 | 0.00 | ±0.00 |
| | | 30 | 0.00 | ±0.00 |
| | 20×20 | 20 | 0.00 | ±0.00 |
| | | 30 | 0.00 | ±0.00 |

## L.3 Centralized vs. Decentralized Control

To investigate the performance on a centralized multi-agent system with a global view, we conducted experiments where we modified our framework to provide agents with a centralized, global view of the environment, removing the local perception constraint (see Table S.5). In this experiment, we designated `Agent_0` as the global commander. Under this setup, communication only occurs between the global commander and other agents, and only `Agent_0` is allowed to command the actions of other agents.

Table S.5: Performance on a centralized multi-agent system with a global view. Each was run 5 times on `gemini-flash-2.0`.

| Task | Average (Centralized) | Std Dev | Original (Decentralized) | Std Dev | % Change |
|---|---|---|---|---|---|
| Flocking | **9.90** | $\pm$ 0.67 | 9.40 | $\pm$ 0.80 | **+5%** |
| Foraging | **8.20** | $\pm$ 0.98 | 5.80 | $\pm$ 4.35 | **+41%** |
| Pursuit | **8.80** | $\pm$ 0.75 | 8.80 | $\pm$ 1.60 | **0%** |
| Sync. | **9.00** | $\pm$ 3.35 | 3.40 | $\pm$ 2.94 | **+164%** |
| Transport | **0.00** | $\pm$ 0.00 | 0.00 | $\pm$ 0.00 | **0%** |

## L.4 ROBUSTNESS ANALYSIS UNDER NOISE AND DELAY

We conducted additional experiments with stochastic communication noise (20% corruption chance) and delay (0-4 steps). See Table S.6.

Table S.6: Noise and delay experiment. (Each was run 5 times.)

| Model | Flocking | Foraging | Pursuit | Sync. | Transport | Original Avg. | Avg. | Change |
|---|---|---|---|---|---|---|---|---|
| sonnet-3.7 | 8.60 ±0.59 | 2.60 ±0.49 | 3.60 ±0.49 | 2.40 ±0.80 | 0.00 ±0.00 | 5.14 | 3.44 | -33.1% |
| deepseek-v3 | 8.20 ±0.70 | 1.60 ±0.74 | 1.60 ±1.85 | 1.00 ±0.63 | 0.00 ±0.00 | 3.44 | 2.48 | -27.9% |
| o4-mini | 8.30 ±1.03 | 4.60 ±2.06 | 12.00 ±2.00 | 1.80 ±0.40 | 0.00 ±0.00 | 5.32 | 5.34 | +0.4% |
| gemini-2.0-flash | 7.30 ±0.25 | 1.40 ±0.80 | 4.40 ±1.40 | 0.00 ±0.00 | 0.00 ±0.00 | 5.48 | 2.62 | -52.2% |
| llama-4-scout | 7.40 ±0.67 | 1.20 ±0.75 | 1.40 ±0.80 | 0.00 ±0.00 | 0.00 ±0.00 | 1.90 | 2.00 | +5.3% |
| llama-3.1-70B | 7.00 ±0.55 | 0.00 ±0.00 | 0.00 ±0.00 | 0.00 ±0.00 | 0.00 ±0.00 | 1.98 | 1.40 | -29.3% |
| gpt-4.1-mini | 7.10 ±0.97 | 0.00 ±0.00 | 0.00 ±0.00 | 1.00 ±1.10 | 0.00 ±0.00 | 1.68 | 1.62 | -3.6% |
| gpt-4.1 | 6.90 ±1.32 | 1.20 ±1.17 | 9.40 ±0.80 | 2.20 ±1.94 | 0.00 ±0.00 | 4.02 | 3.94 | -2.0% |
| qwq-32b | 5.10 ±0.38 | 1.20 ±1.47 | 1.60 ±1.85 | 0.00 ±0.00 | 0.00 ±0.00 | 2.02 | 1.58 | -21.8% |
| deepseek-r1 | 5.20 ±0.70 | 3.40 ±1.20 | 1.00 ±1.55 | 1.80 ±1.33 | 0.00 ±0.00 | 2.00 | 2.28 | +14.0% |
| gpt-4o | 4.50 ±0.55 | 0.00 ±0.00 | 2.40 ±1.40 | 0.80 ±0.40 | 0.00 ±0.00 | 2.36 | 1.54 | -34.7% |
| haiku-3.5 | 1.70 ±0.40 | 0.00 ±0.00 | 1.00 ±0.00 | 1.40 ±1.02 | 0.00 ±0.00 | 1.44 | 0.82 | -43.1% |
| o3-mini | 0.80 ±0.93 | 0.00 ±0.00 | 3.40 ±1.50 | 2.00 ±1.41 | 0.00 ±0.00 | 2.22 | 1.24 | -44.1% |

The results show clear differences in swarm robustness. Some models (e.g., o4-mini, deepseek-r1) were highly resilient, while others (e.g., gemini-2.0-flash) were fragile, with performance dropping by 52.19%. This highlights how some strategies depend heavily on ideal communication channels.

## L.5 QUANTITATIVE ANALYSIS OF FAILURE MODES

Here, we try to categorize LLM's failures more formally in terms of swarm theory.

For example, we frame the "Movement Bias" as premature convergence (March, 1991), where an LLM's pattern-matching capabilities cause it to lock into a suboptimal strategy. This can be directly measured by calculating the action imbalance of the agent. We use the Gini coefficient as the measure.

"Information Silos" result from spontaneous strong community structure formation in the agent network, where agents create tightly-knit groups with sparse inter-group connections (Girvan & Newman, 2002), causing network fragmentation and preventing global consensus, which can be measured by the number of connected components (constructing a graph using the agent's visual range).

In terms of the "Traffic Jams", for example, we can directly modify the Separation Rule in the Boids model (Reynolds, 1987) to calculate the repulsive forces that each agent should experience in order to evaluate whether congestion phenomena exist.

Finally, we attribute the "Memory of a Goldfish" to the LLM's volatile context window, which prevents the formation of a persistent, environment-mediated memory, a function served by stigmergy in natural swarms (Grass, 1959). This failure mode may be relatively difficult to measure directly, but we believe it can be indirectly measured by analyzing the impact of increasing the LLM's context window (number of memory frames) on the overall score.

We analyzed three failure modes quantitatively. As shown in Table S.7, we examined the tasks consistent with Figure 6 (i.e., Pursuit, Synchronization, and Foraging) to explore how these metrics correlate with scores. Interestingly, all three metrics showed negative correlations with the final scores, which aligns with our expectations.

Table S.7: Quantitative Analysis of Failure Modes

| Task | Metric | $r$ | $p$-value | Significance |
|---|---|---|---|---|
| Pursuit | Action Direction Imbalance | $-0.668$ | 0.000 | $* * *$ |
| Synchronization | Number of Connected Components | $-0.185$ | 0.140 | $-$ |
| Foraging | Separation Force | $-0.309$ | 0.012 | $*$ |

## L.6 ANALYSIS OF COMMUNICATION PROTOCOL CONVERGENCE

In our supplementary videos (e.g., `flocking_o4-mini_best.gif`), agents' messages often start as varied and verbose, but over time, they converge to a shorter, more structured format (e.g., "`Taking slot (2,4) TL=(2,3)`").

To quantify this phenomenon and its connection to task success, we introduced two new metrics: (a) the Increase in Information Homogeneity (semantic similarity) and (b) the Increase in Edit Distance Consistency (syntactic similarity) over the course of each game. We then correlated these metrics with the final task score (see Table S.8 and Table S.9, $*p < 0.05$, $**p < 0.01$, $***p < 0.001$)

Table S.8: Correlation of Protocol Convergence with Final Score (by Task)

| Task | Homogeneity | | Edit Consistency | |
|------|------|------|------|------|
| | $r$ | $p$-value | $r$ | $p$-value |
| Flocking | 0.382 | 0.002** | 0.458 | 0.000*** |
| Foraging | -0.102 | 0.419 | -0.027 | 0.832 |
| Pursuit | -0.447 | 0.000*** | -0.439 | 0.000*** |
| Sync. | -0.134 | 0.286 | -0.159 | 0.205 |
| Transport | 0.056 | 0.660 | -0.115 | 0.361 |

Table S.9: Correlation of Protocol Convergence with Final Score (by Model)

| Model | Homogeneity | | Edit Consistency | |
|------|------|------|------|------|
| | $r$ | $p$-value | $r$ | $p$-value |
| gemini-2.0-flash | 0.241 | 0.246 | -0.061 | 0.774 |
| o4-mini | 0.305 | 0.138 | 0.372 | 0.067 |
| claude-3.7-sonnet | -0.147 | 0.483 | -0.141 | 0.500 |
| gpt-4.1 | -0.044 | 0.835 | -0.031 | 0.884 |
| deepseek-v3 | -0.307 | 0.136 | -0.255 | 0.218 |
| llama-3.1-70B | -0.499 | 0.011* | -0.267 | 0.197 |
| gpt-4o | 0.031 | 0.884 | -0.231 | 0.267 |
| llama-4-scout | -0.016 | 0.941 | 0.094 | 0.656 |
| deepseek-r1 | -0.144 | 0.493 | -0.118 | 0.574 |
| qwq-32B | -0.135 | 0.520 | -0.139 | 0.508 |
| o3-mini | -0.528 | 0.007** | -0.450 | 0.024* |
| gpt-4.1-mini | -0.338 | 0.098 | -0.163 | 0.436 |
| claude-3.5-haiku | -0.005 | 0.980 | -0.058 | 0.783 |

First, contrary to intuition, a stronger convergence of the communication protocol often correlates with a lower final score. This suggests that for complex, dynamic scenarios, maintaining communicative diversity and richness may be more beneficial than prematurely locking into a rigid, simplistic protocol.

The `Flocking` task, however, is a notable exception with a significant positive correlation. This distinction is illuminating: `Flocking` is a task that benefits from converging on a simple, efficient protocol for sharing positional data without extraneous information. In contrast, more dynamic tasks may require richer communication to adapt. This provides a much deeper, data-driven reason for failure modes: a swarm's failure to converge on a protocol is not always a deficiency; in some cases, a rigid convergence is itself the strategy that fails.

### L.7 ANALYSIS OF LLM SAMPLING PARAMETERS

We used `temperature=1.0` and `top_p=1.0` in experiments in the main text for all LLMs. We also performed ablation experiments (see Table S.10-S.11). Each experiment was run at least twice.

Table S.10: Temperature experiments.

| Temp. | 0 | 0.5 | 1 | 1.5 |
|---|---|---|---|---|
| Flocking | **9.50**±0.50 | **9.50**±0.00 | 9.40±0.80 | 7.25±2.25 |
| Foraging | **7.00**±1.00 | 5.50±0.50 | 5.80±4.30 | 6.00±2.00 |
| Pursuit | 1.50±0.50 | 1.50±1.50 | **8.80**±1.60 | 1.50±0.50 |
| Sync. | 1.00±0.00 | 1.50±0.50 | **3.40**±2.90 | 0.50±0.50 |
| Transport | 0.00±0.00 | 0.00±0.00 | 0.00±0.00 | 0.00±0.00 |

Table S.11: Top-p experiments.

| top_p | 0.3 | 0.7 | 0.95 | 1 |
|---|---|---|---|---|
| Flocking | 7.00±2.00 | **10.00**±0.00 | **10.00**±0.00 | 9.40±0.80 |
| Foraging | 3.50±0.50 | **7.50**±0.50 | **7.50**±0.50 | 5.80±4.30 |
| Pursuit | 4.50±4.50 | 1.00±0.00 | 7.50±1.50 | **8.80**±1.60 |
| Sync. | 0.50±0.50 | 1.00±0.00 | 1.00±0.00 | **3.40**±2.90 |
| Transport | 0.00±0.00 | 0.00±0.00 | 0.00±0.00 | 0.00±0.00 |

Interestingly, when it comes to highly dynamic tasks (such as `Pursuit` and `Synchronization`), higher `temperature` and higher `top_p` perform better, which may suggest that these tasks require diversity; while other tasks show the opposite pattern.

# M    MODEL-SPECIFIC PERFORMANCE AND DYNAMICS VISUALIZATIONS

## M.1    CLAUDE-3.5-HAIKU

### M.1.1    PURSUIT TASK

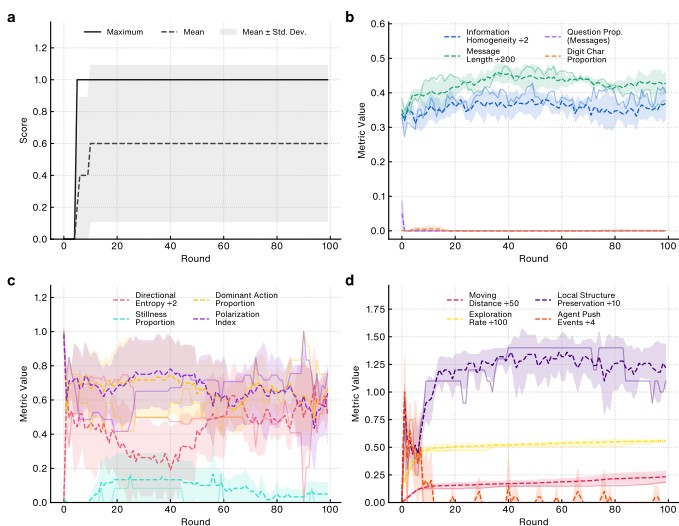

Figure S.35: **Metrics for `claude-3.5-haiku` on the Pursuit task.**

### M.1.2    SYNCHRONIZATION TASK

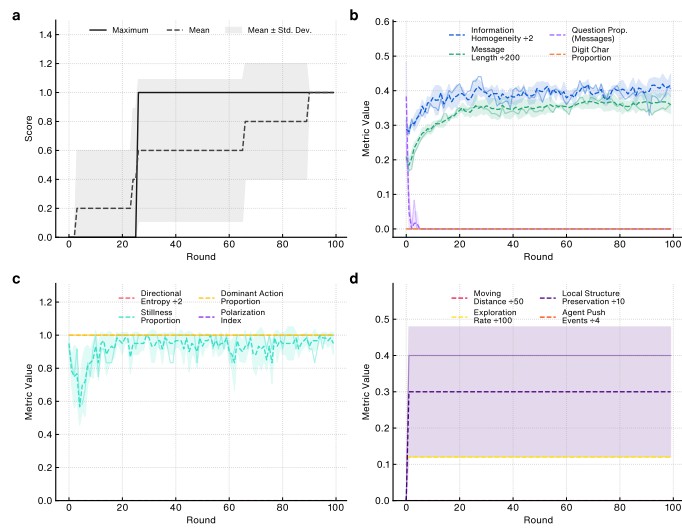

Figure S.36: **Metrics for `claude-3.5-haiku` on the Synchronization task.**

### M.1.3 FORAGING TASK

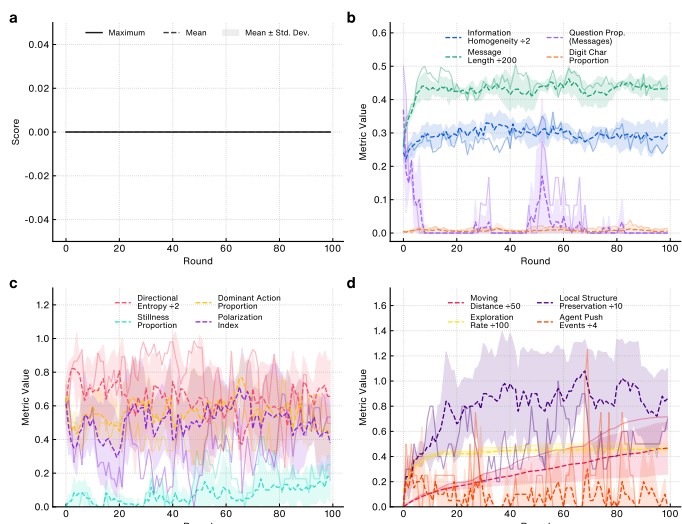

Figure S.37: **Metrics for `claude-3.5-haiku` on the Foraging task.**

### M.1.4 FLOCKING TASK

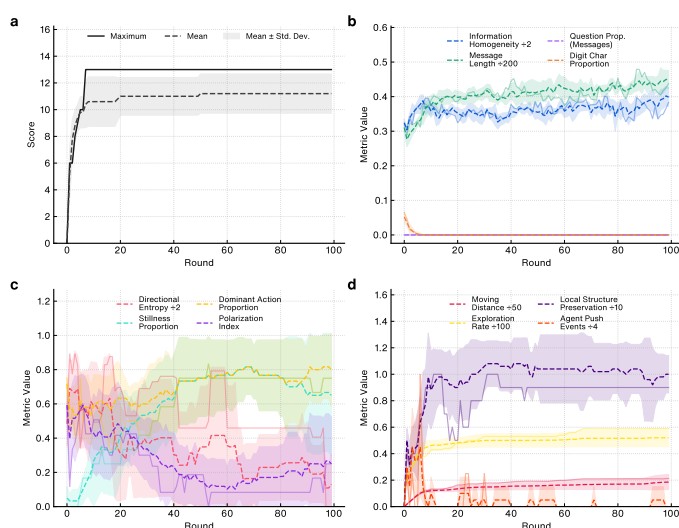

Figure S.38: **Metrics for `claude-3.5-haiku` on the Flocking task.**

## M.1.5 TRANSPORT TASK

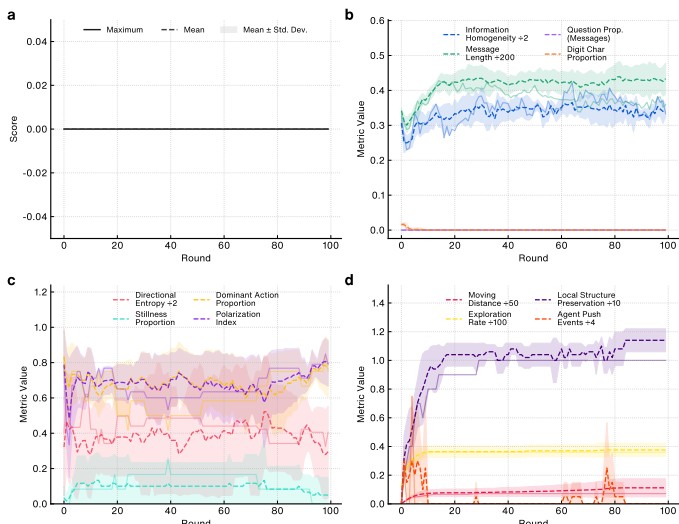

Figure S.39: **Metrics for `claude-3.5-haiku` on the Transport task.**

## M.2 CLAUDE-3.7-SONNET

### M.2.1 PURSUIT TASK

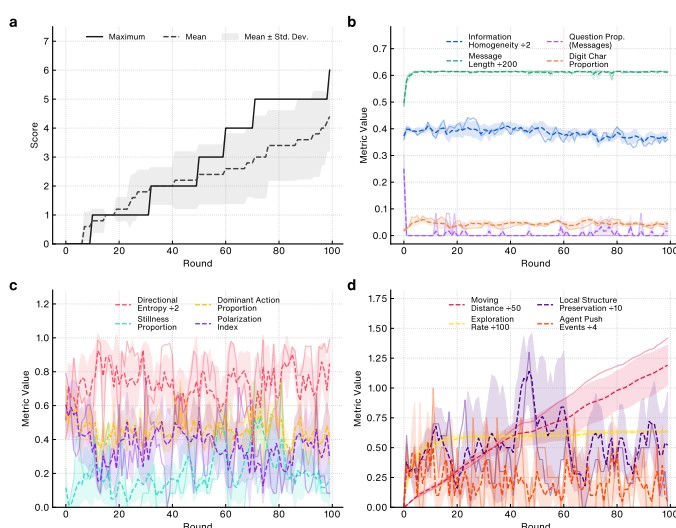

Figure S.40: **Metrics for `claude-3.7-sonnet` on the Pursuit task.**

### M.2.2 SYNCHRONIZATION TASK

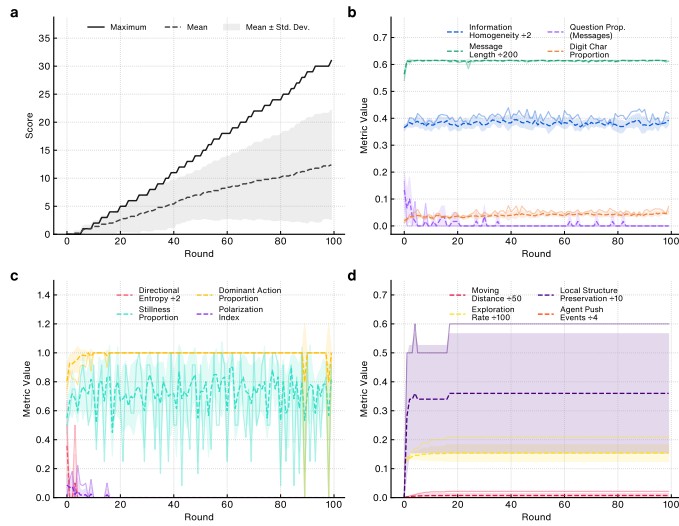

Figure S.41: **Metrics for `claude-3.7-sonnet` on the Synchronization task.**

### M.2.3 FORAGING TASK

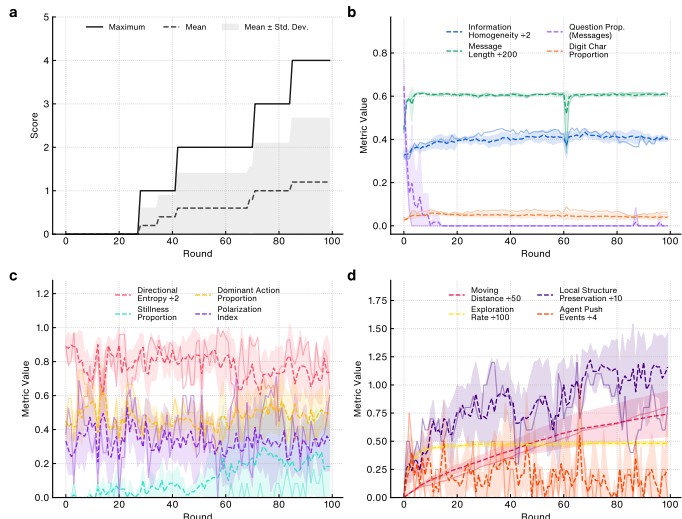

Figure S.42: **Metrics for `claude-3.7-sonnet` on the Foraging task.**

### M.2.4 FLOCKING TASK

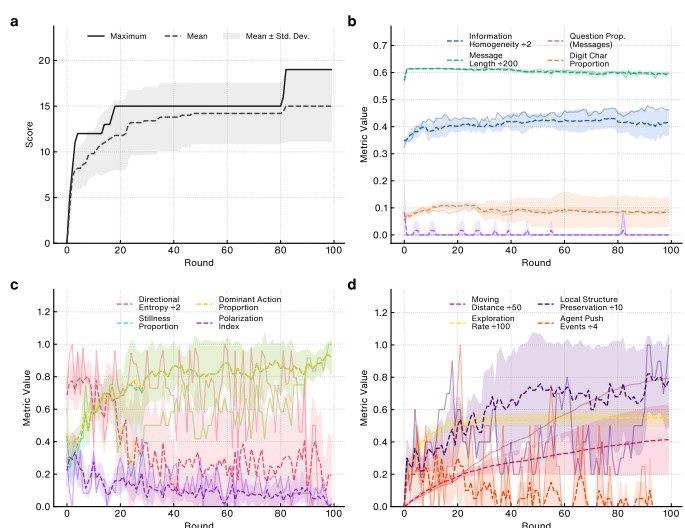

Figure S.43: **Metrics for `claude-3.7-sonnet` on the Flocking task.**

## M.2.5 TRANSPORT TASK

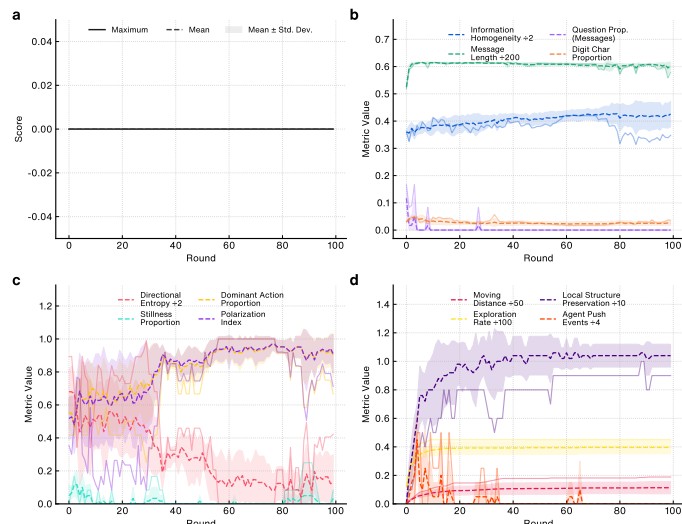

Figure S.44: **Metrics for `claude-3.7-sonnet` on the Transport task.**

## M.3 DEEPSEEK-R1

### M.3.1 PURSUIT TASK

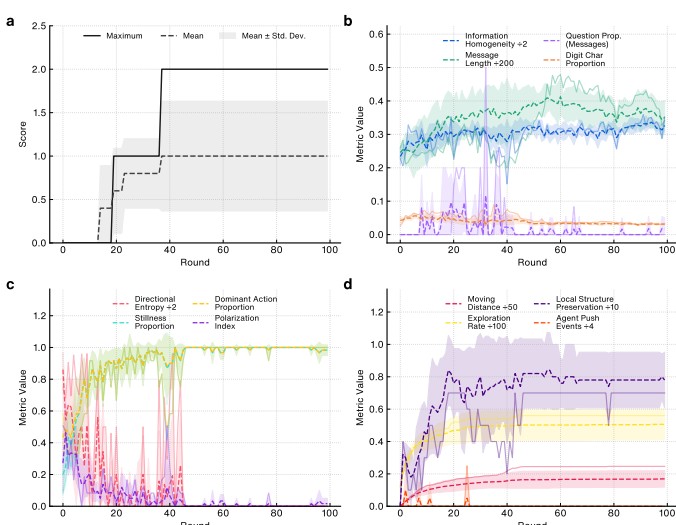

Figure S.45: **Metrics for deepseek-r1 on the Pursuit task.**

### M.3.2 SYNCHRONIZATION TASK

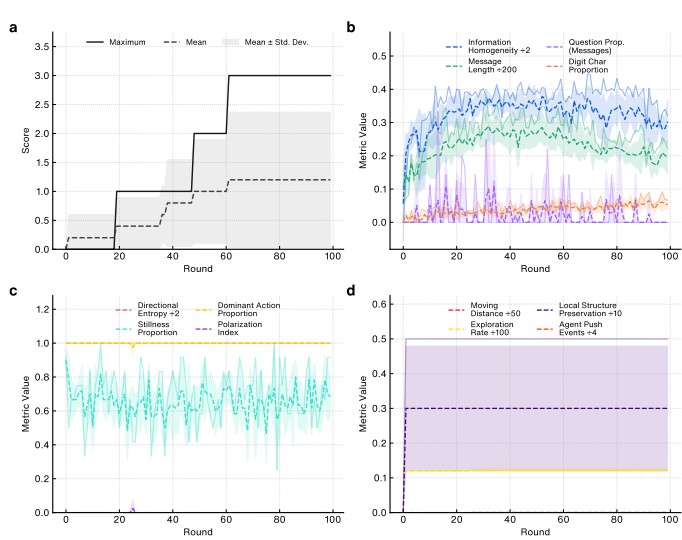

Figure S.46: **Metrics for deepseek-r1 on the Synchronization task.**

### M.3.3 FORAGING TASK

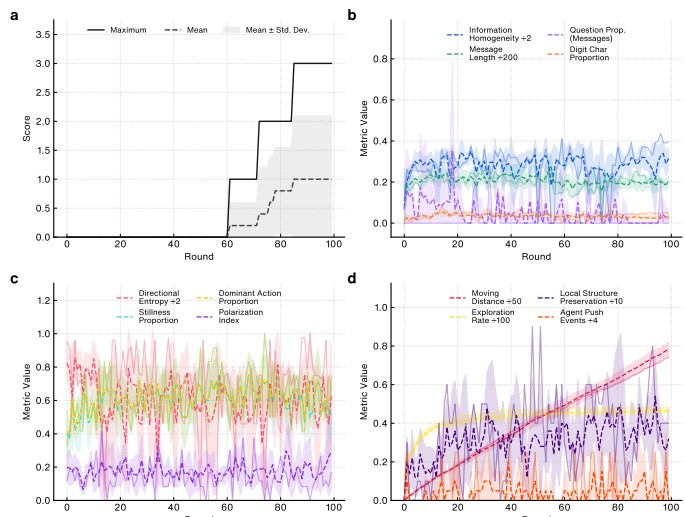

Figure S.47: **Metrics for `deepseek-r1` on the Foraging task.**

### M.3.4 FLOCKING TASK

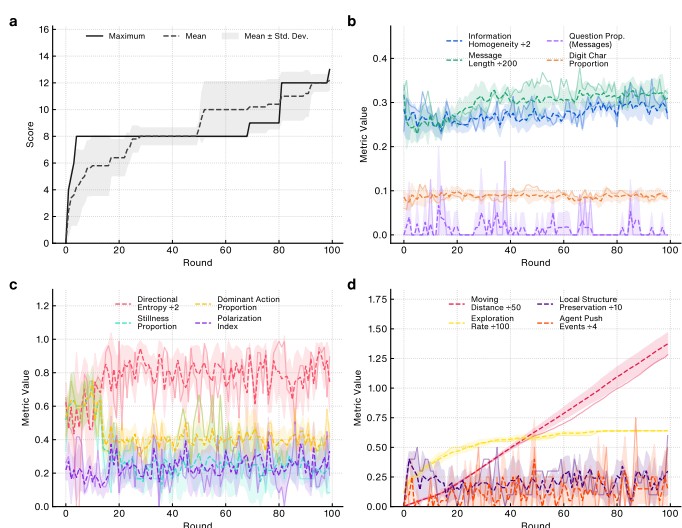

Figure S.48: **Metrics for `deepseek-r1` on the Flocking task.**

## M.3.5 TRANSPORT TASK

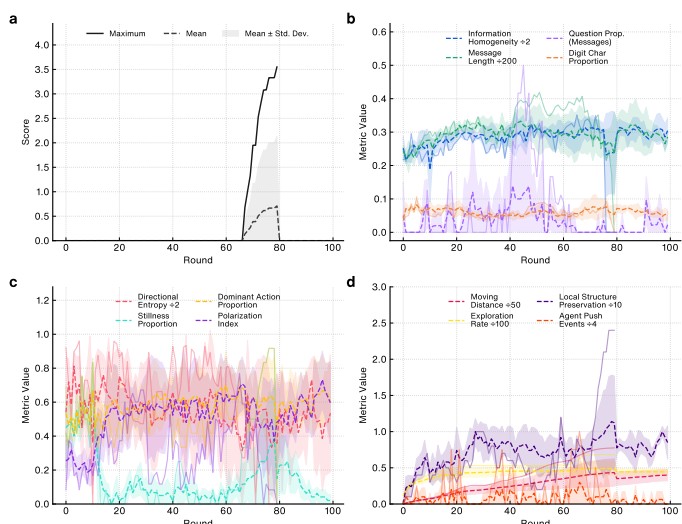

Figure S.49: **Metrics for `deepseek-r1` on the Transport task.**

## M.4 DEEPSEEK-V3 (0324)

### M.4.1 PURSUIT TASK

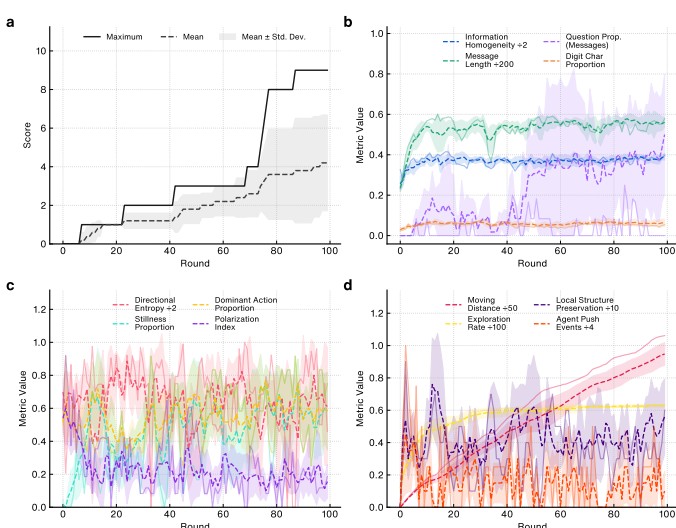

Figure S.50: **Metrics for `deepseek-v3` on the Pursuit task.**

### M.4.2 SYNCHRONIZATION TASK

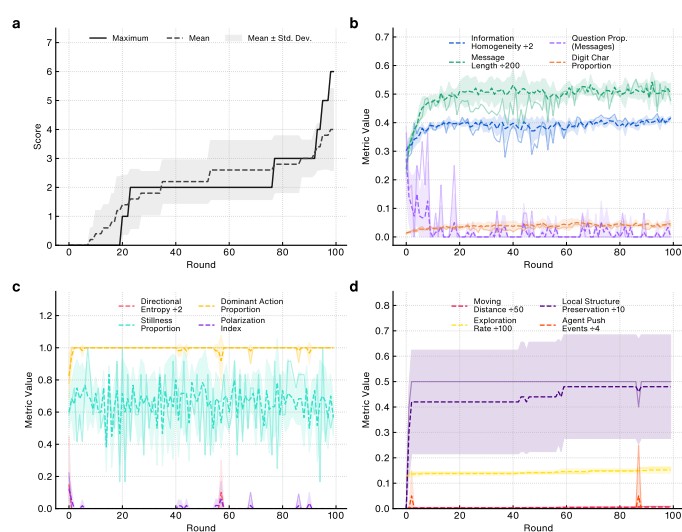

Figure S.51: **Metrics for `deepseek-v3` on the Synchronization task.**

### M.4.3 FORAGING TASK

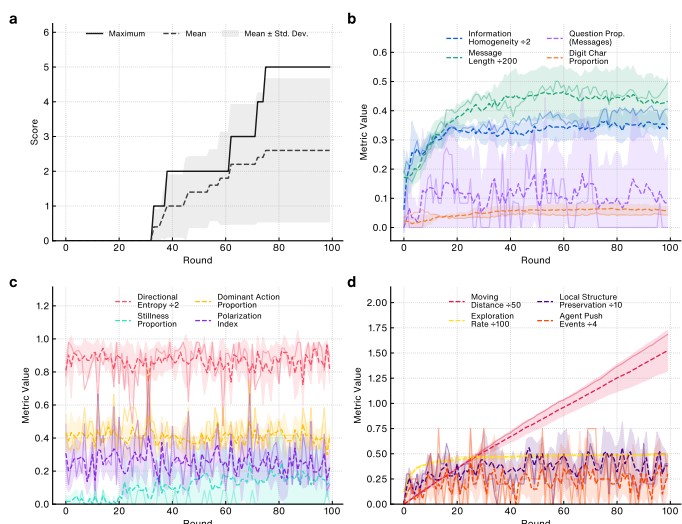

Figure S.52: **Metrics for `deepseek-v3` on the Foraging task.**

### M.4.4 FLOCKING TASK

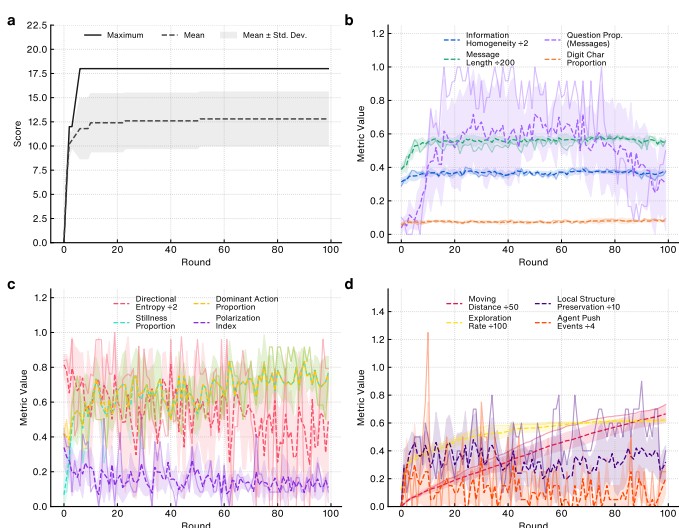

Figure S.53: **Metrics for `deepseek-v3` on the Flocking task.**

## M.4.5 TRANSPORT TASK

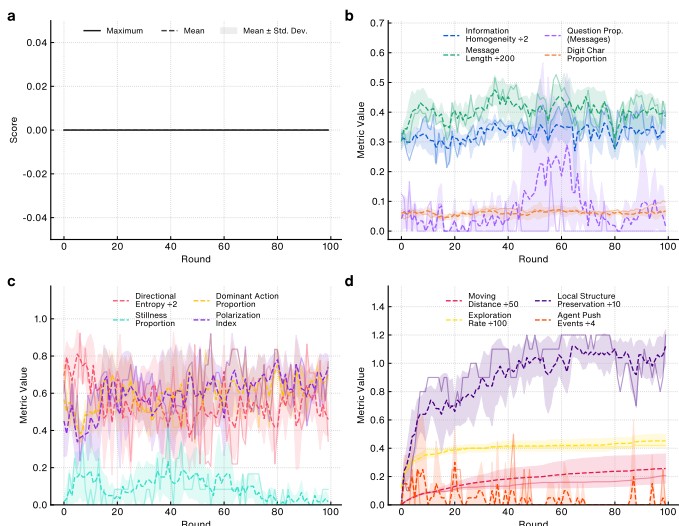

Figure S.54: **Metrics for `deepseek-v3` on the Transport task.**

**M.5** GEMINI-2.0-FLASH

**M.5.1 PURSUIT TASK**

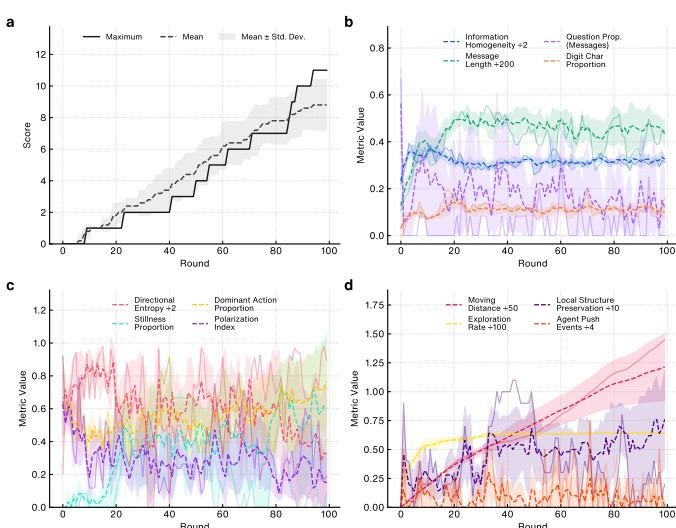

Figure S.55: **Metrics for gemini-2.0-flash on the Pursuit task.**

**M.5.2 SYNCHRONIZATION TASK**

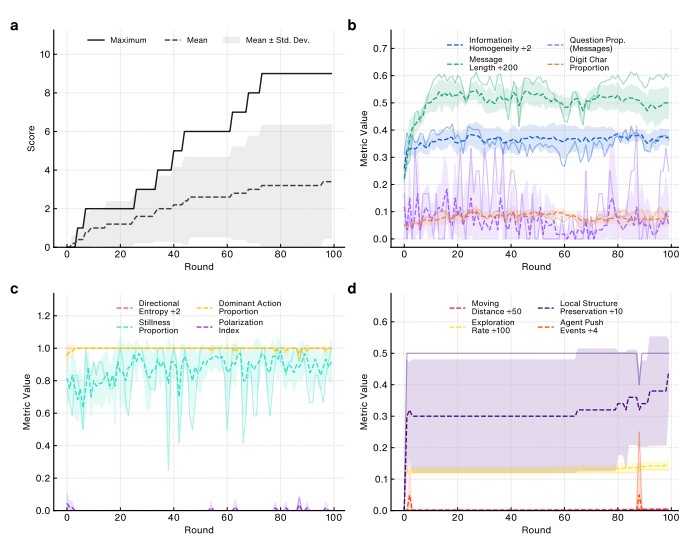

Figure S.56: **Metrics for gemini-2.0-flash on the Synchronization task.**

### M.5.3 FORAGING TASK

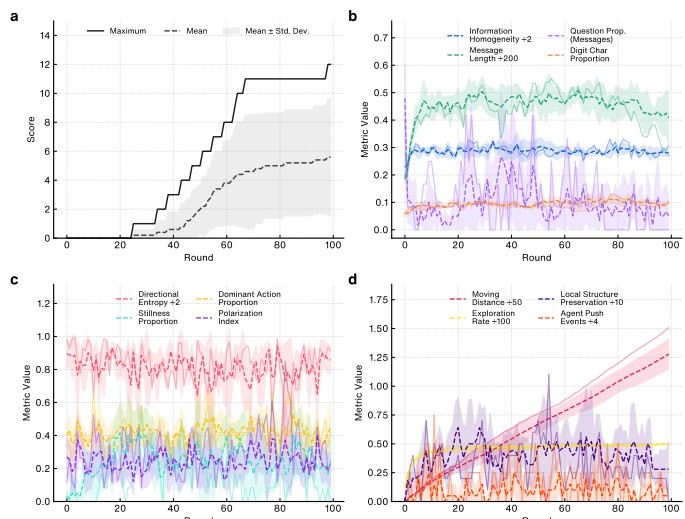

Figure S.57: **Metrics for `gemini-2.0-flash` on the Foraging task.**

### M.5.4 FLOCKING TASK

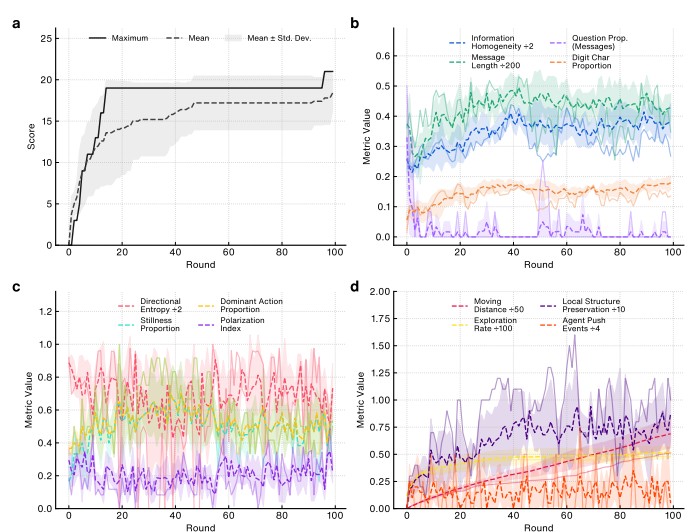

Figure S.58: **Metrics for `gemini-2.0-flash` on the Flocking task.**

## M.5.5 TRANSPORT TASK

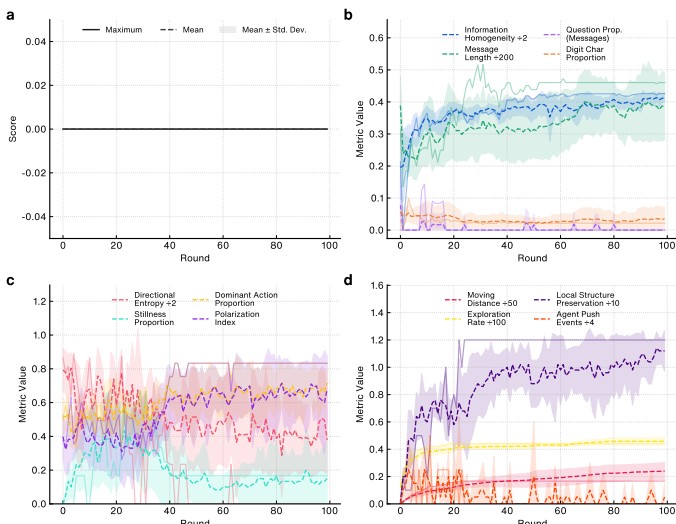

Figure S.59: **Metrics for `gemini-2.0-flash` on the Transport task.**

## M.6 GPT-4.1

### M.6.1 PURSUIT TASK

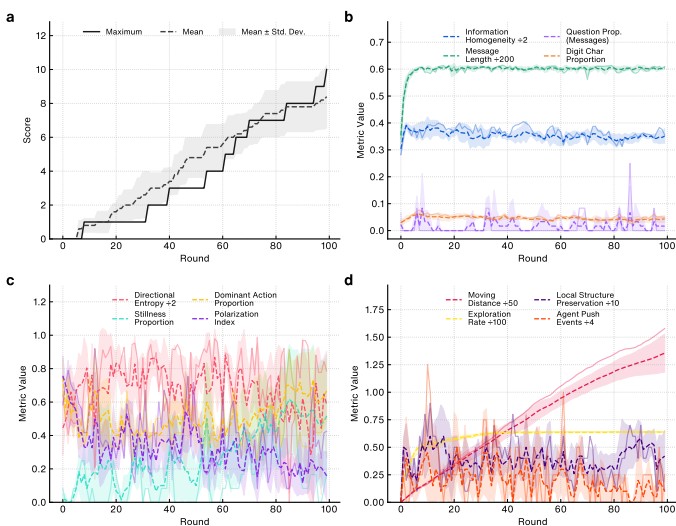

Figure S.60: **Metrics for gpt-4.1 on the Pursuit task.**

### M.6.2 SYNCHRONIZATION TASK

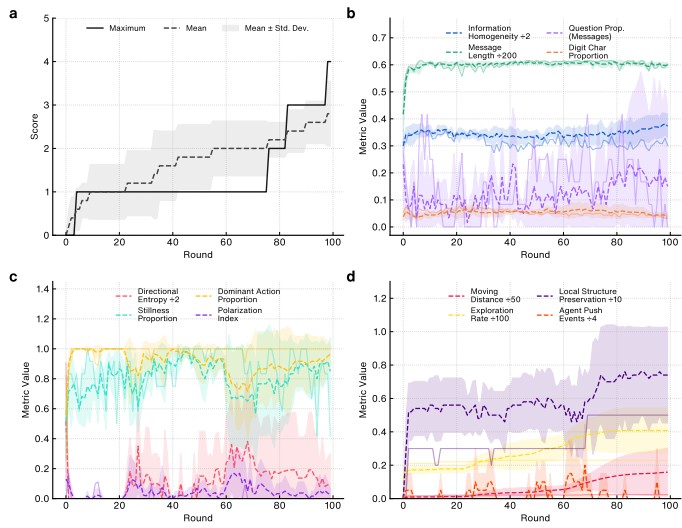

Figure S.61: **Metrics for gpt-4.1 on the Synchronization task.**

### M.6.3 FORAGING TASK

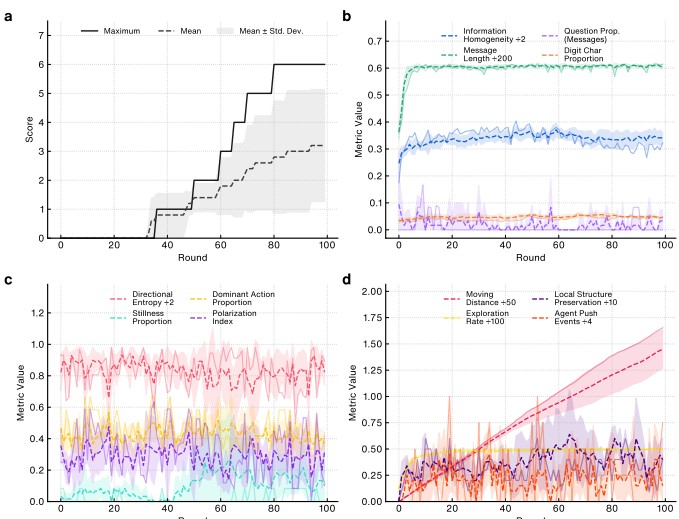

Figure S.62: **Metrics for gpt-4.1 on the Foraging task.**

### M.6.4 FLOCKING TASK

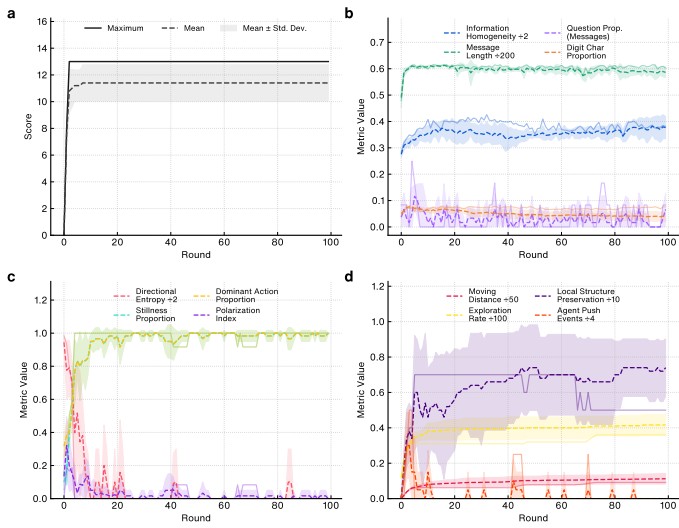

Figure S.63: **Metrics for gpt-4.1 on the Flocking task.**

## M.6.5 TRANSPORT TASK

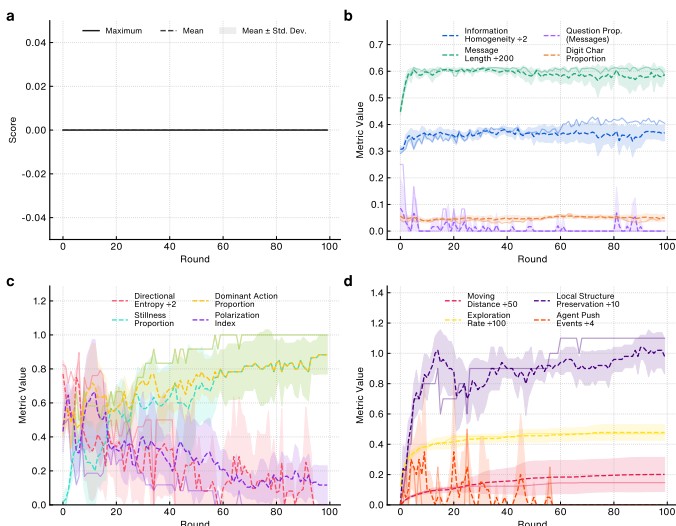

Figure S.64: **Metrics for `gpt-4.1` on the Transport task.**

## M.7 GPT-4.1-MINI

### M.7.1 PURSUIT TASK

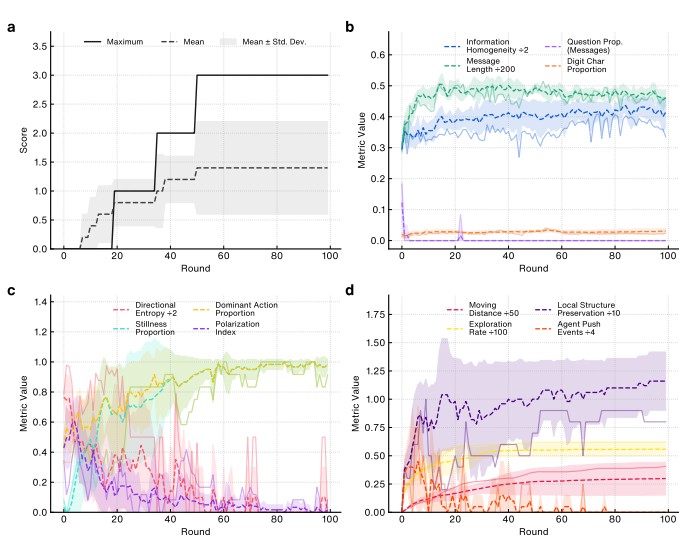

Figure S.65: **Metrics for gpt-4.1-mini on the Pursuit task.**

### M.7.2 SYNCHRONIZATION TASK

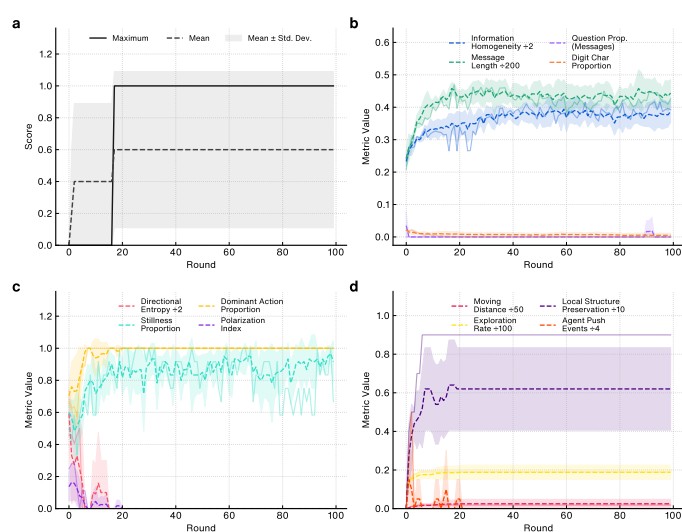

Figure S.66: **Metrics for gpt-4.1-mini on the Synchronization task.**

### M.7.3 FORAGING TASK

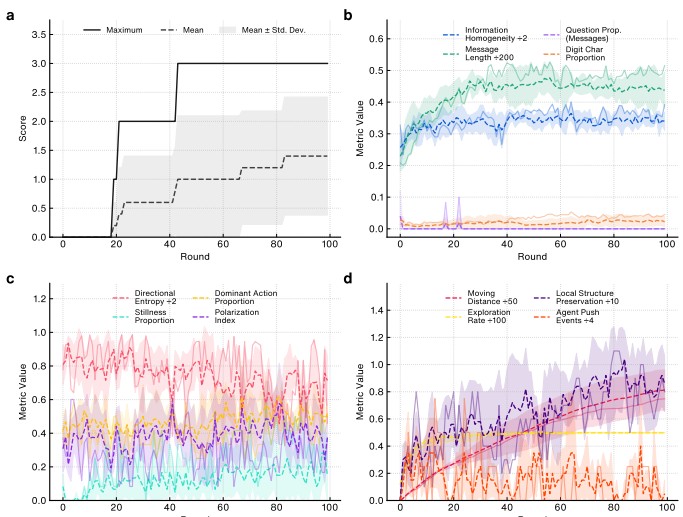

Figure S.67: **Metrics for `gpt-4.1-mini` on the Foraging task.**

### M.7.4 FLOCKING TASK

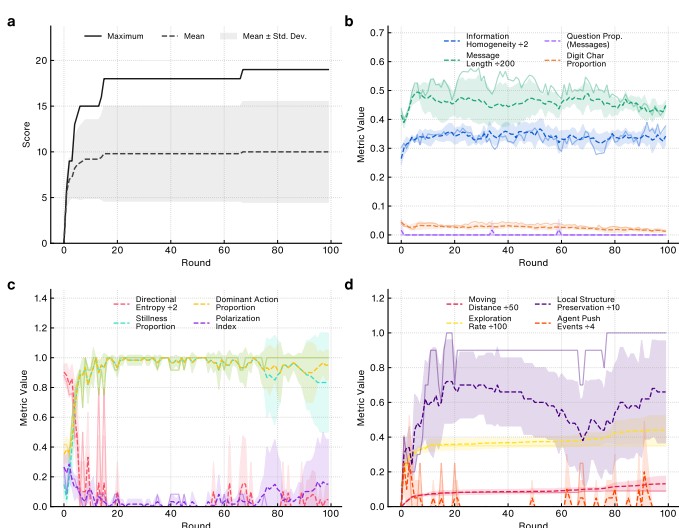

Figure S.68: **Metrics for `gpt-4.1-mini` on the Flocking task.**

## M.7.5 TRANSPORT TASK

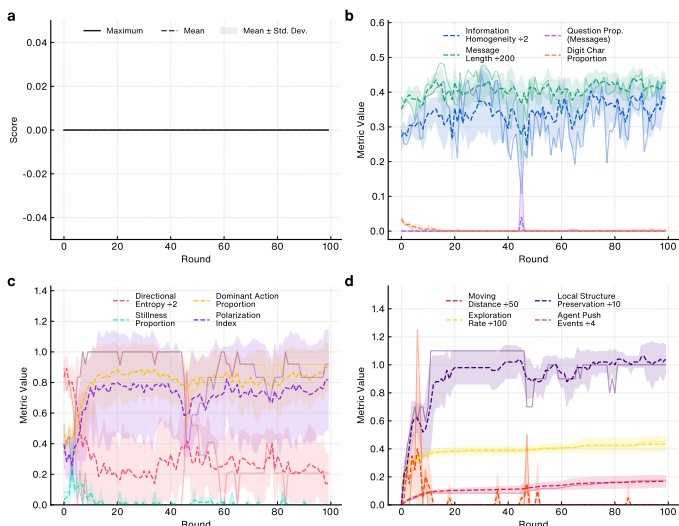

Figure S.69: **Metrics for `gpt-4.1-mini` on the Transport task.**

## M.8 GPT-4O

### M.8.1 PURSUIT TASK

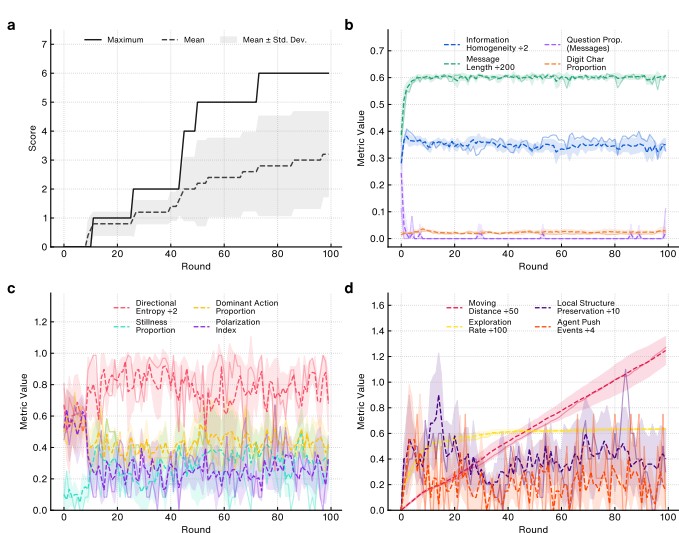

Figure S.70: **Metrics for gpt-4o on the Pursuit task.**

### M.8.2 SYNCHRONIZATION TASK

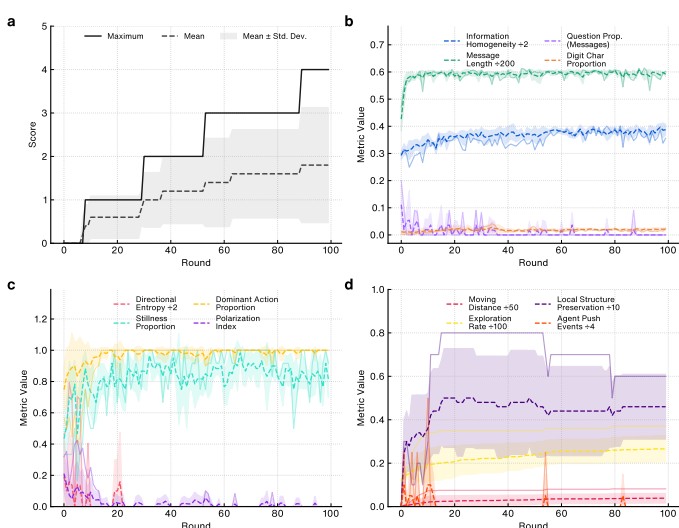

Figure S.71: **Metrics for gpt-4o on the Synchronization task.**

### M.8.3 FORAGING TASK

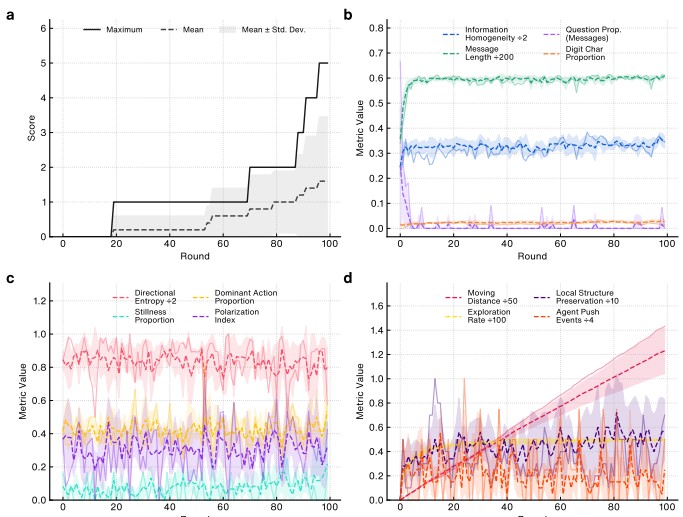

Figure S.72: **Metrics for gpt-4o on the Foraging task.**

### M.8.4 FLOCKING TASK

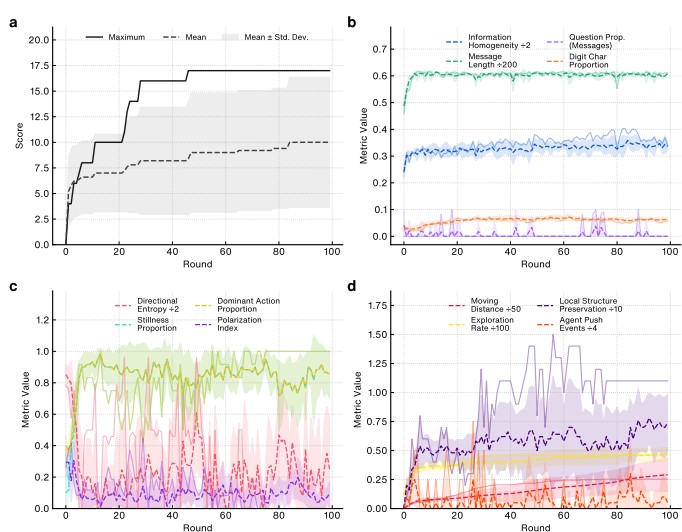

Figure S.73: **Metrics for gpt-4o on the Flocking task.**

## M.8.5 TRANSPORT TASK

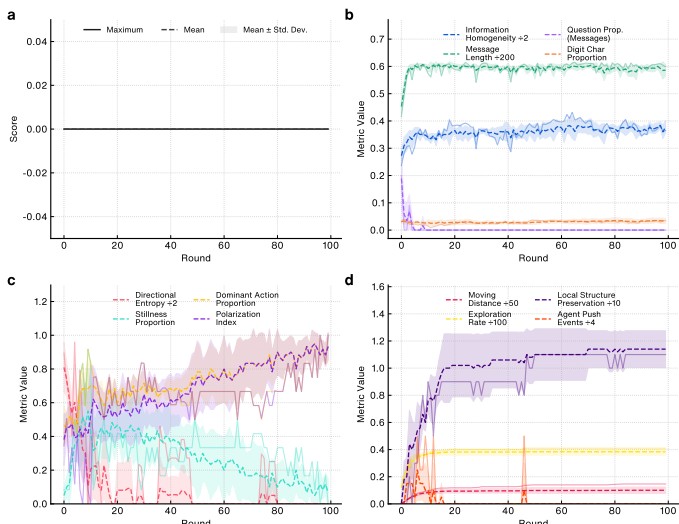

Figure S.74: **Metrics for gpt-4o on the Transport task.**

## M.9 LLAMA-3.1-70B

### M.9.1 PURSUIT TASK

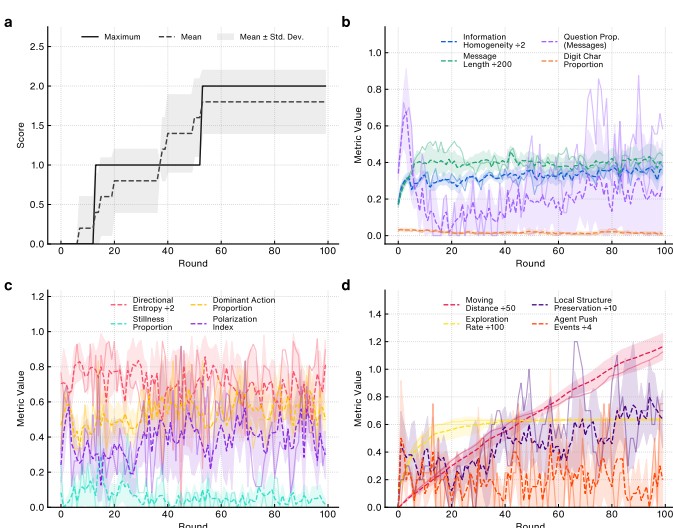

Figure S.75: **Metrics for `llama-3.1-70b` on the Pursuit task.**

### M.9.2 SYNCHRONIZATION TASK

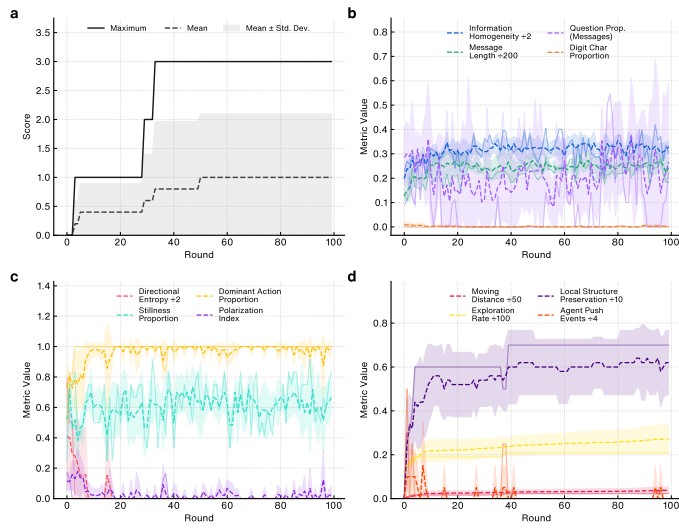

Figure S.76: **Metrics for `llama-3.1-70b` on the Synchronization task.**

### M.9.3 FORAGING TASK

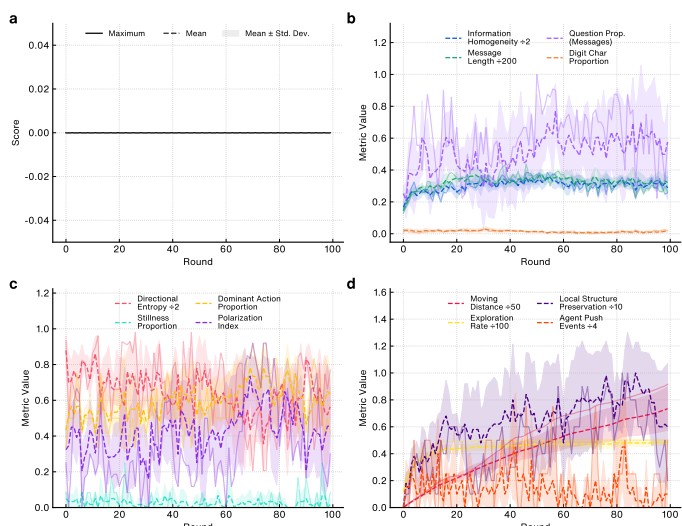

Figure S.77: **Metrics for `llama-3.1-70b` on the Foraging task.**

### M.9.4 FLOCKING TASK

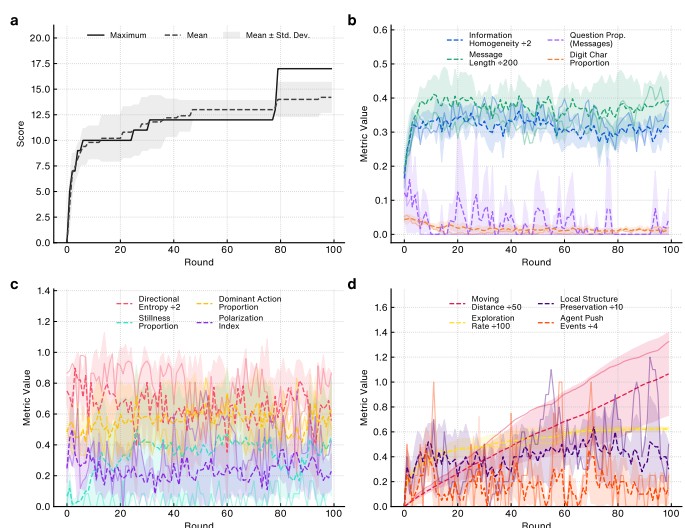

Figure S.78: **Metrics for `llama-3.1-70b` on the Flocking task.**

## M.9.5 TRANSPORT TASK

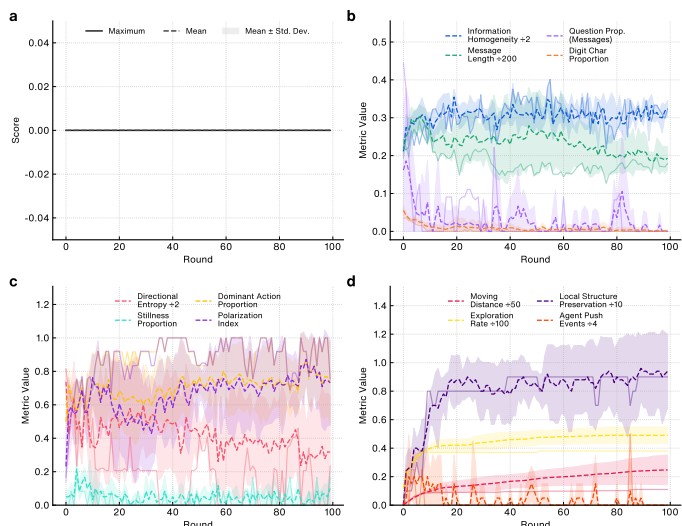

Figure S.79: **Metrics for `llama-3.1-70b` on the Transport task.**

## M.10 LLAMA-4-SCOUT

### M.10.1 PURSUIT TASK

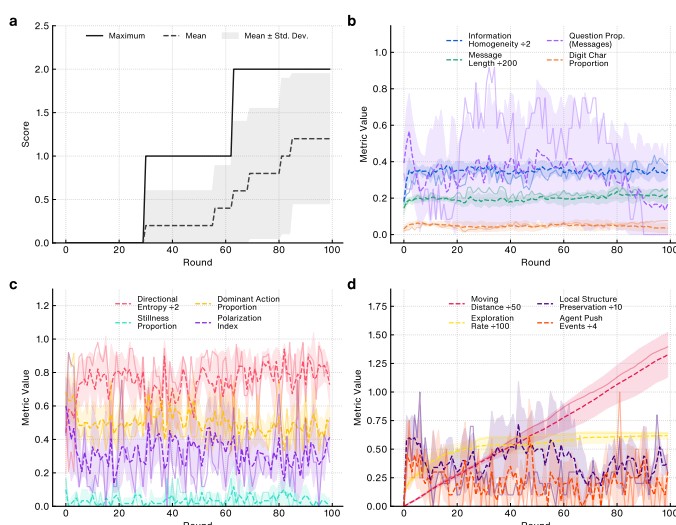

Figure S.80: **Metrics for `llama-4-scout` on the Pursuit task.**

### M.10.2 SYNCHRONIZATION TASK

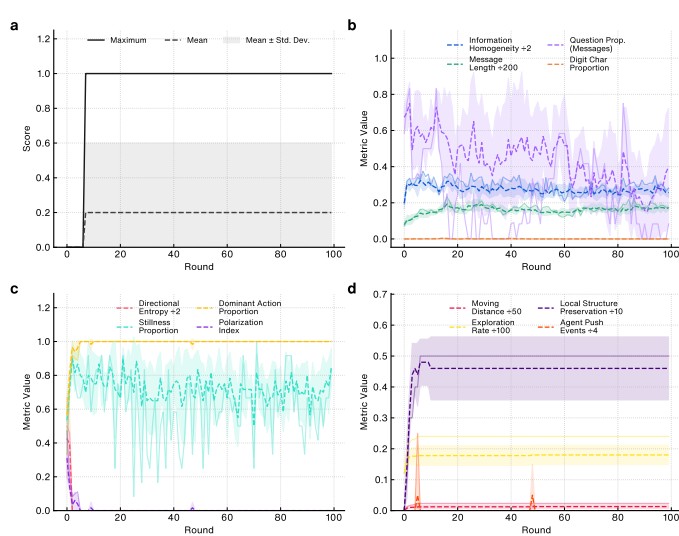

Figure S.81: **Metrics for `llama-4-scout` on the Synchronization task.**

### M.10.3 FORAGING TASK

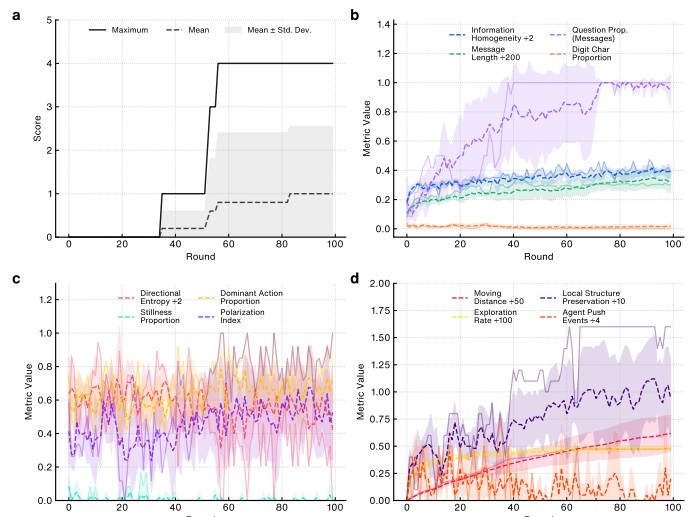

Figure S.82: **Metrics for `llama-4-scout` on the Foraging task.**

### M.10.4 FLOCKING TASK

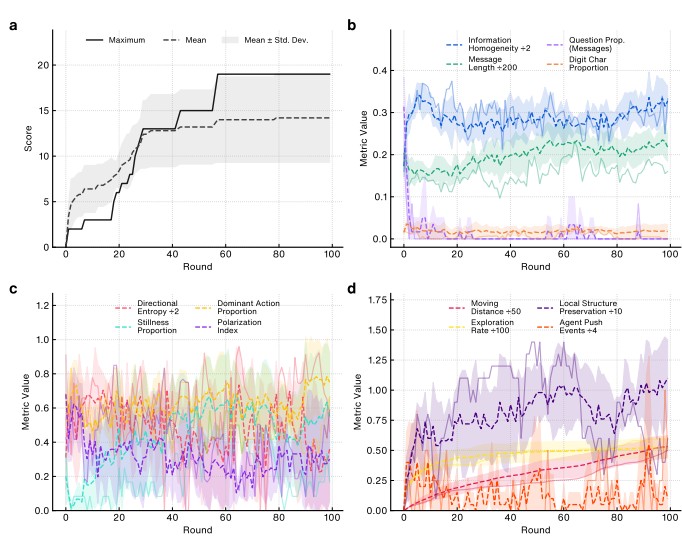

Figure S.83: **Metrics for `llama-4-scout` on the Flocking task.**

## M.10.5 TRANSPORT TASK

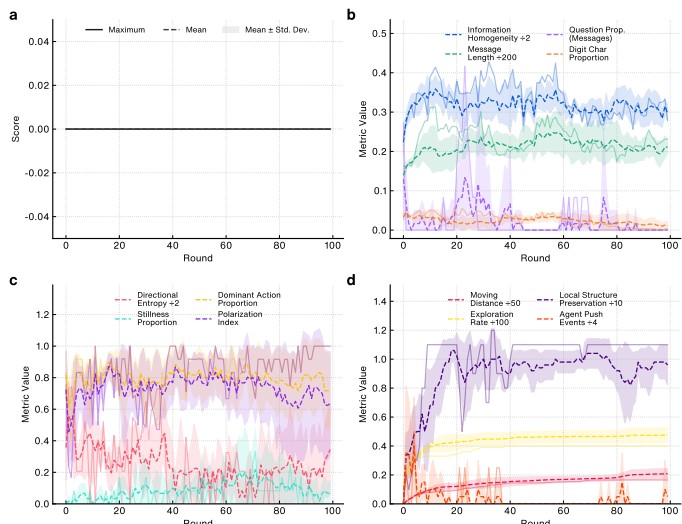

Figure S.84: **Metrics for `llama-4-scout` on the Transport task.**

## M.11 o3-MINI

### M.11.1 PURSUIT TASK

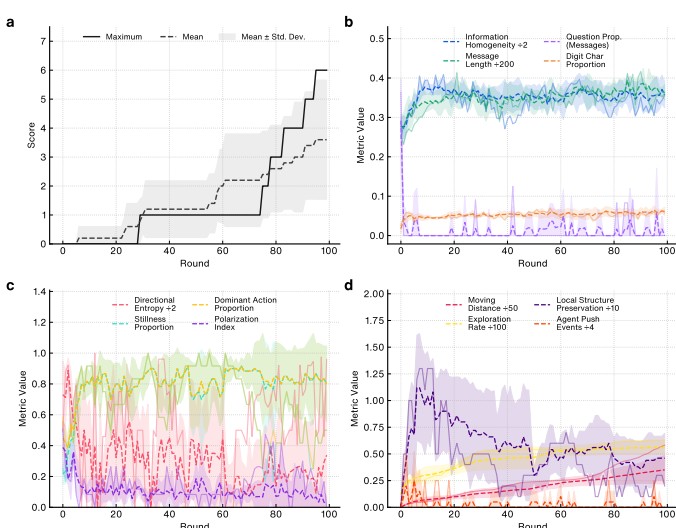

Figure S.85: **Metrics for o3-mini on the Pursuit task.**

### M.11.2 SYNCHRONIZATION TASK

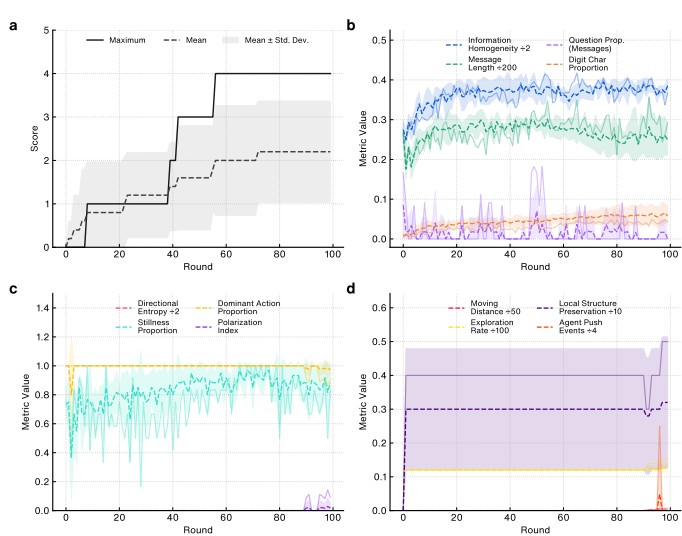

Figure S.86: **Metrics for o3-mini on the Synchronization task.**

### M.11.3 FORAGING TASK

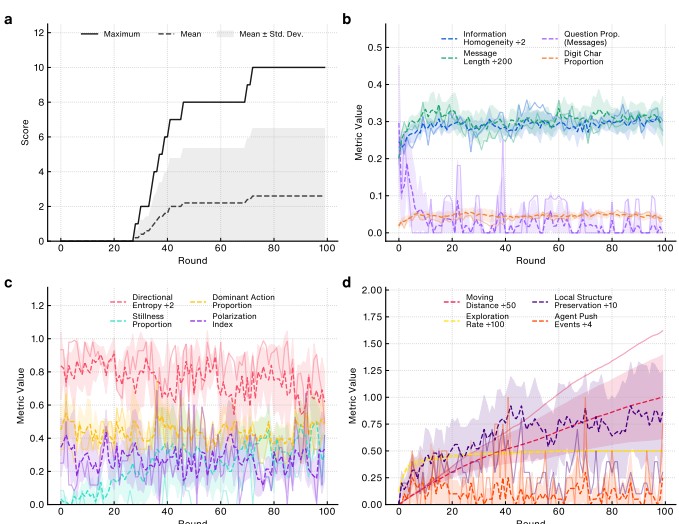

Figure S.87: **Metrics for `o3-mini` on the Foraging task.**

### M.11.4 FLOCKING TASK

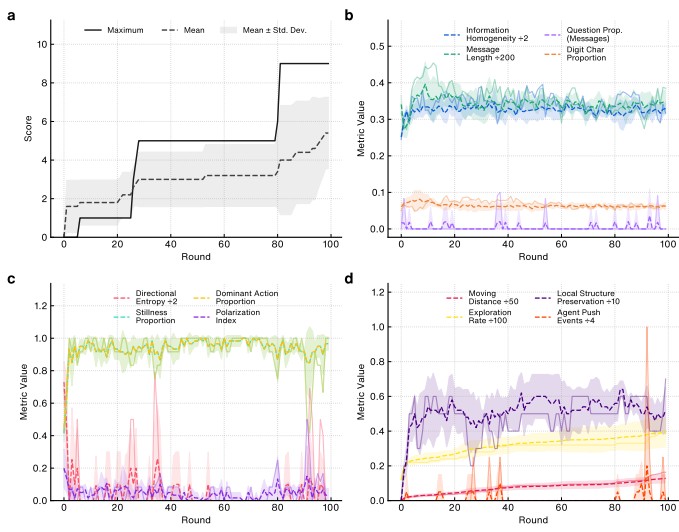

Figure S.88: **Metrics for `o3-mini` on the Flocking task.**

### M.11.5 TRANSPORT TASK

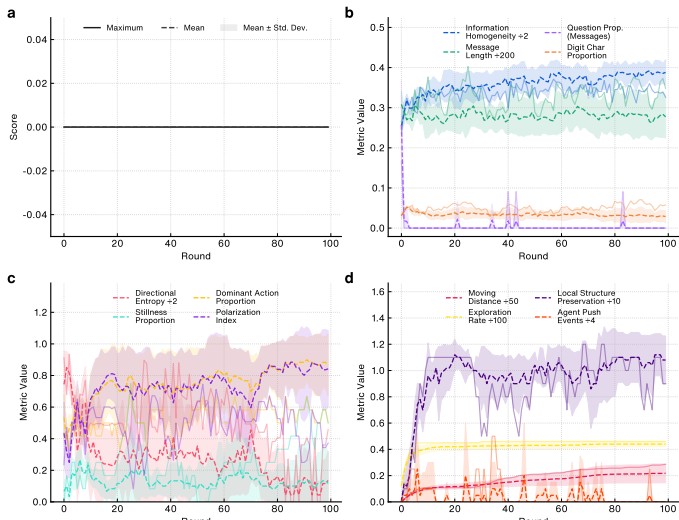

Figure S.89: **Metrics for `o3-mini` on the Transport task.**

## M.12 o4-mini

### M.12.1 PURSUIT TASK

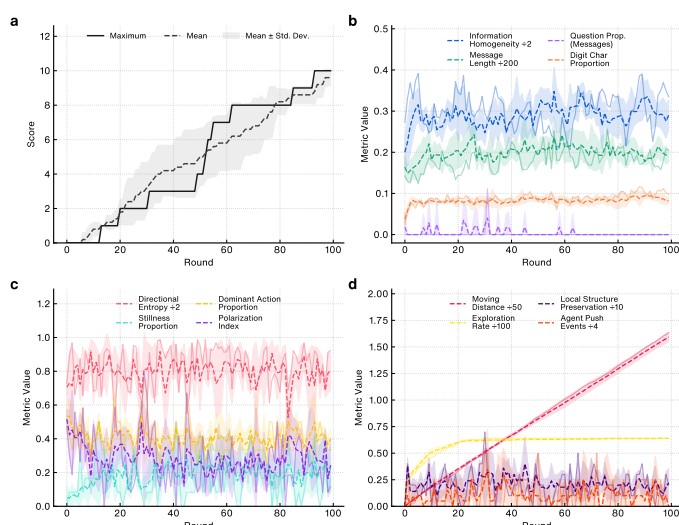

Figure S.90: **Metrics for o4-mini on the Pursuit task.**

### M.12.2 SYNCHRONIZATION TASK

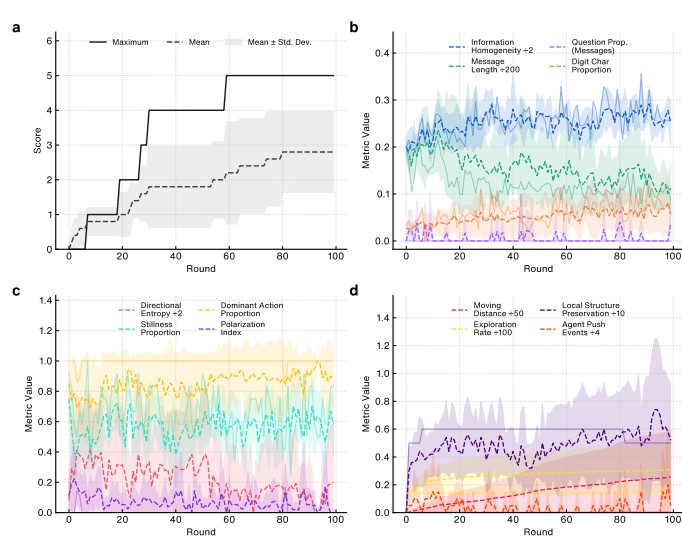

Figure S.91: **Metrics for o4-mini on the Synchronization task.**

### M.12.3 FORAGING TASK

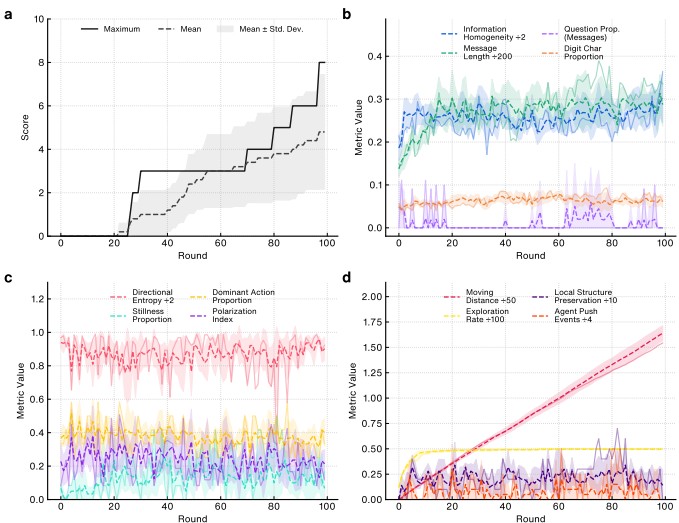

Figure S.92: **Metrics for `o4-mini` on the Foraging task.**

### M.12.4 FLOCKING TASK

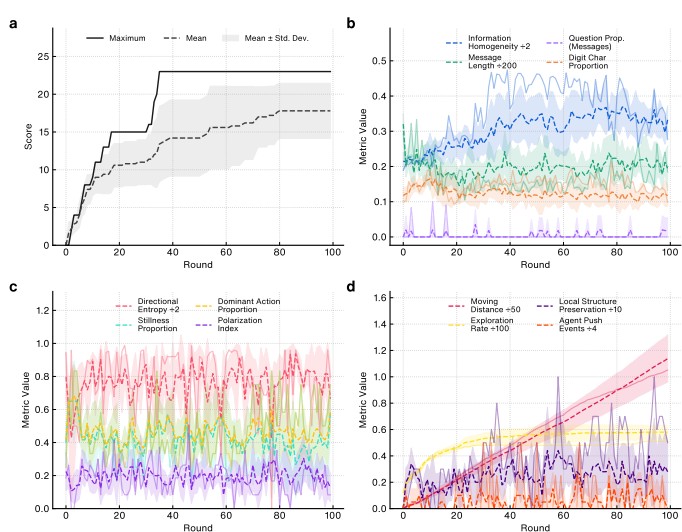

Figure S.93: **Metrics for `o4-mini` on the Flocking task.**

## M.12.5   TRANSPORT TASK

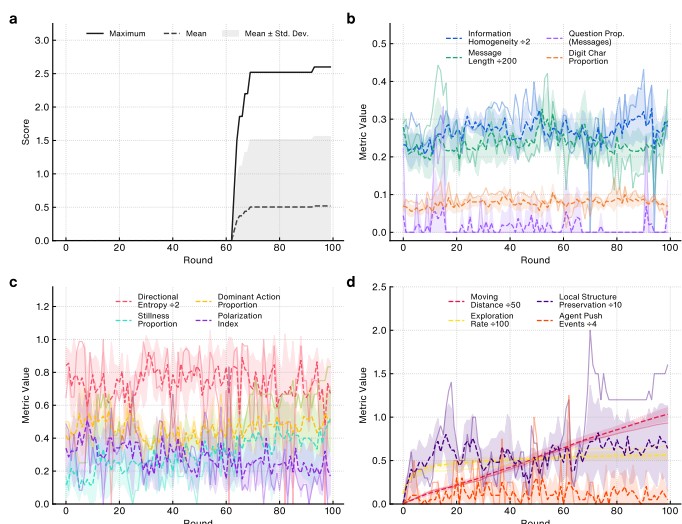

Figure S.94: **Metrics for `o4-mini` on the Transport task.**

## M.13 QWQ-32B

### M.13.1 PURSUIT TASK

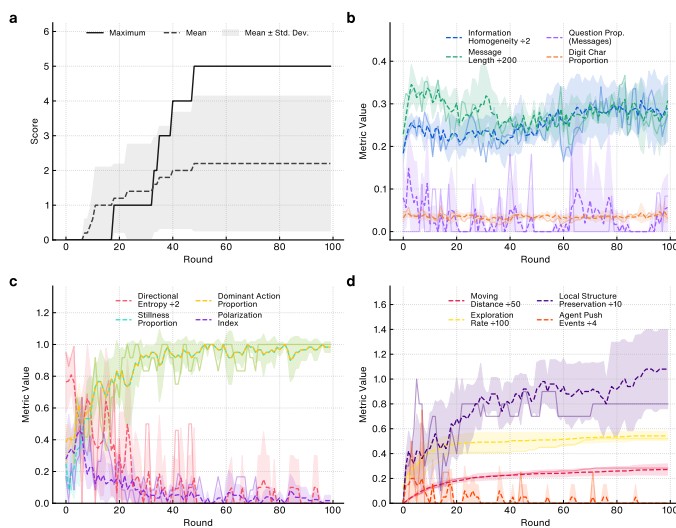

Figure S.95: **Metrics for qwq-32b on the Pursuit task.**

### M.13.2 SYNCHRONIZATION TASK

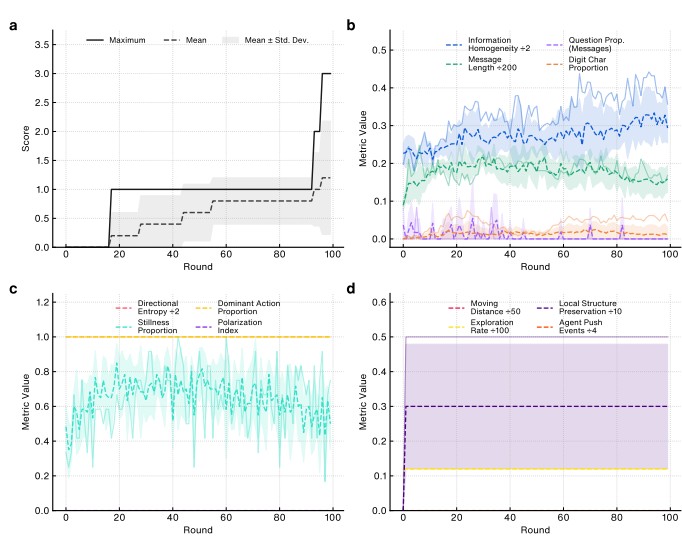

Figure S.96: **Metrics for qwq-32b on the Synchronization task.**

### M.13.3 FORAGING TASK

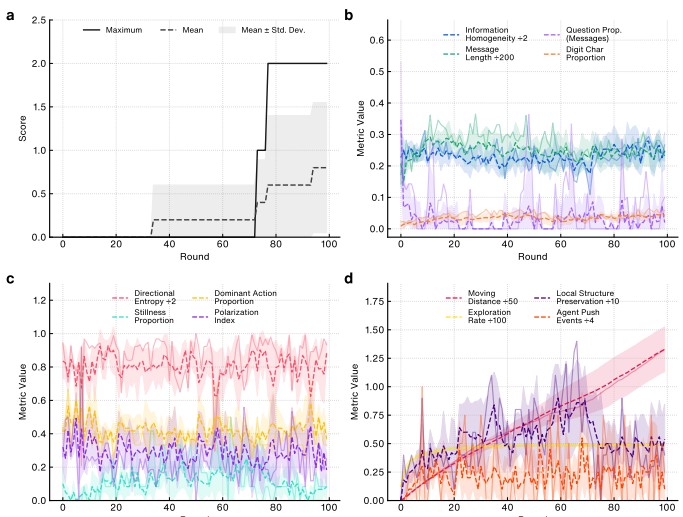

Figure S.97: **Metrics for `qwq-32b` on the Foraging task.**

### M.13.4 FLOCKING TASK

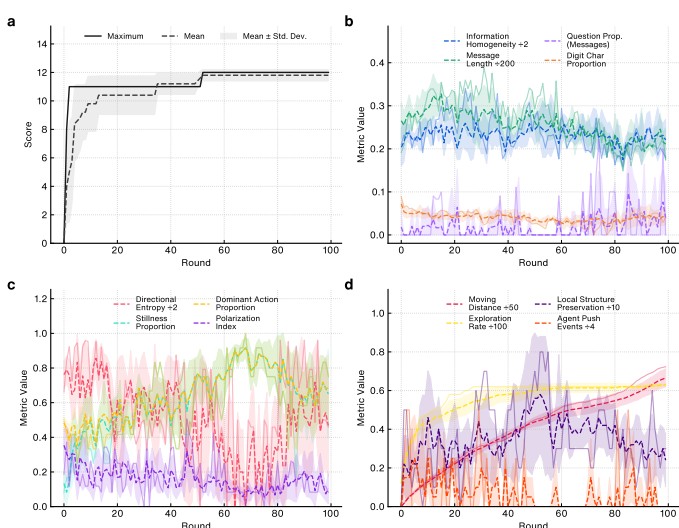

Figure S.98: **Metrics for `qwq-32b` on the Flocking task.**

### M.13.5 TRANSPORT TASK

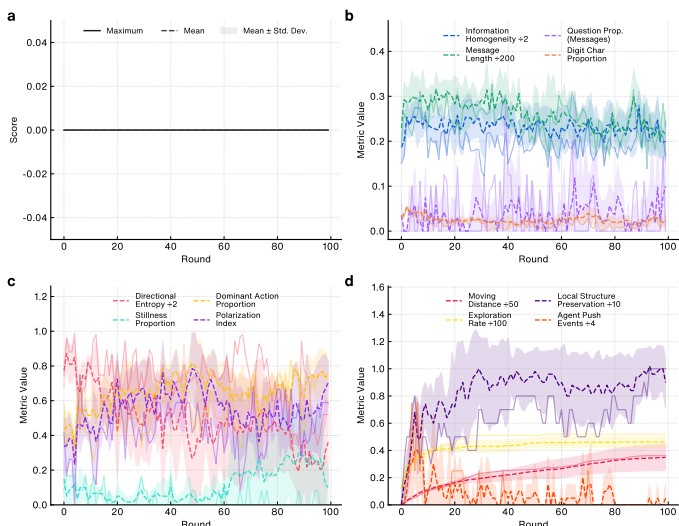

Figure S.99: **Metrics for `qwq-32b` on the Transport task.**

