# OpenReview forum: "Benchmarking LLMs' Swarm intelligence"
_ICLR.cc/2026/Conference — ICLR 2026 Conference Withdrawn Submission_

### Official Review · Reviewer_LbnN · 2025-10-21

**Soundness:** 2
**Presentation:** 3
**Contribution:** 2
**Rating:** 4
**Confidence:** 4

**Summary:**

The paper introduces **SwarmBench**, a benchmark framework for evaluating the collaboration abilities of **LLM-based agents** in decentralized swarm settings. Each agent operates independently with a limited local view and communicates through local messages. The study evaluates contemporary LLMs across multiple classic swarm coordination tasks, including **Pursuit**, **Synchronization**, **Foraging**, **Flocking**, and **Transport**.

The authors analyze group dynamics and failure modes, identifying issues such as limited memory and inefficient coordination. Through Action Attribution Analysis, they show that agents’ actions are more strongly influenced by received messages than by their own visual observations, revealing a gap between local responsiveness and global coordination. Overall, the paper provides a systematic framework for understanding and benchmarking LLM-driven swarm intelligence.

**Strengths:**

1. This paper is the first to systematically evaluate the collaborative capabilities of LLM-based agents in swarm settings. The proposed benchmark environments capture classic coordination scenarios with broad applicability across multiple domains.
2. The paper is well-structured and clearly written, making it easy to follow the methodology and experimental findings.
3. The ablation studies are thoughtfully designed and provide meaningful insights. In particular, the analyses of local perception range and action attribution effectively reveal the key bottlenecks and future directions in developing collaborative LLM agents.

**Weaknesses:**

1. **The setting is not real enough for swarm tasks.** In particular, the benchmark enforces anonymous, purely local broadcast messages, which preclude agents from maintaining stable partner identities even when they remain within each other’s field of view. This design blocks **consistent, neighbor-specific conventions (e.g., leader–follower handoffs, partner lock-on, trust updating)**, thereby underestimating the achievable coordination of embodied multi-agent systems with **local but consistent** IDs (or persistent neighbor association).
2. The current memory design for the agent is too naive. A short fixed-length history buffer
is unlikely to support **realistic long‐horizon planning or multi-step strategy adaptation for any LLMs.** To accurately measure the performance of LLMs, I recommend evaluating LLM agents with a stronger memory regime. For example, they might introduce a multi‐tier memory (working buffer + episodic store).
3. **Limited interaction between agents.** The only available form of interaction among agents is local communication, which is insufficient to capture the complexity of behaviors observed in swarm settings. Common interaction types in real-world scenarios, such as direct messaging and object passing, are not supported in this framework.

**Questions:**

1. See Weaknesses.
2. I found your Action Attribution Analysis in Section 4.4 particularly interesting — especially the finding that agents’ actions are more strongly influenced by received messages than by local observations. In my opinion, this is a major issue in developing collaborative agents. Have you considered developing mechanisms that promote action consistency over time to stabilize agents’ individual policies?
3. Could the author explain why some LLMs perform well in certain environments but not in general (for example, Claude-3.7-Sonnet on the Synchronization task)? It is important to understand which foundational abilities (e.g., reasoning, spatial perception, and effective communication) are crucial for developing swarm agents.

---

### Official Review · Reviewer_eogw · 2025-10-31

**Soundness:** 2
**Presentation:** 1
**Contribution:** 2
**Rating:** 4
**Confidence:** 2

**Summary:**

The paper proposes SwarmBench, a 2D grid world benchmark grounded in swarm intelligence constraints, designed to assess emergent decentralized coordination in LLM swarms under strict perception and communication constraints. The paper evaluates 13 LLMs and makes a lot of analysis showing that current LLMs struggle with robust collective behavior under strict decentralization and local information limits.

**Strengths:**

- Benchmarking LLMs' capability in coordination and self-organization at the swarm level is important.

- The paper is well-motivated.

- There are tons of results and analyses in the Appendix.

**Weaknesses:**

- The main paper doesn't convey much information; all details are deferred to the Appendix, including the critical part of how the benchmark is designed, what the observation and action space for agents are, and what the metric is. With 80+ pages of the Appendix, it's hard to evaluate the true value of the paper.

- The benchmark seems more designed for testing llm agents rather than LLMs directly. Without memory or planning structures, the tasks seem prohibitively challenging.

- No valuable insights gained after reading the main paper and glancing over the appendix.

- A better organization of the results is needed. E.g. Do reasoning models behave significantly differently?

**Questions:**

What's the major takeaway of the benchmarking results?

---

### Official Review · Reviewer_qHCn · 2025-10-31

**Soundness:** 3
**Presentation:** 1
**Contribution:** 3
**Rating:** 4
**Confidence:** 4

**Summary:**

The authors propose a benchmark for multi-agent AI-agents (including LLMs) aimed at testing the “swarm intelligence” of agents in a network (a 2D lattice, in this case), i.e., their capability to coordinate to solve complex tasks. The settings are represented as grids (standard in RL and MARL) and allow for local communication.
The experiments show that LLMs still struggle with basic coordination mechanisms in long-term planning.

**Strengths:**

The article proposes a useful benchmark, and the code is released as an open-source toolkit, making it easier to replicate the experiments and use it to train LLMs and other agents on the task.

The experiments are run on several models and, apart from Figure 3, the others and the Results in general are easy to read and understand.

A Strength, that is also a Weakness, is how results are presented.
The Appendix contains most of the interesting results.
For example, Appendix L5 should be in the main paper, and the authors should discuss why scaling leads to zero performance (Appendix L2). The authors can remove Figures like Fig. 2 and 3, which are not that important to understand the paper.

**Weaknesses:**

While the benchmark is indeed useful and shows some coordination failures of top-performing LLMs, the paper does not propose any mechanism to mitigate the aforementioned issues, and some benchmarks’ results are difficult to interpret.

Some insights are interesting (Claude outperforms any other model at Synchronization, all the models are bad at Transport): on the other hand, the paper lacks an insightful analysis of the reasons behind this failure (beyond saying that models cannot do long-horizon planning), and potential ways to mitigate such issues.

Figure 3 is confusing and does not convey the message the authors were trying to convey. I suggest they enlarge it. Figure 2 and also be moved to the Appendix: showing an example of the code is indeed useful, but the authors should put more emphasis on the design choices and results, in my opinion.

The comparison with humans should also be in the main paper and discussed. For example, Table S3 shows that humans perform poorly on Transport: that makes me think that the task is not formulated correctly (given the initial condition and the agents’ position, that task should be solvable with ~100% accuracy by an algorithm), as no baseline or algorithm/LLM achieves an acceptable score. The authors mention ARC-AGI as an inspiring method for their benchmark; yet, humans perform very well on that task, and that’s what makes it valuable.

To conclude, while I believe that this article has more reasons to be accepted than rejected, I remain with the question, “What did we learn and what can we do to improve on these tasks?”. I recommend that the authors think about potential mitigating strategies to improve the coordination capabilities of their models, or, if that is out of the scope, to discuss and identify the concrete, main reasons of failure (Section 4.3 is mainly quantitative (no metric is reported) and not particularly informative on the global reasons of failure). I read Appendix L5, and I believe that it should be expanded and put in the main paper, alongside some future directions, hints, and methods to improve and mitigate such issues.

**Questions:**

Q1. As someone working in similar topics, have you tried to optimise the amount of information you provide in each prompt? Each agent receives a lot of information (Appendix C, prompt design), and that may confuse small/non-reasoning models.

Q2. In general, the environments are described as textual grids, and that may not be optimal (we, as humans, do not observe or reason on a grid). Have you thought about other representation methods? For example, some sort of multi-modality (images).

Q3. Have the authors tried networks with heterogeneous LLMs? That would be interesting to see if there are agents that block or create issues, etc.

Q4. Why is Claude so good at synchronisation compared to other models?

---

### Official Review · Reviewer_Rwrb · 2025-11-03

**Soundness:** 3
**Presentation:** 3
**Contribution:** 4
**Rating:** 6
**Confidence:** 4

**Summary:**

This paper introduces SwarmBench, a benchmark designed to evaluate the swarm intelligence of Large Language Models (LLMs) under strict decentralized coordination constraints. Inspired by natural swarm systems, the benchmark simulates five multi-agent tasks (Pursuit, Synchronization, Foraging, Flocking, and Transport) in a 2D grid world. Agents operate with restricted local perception and limited communication, testing the models' ability to exhibit emergent coordination behavior. The authors conduct a comprehensive zero-shot evaluation of thirteen LLMs, providing insights into model capabilities, failure modes, and communication strategies. SwarmBench is released as an open-source, extensible platform for research on decentralized multi-agent systems powered by LLMs.

**Strengths:**

1. Originality: SwarmBench is a novel benchmark specifically targeting decentralized, emergent coordination, a largely unexplored dimension in LLM-based MAS.

2. Quality: The experimental framework is comprehensive, with well-defined tasks, robust baselines, and rich metrics.

3. Significance: Highlights the limitations of current LLMs in real-world decentralized coordination scenarios, guiding future research.

4. Clarity: Visualizations and benchmark design are intuitive, facilitating understanding of complex emergent behaviors.

**Weaknesses:**

1. Limited exploration of training regimes: The exclusive use of zero-shot evaluation neglects the potential of in-context learning or RL fine-tuning to improve coordination.

2. Shallow communication analysis: Although the paper emphasizes the role of local communication, it does not deeply investigate language content or its evolution over time.

3. Scalability questions: While the benchmark is extensible, the current use case is limited to abstract 2D environments and may not generalize to higher-fidelity domains.

**Questions:**

1. How does performance change if agents are given memory beyond five rounds or allowed limited global information?

2. Have the authors considered curriculum learning to gradually improve swarm coordination capabilities?

3. Could learned communication protocols (e.g., via token supervision) enhance performance over purely emergent protocols?

---

### Note · Authors · 2025-11-12

I have read and agree with the venue's withdrawal policy on behalf of myself and my co-authors.